# Polynomial Convergence of Riemannian Diffusion Models

**Xingyu Xu, Ziyi Zhang, Yorie Nakahira, Guannan Qu, Yuejie Chi**
Department of Electrical and Computer Engineering
Carnegie Mellon University
Pittsburgh, PA 15213, USA
{xingyuxu,ziyizhan,ynakahir,gqu,yuejiec}@andrew.cmu.edu

## Abstract

Diffusion models have demonstrated remarkable empirical success in recent years and are considered one of the state-of-the-art generative models in modern AI. These models consist of a forward process, which gradually diffuses the data distribution to a noise distribution spanning the whole space, and a backward process, which inverts this transformation to recover the data distribution from noise. Most of the existing literature assumes that the underlying space is Euclidean. However, in many practical applications, the data are constrained to lie on a submanifold of Euclidean space. Addressing this setting, De Bortoli et al. (2022) introduced Riemannian diffusion models and proved that using an exponentially small step size yields a small sampling error in the Wasserstein distance, provided the data distribution is smooth and strictly positive, and the score estimate is $L_\infty$-accurate. In this paper, we greatly strengthen this theory by establishing that, under $L_2$-accurate score estimate, a *polynomially small stepsize* suffices to guarantee small sampling error in the total variation distance, without requiring smoothness or positivity of the data distribution. Our analysis only requires mild and standard curvature assumptions on the underlying manifold. The main ingredients in our analysis are Li-Yau estimate for the log-gradient of heat kernel, and Minakshisundaram-Pleijel parametrix expansion of the perturbed heat equation. Our approach opens the door to a sharper analysis of diffusion models on non-Euclidean spaces.

## 1 Introduction

Initially introduced by Sohl-Dickstein et al. (2015) and later advanced by Song & Ermon (2019); Ho et al. (2020); Dhariwal & Nichol (2021), diffusion model has become one of the bedrocks in generative modeling across a variety of application domains such as vision, video, speech, and many others. On a high level, diffusion models generate samples from a target distribution by operating on two stochastic processes:

1. A *forward* process

$$X_0 \xrightarrow{\text{add noise}} X_1 \xrightarrow{\text{add noise}} \cdots \xrightarrow{\text{add noise}} X_T,$$

   where $X_0$ is sampled from the target distribution $p_0$ in $\mathbb{R}^d$, and $X_T$ resembles pure noise.

2. A *reverse* process

$$Y_T \xrightarrow{\text{denoise}} Y_{T-1} \xrightarrow{\text{denoise}} \cdots \xrightarrow{\text{denoise}} Y_0,$$

   where $Y_T$ starts from pure noise, and gradually removes the noise, so that at $Y_0$, we recover a new sample from a distribution close to $p_0$.

The reverse process is built to recover the target data distribution by step-wise reversing the forward process, with a goal of matching the probabilities $Y_t \approx X_t$ in distribution for $t \in \{T, \ldots, 1\}$. Leveraging the theory of backward stochastic differential equations (SDE) (Anderson, 1982; Haussmann & Pardoux, 1986), this can be formally achieved as soon as the as long as the *score function*, i.e.,

the log-gradient of the the marginal density of the forward process, becomes available, which can be estimated via score matching (Hyvärinen & Dayan, 2005).

Tremendous recent progresses have been made in understanding the convergence of diffusion models in the Euclidean space, e.g. Lee et al. (2023); Chen et al. (2023); Benton et al. (2024); Li et al. (2024), which establish near-tight polynomial iteration complexities of discrete-time samplers under $L_2$-accurate score estimates and mild assumptions of the data distribution. These convergence results provide strong justifications to the empirical success of diffusion models for generating from complex multi-modal distributions.

Many scientific domains, however, are intrinsically *non-Euclidean*; examples include orientations on $SO(3)$, directions on spheres, toroidal angles, articulated poses, and symmetric positive definite (SPD) matrices are naturally modeled on Riemannian manifolds (Piggott & Solo, 2016; Muniz et al., 2022). Recently, there has been an increasing interest in effectively sampling from distributions supported on manifolds and providing theoretical guarantees (Girolami & Calderhead, 2011; Gatmiry & Vempala, 2022; Li & Erdogdu, 2023; Guan et al., 2025). Although sampling on manifolds has been studied extensively (Cheng et al., 2023), extending diffusion models to manifolds requires careful treatments to incorporate the manifold constraints into both the time-inhomogeneous forward and reverse processes, with selected attempts in De Bortoli et al. (2022); Huang et al. (2022); Lou et al. (2023); Liu et al. (2023); Fishman et al. (2023).

One notable development is De Bortoli et al. (2022), who introduced *Riemannian Score-Based Generative Models* (RSGMs) with convergence guarantees in the Wasserstein distance. Specifically, they established a time-reversal diffusion process for geometric Brownian motion on manifolds, which can be similarly learned via score matching (Hyvärinen & Dayan, 2005). While groundbreaking, their convergence bound suffers from a few caveats: 1) it requires an exponentially small stepsize, leading to a possibly exponential iteration complexity in some of the manifold parameters; 2) it requires $L_\infty$-accurate score estimates, which are impractical in deep learning; and 3) the data distribution is required to be smooth and strictly positive on compact manifolds. This naturally raises the following questions:

*Can we achieve polynomial iteration complexity for manifold diffusion models using $L_2$-accurate score estimates under mild data assumptions?*

## 1.1 OUR CONTRIBUTION

We provide a discrete-time analysis of the RSGM sampler in De Bortoli et al. (2022), assuming $L_2$-accurate score estimates. Under mild geometric conditions of the manifold without assuming smooth or strictly positive data densities, we establish that *polynomial* stepsizes suffice for accurate sampling on manifolds in *total variation* (TV). More precisely, for some $\varepsilon > 0$, the TV error between the distribution of the output $Y_0$ and $p_\delta$, that is, an approximation to $p_0$ with early-stopping time $\delta > 0$, obeys

$$\mathsf{TV}\big(p_\delta, \mathsf{Law}(Y_0)\big) \lesssim \varepsilon + \varepsilon_{\mathsf{score}} + \sqrt{hT}\,\mathrm{poly}(d, \delta^{-1}),$$

as long as the horizon satisfies $T \gtrsim \lambda_1^{-1}(d \log d + \log(d/\varepsilon))$. Here, $d$ is the dimension of $\mathcal{M}$, $\lambda_1$ is the spectral gap of $-\Delta_M$, $h > 0$ is the stepsize, and $\varepsilon_{\mathsf{score}}$ is the $L_2$ score estimation error.[1] This bound suggests that a polynomial stepsize is sufficient: take $T \asymp \lambda^{-1}(d \log d + \log(d/\varepsilon))$ and $h = \frac{\varepsilon^2}{\mathrm{poly}(d, \delta^{-1})T}$, then the TV error is bounded by $\varepsilon + \varepsilon_{\mathsf{score}}$ after an iteration complexity of

$$N = T/h \asymp \mathrm{poly}(d, \delta^{-1})/(\lambda_1 \varepsilon)^2.$$

This conveys a much more benign message about the efficiency of Riemannian diffusion models, compared with the iteration complexity in De Bortoli et al. (2022) that scales exponentially with the dimension $d$, under relaxed assumptions on both the data distribution and the score estimates.

**Techniques.** Our proof highlights three ingredients: (i) high-probability Li-Yau gradient bounds for the manifold heat kernel together with early stopping to control $\|\nabla \log p_t\|$ without assuming positivity/smoothness of $p_0$; (ii) a localization scheme that "freezes" drifts across nearby tangent

---

[1]For simplicity, we omitted polynomial dependence on manifold geometry parameters and poly-log dependence on $\varepsilon$. The complete version can be found in Theorem 1.

| Work | Structure | Metric | Iteration complexity | Data distribution |
|------|-----------|--------|----------------------|-------------------|
| Benton et al. (2024) | Euclidean | TV | $\widetilde{O}(d/\varepsilon^2)$ | bounded moment |
| Li et al. (2024) | Euclidean | TV | $\widetilde{O}(\mathrm{poly}(d)/\varepsilon)$ | bounded support |
| Li & Yan (2025) | Euclidean | TV | $\widetilde{O}(d/\varepsilon)$ | bounded moment |
| De Bortoli et al. (2022) | Manifold | $W_p$ | $\widetilde{O}(\exp(O(d))/\varepsilon^{-1/\lambda_1})$[a] | smooth, strictly positive |
| This work | Manifold | TV | $\tilde{O}\big(\frac{\mathrm{poly}(d)}{\lambda_1^2 \varepsilon^2}\big)$ | None (early stopping) |

---

[a]The original $W_p$ error in De Bortoli et al. (2022) is stated in the form of $\widetilde{O}\big(C\mathrm{e}^{-\lambda_1 T} + \mathrm{e}^T\sqrt{h}\big)$, where $C$ is defined in Proposition C.6 therein, which in turn is specified by Urakawa (2006, Proposition 2.6) as the supremum of $t^{d/2}H(t,x,y)$, where $H$ is the heat kernel on the manifold. In general, the best estimate for this is due to Li-Yau (Li & Yau, 1986), which gives $C \leq \mathrm{e}^{O(d)}$. To achieve $\varepsilon$-error, we must set $T = \lambda_1^{-1}(\Omega(d) + \log \varepsilon^{-1})$ and $h = \mathrm{e}^{-2T}\varepsilon^2$, then the iteration complexity $T/h$ has the claimed form.

Table 1: Comparison of the current theoretical guarantees on diffusion probabilistic models on Euclidean spaces and manifolds. Here, $\lambda_1 > 0$ is the spectral gap of the Laplace-Beltrami operator.

spaces but preserves continuous Brownian motion (BM), to separate the effects of discretizing scores and BM; and (iii) a quantitative estimates for Minakshisundaram–Pleijel parametrix that controls one-step deviations between the manifold heat flow and its discretized proxy. These components allow us to handle the discretization errors sharply to avoid exponential dependence.

### 1.2 RELATED WORKS

**Non-asymptotic convergence for Euclidean diffusion models.** Early convergence analyses of diffusion models require $L_\infty$-accurate score estimates (De Bortoli et al., 2021). For stochastic samplers such as DDPM (Ho et al., 2020), early bounds under Lipschitz/smoothness assumptions of the data distribution admit an $O(T^{-\frac{1}{2}})$ iteration complexity in the total variation distance assuming $L_2$-accurate score estimates (Chen et al., 2023), with subsequent analyses relaxing the Lipschitz assumption yet retaining the same complexity (Lee et al., 2023; Benton et al., 2024; Li et al., 2024). More recently, Li & Yan (2025) has improved the iteration complexity to $\widetilde{O}(T^{-1})$. For deterministic samplers, Chen et al. (2023) established polynomial convergence with exact scores, and Li et al. (2024) established a convergence rate of $O(T^{-1})$ under $L_2$-accurate scores. See Beyler & Bach (2025); Liang et al. (2024); Li & Jiao (2024) for additional analyses that established convergence in the Wasserstein distance and improved discrete-time rates. Several works (Li & Yan, 2024; Liang et al., 2025; Huang et al., 2024; Potaptchik et al., 2024) also developed non-asymptotic convergence rates of diffusion models under the manifold hypothesis, suggesting diffusion models are adaptive to low-dimensional structures. This line of work should not be confused with ours, where the diffusion process is designed specifically to be constrained on the manifold.

**Sampling on Riemannian manifold.** Cheng et al. (2022; 2023) analyzed the geometric Euler–Maruyama (EM) discretization for time-homogeneous SDEs, and proved a polynomial complexity guarantee under dissipative-distant geometric assumptions on the manifold. See also Bharath et al. (2025) for follow-ups. Guan et al. (2025) proposed a Riemannian proximal sampler with convergence guarantees under the log-Sobolev inequality. Various sampling algorithms are also studied for a related problem known as sampling from constrained spaces (Srinivasan et al., 2024; Ahn & Chewi, 2021). Nonetheless, convergence analyses of Riemannian diffusion models under general data distributions remain highly limited, with De Bortoli et al. (2022) being the only prior work with non-asymptotic convergence rates.

## 2 BACKGROUNDS

### 2.1 DIFFUSION MODELS ON EUCLIDEAN SPACE

We briefly recall diffusion processes on $\mathbb{R}^d$. Let $(W_t)_{t \geq 0}$ be a standard Brownian motion in $\mathbb{R}^d$.

**Forward SDE and Fokker–Planck.** Given a drift term $b_t : \mathbb{R}^d \to \mathbb{R}^d$, the forward process $(X_t)_{t \in [0,T]}$ solves the Itô SDE

$$\mathrm{d}X_t = b_t(X_t)\,\mathrm{d}t + \mathrm{d}W_t, \qquad X_0 \sim p_0.$$

Let $p_t$ denote the law of $X_t$, then $p_t$ satisfies the Fokker–Planck equation $\partial_t p_t = -\nabla(b_t p_t) + \frac{1}{2}\Delta p_t$. In the driftless setting where $b_t \equiv 0$, the marginal $p_t = p_0 * \varphi_t$ is a Gaussian smoothing of $p_0$ with kernel $\varphi_t(z) = (2\pi t)^{-d/2} \exp\left(-\|z\|^2/(2t)\right)$.

**Score and reverse process.** The *score* of the forward process at time $t$ is defined as $s_t(x) \coloneqq \nabla \log p_t(x)$. The time-reversal identity (Anderson, 1982) yields a reverse-time process $(Y_t)_{t \in [0,T]}$ whose marginals match those of $(X_t)_{t \in [0,T]}$ which solves the reverse-time SDE (note that $t$ flows from $T$ to 0):

$$dY_\tau = [-b_\tau(Y_t) + s_\tau(Y_t)]\,\mathrm{d}t + \mathrm{d}W_t, \qquad \tau = T - t, \quad Y_T \sim p_{X_T}.$$

**Discretization with approximate score.** Let $0 = t_0 < t_1 < \cdots < t_N = T$, where $t_i - t_{i-1} =: h$ be a time grid. In practice, exact score function is often unavailable. Instead, we use an approximation $\widehat{s}_t$ trained via score matching (Hyvärinen & Dayan, 2005). The Euler–Maruyama discretization of the reverse-time SDE above is given by

$$y_{k-1} = y_k + h\left[-b_{t_k}(y_k) + \widehat{s}_{t_k}(y_k)\right] + \sqrt{h}g_k, \quad g_k \sim \mathcal{N}(0, I_d).$$

In our driftless setting, this reduces to

$$y_{k-1} = y_k + h\widehat{s}_{t_k}(y_k) + \sqrt{h}g_k, \quad g_k \sim \mathcal{N}(0, I_d).$$

## 2.2 Geometry and notation

We assume some familiarity with Riemannian geometry, and make use of standard notation. Please refer to Jost (2017); Petersen (2006) for a more in-depth treatment. In particular, we use $\alpha, \beta, \xi, \zeta$, etc., to index *coordinate representation* of tensors, and assume Einstein's summation convention. Let $(\mathcal{M}, g)$ be a connected, compact $d$-dimensional Riemannian manifold, with geodesic distance $\rho(\cdot, \cdot)$ and volume measure $\mu$. We assume $\mu(\mathcal{M}) = 1$. The Levi–Civita connection is denoted by $\nabla$, and the Laplace–Beltrami operator by

$$\Delta_{\mathcal{M}} f \coloneqq \nabla_\alpha \nabla^\alpha f.$$

We use $T_x\mathcal{M}$ for the tangent space at $x$ and use $\exp_x : T_x\mathcal{M} \to \mathcal{M}$ for the exponential map and $\log_x$ for its local inverse on the normal neighborhood of $x$. Furthermore, we define *geodesic diameter* of $(\mathcal{M}, g)$ is

$$\mathrm{Diam}(\mathcal{M}) \coloneqq \sup_{x,y \in \mathcal{M}} \rho(x, y),$$

where $\rho(\cdot, \cdot)$ is the geodesic distance induced by $g$. We further denote $\mathrm{Rm}$ as the Riemannian curvature tensor. Geodesic ball centered at $x$ with radius $r$ is denoted $B_x(r)$.

We use the *total variation* (TV) and the *Kullback-Leibler* (KL) distance to measure the discrepancy between two distributions $p, q$:

$$\mathsf{TV}(p, q) = \int_{\mathcal{M}} |\mathrm{d}p - \mathrm{d}q|, \quad \mathsf{KL}(p \| q) = \int_{\mathcal{M}} \left(\log \frac{\mathrm{d}p}{\mathrm{d}q}\right)\mathrm{d}p.$$

## 2.3 Heat flow, Brownian motion, and diffusion on $\mathcal{M}$

We also recall the setup for SDE and diffusion processes on Riemannian manifolds introduced in De Bortoli et al. (2022); Cheng et al. (2023). Let $(W_t)_{t \geq 0}$ be a standard Brownian motion in $\mathbb{R}^d$ and $U_x : \mathbb{R}^d \to T_x\mathcal{M}$ any orthonormal frame at $x$. The *Geometric Brownian motion* solves

$$\mathrm{d}X_t = U_{X_t} \circ \mathrm{d}W_t,$$

where $\circ$ denotes Stratonovich integral, and its transition density $p_t(x, y)$ with respect to $\mu$ solves the heat equation

$$\partial_t p_t(\cdot, y) = \frac{1}{2}\Delta_{\mathcal{M}} p_t(\cdot, y).$$

---

**Algorithm 1** Riemannian Score-Based Generative Models (RSGM)

---

1: Manifold $(\mathcal{M}, g)$; score $\widehat{s}_t(x)$; early stopping time $\delta > 0$; reverse time grid $\delta = t_0 < t_1 < \cdots < t_N = T$; step size $h = t_k - t_{k-1}$; initial $x_N \sim \mu$ (uniform distribution);
2: **for** $k \in \{N, \ldots, 1, 0\}$ **do**
3:     Choose an orthonormal frame $U_k$ at $Y_k$, which is a linear map from $\mathbb{R}^d$ to $T_{Y_k}\mathcal{M}$.
4:     $\xi_k \sim \mathcal{N}(0, I_d)$ in $\mathbb{R}^d$;    $G_k \leftarrow U_k \xi_k \in T_{Y_k}\mathcal{M}$.
5:     $b_k \leftarrow \widehat{s}_{t_k}(Y_k) \in T_{Y_k}\mathcal{M}$
6:     $\Delta_k \leftarrow h b_k + \sqrt{h}\, G_k \in T_{Y_k}\mathcal{M}$
7:     **if** $\|\Delta_k\| \leq h^{1/4}$ **then**
8:         $Y_{k-1} \leftarrow \exp_{Y_k}(\Delta_k)$
9:     **else**
10:        $Y_{k-1} \sim \mu$
11: **return** $Y_0$

---

Equivalently, Brownian motion can be defined abstractly as the solution to the martingale problem for the operator $\frac{1}{2}\Delta_\mathcal{M}$. Concretely, for any $f \in C^\infty([0,\infty) \times \mathcal{M})$, the process

$$M_t^f := f(t, X_t) - f(0, X_0) - \int_0^t \left(\partial_s + \frac{1}{2}\Delta_\mathcal{M}\right) f(s, X_s)\mathrm{d}s$$

is a martingale with respect to the natural filtration of $X$. More generally, a forward diffusion process with drift is given by

$$\mathrm{d}X_t = b_t(X_t)\,\mathrm{d}t + U_{X_t} \circ \mathrm{d}W_t,$$

with Fokker–Planck equation $\partial_t p_t = -\nabla(b_t p_t) + \frac{1}{2}\Delta_\mathcal{M} p_t$. Note that in this setting, the following process is a martingale for smooth $f$:

$$M_t^f := f(t, X_t) - f(0, X_0) - \int_0^t \left(\partial_s f + \langle b_t, \nabla f \rangle + \frac{1}{2}\Delta_\mathcal{M} f\right)(s, X_s)\mathrm{d}s. \tag{1}$$

Let $p_t$ denote the density of $X_t$ w.r.t. $\mu$, and define the *score* $s_t := \nabla \log p_t$. The time-reversal identity on manifolds yields a reverse SDE:

$$\mathrm{d}\widetilde{X}_\tau = (-b_t(\widetilde{X}_\tau) + \nabla \log p_\tau(\widetilde{X}_\tau))\mathrm{d}t + U_{\widetilde{X}_\tau} \circ \mathrm{d}W_t, \qquad \tau = T - t, \quad \widetilde{X}_T \sim p_{X_T}.$$

In practice, the score $\nabla \log p_t$ is approximated by a trained neural network $\widehat{s}_t(x)$.

Last but not least, note that on compact manifolds, $-\Delta_\mathcal{M}$ admits a spectral gap $\lambda_1 > 0$. Any initial distribution mixes to the uniform distribution $\mu$ along the heat flow with rate $\mathrm{e}^{-\lambda_1 t}$.

## 3   MAIN RESULT

In this section, for completeness, we first introduce the RSGM algorithm in De Bortoli et al. (2022). Then, we offer our polynomial convergence guarantee in Theorem 1. For simplicity, we use a driftless forward process:

$$\mathrm{d}X_t = U_{X_t} \circ \mathrm{d}W_t, \qquad X_0 \sim p_0.$$

The time-reversal identity yields the *reverse-time SDE*

$$\mathrm{d}Y_t = \nabla \log p_\tau(Y_\tau)\,\mathrm{d}t + U_{Y_\tau} \circ \mathrm{d}W_t, \quad \tau = T - t, \quad Y_T \sim p_T. \tag{2}$$

In Algorithm 1, we provide an outline of discretized reverse-time SDE on Riemannian manifold, modified from De Bortoli et al. (2022). In each reverse step $k \in \{N, \ldots, 1, 0\}$, we select an orthonormal frame $U_k$ at $y_k$, then sample Gaussian noise $\xi_k$ and lift it to the tangent space $T_{y_k}\mathcal{M}$ using the orthonormal frame, obtaining $G_k \in T_{y_k}\mathcal{M}$. Afterwards, we propose a tangent update $\Delta_k = h\widehat{s}_{t_k}(y_k) + \sqrt{h}G_k$ and the project to the manifold using the exponential map. To prevent the update from exiting the injective radius, we perform a rejection sampling step that rejects exceedingly large update. The algorithm terminates at $k = 0$ and returns the final iterate $y_0$. In this way, we ensure every update is well-defined in normal coordinates during the algorithm.

Before presenting the main theorem, we formalize the assumptions needed for the convergence guarantee.

**Assumption 1** (Regularity). *Let $(\mathcal{M}, g)$ be a connected, compact $d$-dimensional Riemannian manifold. We assume the following conditions on $\mathcal{M}$:*

(A1) **Positive injectivity radius:** *there exists some constant $K \geq 1$ such that the injective radius $\geq 1/K$.*

(A2) **Uniform curvature bounds:** *for the same constant $K$ (which can be enlarged if necessary), we have*

$$\max \left\{ \operatorname{Diam}(\mathcal{M}), \|\mathrm{Rm}\|_{L^\infty}, \|\nabla \mathrm{Rm}\|_{L^\infty}, \|\nabla^2 \mathrm{Rm}\|_{L^\infty} \right\} \leq K.$$

(A3) **Regularity of score estimates:** *there exists a polynomial $\operatorname{poly}(d, K)$, such that*

$$\|\widehat{s}_{t_k}(x)\| \leq \operatorname{poly}(d, K) \left( \|\nabla \log p_{t_k}(x)\| + t_k^{-1} \right), \quad \forall x \in \mathcal{M}.$$

In Assumption 1, we made the standard "bounded geometry" assumption; similar assumptions also occur in Cheng et al. (2022); De Bortoli et al. (2022). A positive injective radius ensures that we have sufficient room to operate on the tangent spaces as a proxy of operating on manifolds, since for every $x \in \mathcal{M}$, the exponential map $\exp_x$ is a diffeomorphism on the geodesic ball within injectivity radius. Bounds on Riemannian tensors rule out pathological cases, which helps to control the error propagation along the reverse diffusion. Lastly, compactness ensures a positive spectral gap of $\Delta_{\mathcal{M}}$ with $\lambda_1 > 0$, which is necessary to guarantee that the forward process mixes. The mild assumption (A3) on the score estimates avoids excessively large drifts in diffusion, and can be implemented easily in practice by clipping. In addition to the above, we also need a standard assumption on the score estimation error (Chen et al., 2023).

**Assumption 2** (Score estimation error). *There exists $\varepsilon_{\mathsf{score}} > 0$ such that*

$$\sum_{k=1}^{N} (t_k - t_{k-1}) \mathbb{E} \|\widehat{s}_{t_k}(Y_{t_k}) - \nabla \log p_{t_k}(Y_{t_k})\|^2 \leq \varepsilon_{\mathsf{score}}^2.$$

With the above assumptions, we are now ready to present our main convergence guarantee for RSGM, as outlined in the following TV-accuracy bound.

**Theorem 1.** *Assume Assumptions 1 and 2 hold. There exists some universal constant $C, C' > 0$ such that the following holds. If $T \geq \frac{C}{\lambda_1}(d \log(Kd) + K + \log(\frac{N}{\varepsilon}))$, then the output $Y_0$ of Algorithm 1 obeys*

$$\mathsf{TV}(p_\delta, \mathsf{Law}(Y_0)) \leq \varepsilon + C' \varepsilon_{\mathsf{score}} + \sqrt{hT} \operatorname{poly}(d, K, \delta^{-1}),$$

*where $h$ is the discretization step size, $\lambda_1 > 0$ is the mixing rate of the geometric Brownian Motion on $\mathcal{M}$, i.e., the smallest eigenvalue of $-\Delta_{\mathcal{M}}$ in $L^2(\mu)$.*

A few remarks are in order.

**Iteration complexity.** The error bound decomposes cleanly into three terms: $\varepsilon$ results from mixing of the heat semigroup at the spectral gap $\lambda_1$, $\varepsilon_{\mathsf{score}}$ captures error from imperfect score estimation, and $\sqrt{hT} \operatorname{poly}(d, K, \delta^{-1})$ is the discretization error controlled by the step size and curvature. Consequently, choosing $T \asymp \lambda_1^{-1}(d \log d + \log(d/\varepsilon))$ and $h = \frac{\varepsilon^2}{\operatorname{poly}(d, K, \delta^{-1})T}$, then the TV error is bounded by $\varepsilon + \varepsilon_{\mathsf{score}}$ after polynomially many iterations

$$N = T/h \asymp \frac{\operatorname{poly}(d, K, \delta^{-1})}{(\lambda_1 \varepsilon)^2}.$$

Compared to prior convergence rates in the Wasserstein metric (De Bortoli et al., 2022), which require exponential complexity, we achieve polynomial convergence of Riemannian diffusion models for the first time. Nonetheless, we emphasize that TV and Wasserstein distances are incomparable with each other in general, and our guarantee complements prior Wasserstein results (De Bortoli et al., 2022) by ensuring distributional closeness in a different notion with a much smaller number of iterations.

**Possible improvements.** We note that the bound established in Theorem 1 holds under very mild geometric assumptions, requiring only constraints on the injective radius and Riemannian curvature. The purpose of this study is to demonstrate that, in the manifold setting, the exponential blow-up in $T$ can be avoided and polynomial complexity can be achieved. To keep the exposition as simple as possible and to clearly highlight the key ideas, we have not attempted to optimize the current bound on the degree of the polynomial. Potential approaches for sharper bounds include: (i) a better design of discretization schedule, possibly adaptive to the manifold geoometry, and a more careful computation of discretization error, such as those in Li & Jiao (2024), Benton et al. (2024) (notably, the dependence on $\delta$ might be improved to poly-logarithmic in this way); (ii) a tailored analysis for TV error that does not rely on Pinsker's inequality, like those in Li & Yan (2025), may also be extended to manifolds; (iii) a tighter version of our Minakshisundaram-Pleijel parametrix bound. We leave these improvements as future work.

## 4 PROOF OUTLINE

Throughout the proof, we assume that

$$h \leq \frac{1}{\mathrm{poly}(d, K, \delta^{-1})}, \tag{3}$$

since otherwise the bound in Theorem 1 would be trivial (recall that TV distance is always bounded by 2). We start by recalling the sequence considered in RSGM. Let $(Y_k)_{k \in \{0,\dots,N\}}$ be given by $Y_N \sim \mu$ and for any $k \in \{0, \dots, N-1\}$:

$$Y_{k-1} = \begin{cases} \exp_{Y_k}\left[h\widehat{s}_{t_k}(Y_k) + \sqrt{h}\,G_k\right], & \|h\widehat{s}_{t_k}(Y_k) + \sqrt{h}\,G_k\| \leq h^{1/4}, \\ \text{drawn from } \mu, & \text{otherwise.} \end{cases}$$

This defines a sequence of probability transition kernels $\widehat{\mathsf{K}}_{t_k, t_{k-1}}$. For simplicity, we denote this by $\widehat{\mathsf{K}}_k$. Let $q_k$ be the law of $Y_k$. We have

$$q_0 = q_N \widehat{\mathsf{K}}_N \widehat{\mathsf{K}}_{N-1} \cdots \widehat{\mathsf{K}}_1.$$

Similarly, the probability transition kernel from time $t_k$ to $t_{k-1}$ in (2) is denoted by $\mathsf{K}_{t_k, t_{k-1}}$ or $\mathsf{K}_k$ in short. We have

$$p_0 = p_N \mathsf{K}_N \mathsf{K}_{N-1} \cdots \mathsf{K}_1.$$

Our goal would be to bound $\mathsf{TV}(p_0, q_0)$ as in Theorem 1, by decomposing the total error into four components:

(initialization error) + (score error) + (drift discretization error) + (BM simulation error).

More concretely:

- **Initialization error** arises from initializing $Y_N$ with $\mu$ instead of the true marginal $p_N$;
- **Score error** arises from imperfect score estimation;
- **Drift discretization error** arises from approximating the continuous-time drift $\widehat{s}_t(Y_t)$ by its "time-frozen" counterpart $\widehat{s}_{t_k}(Y_{t_k})$;
- **Brownian motion (BM) simulation error** is a distinctive feature of the manifold setting. Unlike in Euclidean space — where the transition kernel of Brownian motion over $[t_k, t_{k-1}]$ is exactly Gaussian with variance $(t_k - t_{k-1})$ — the transition kernel of manifold-valued Brownian motion cannot be simulated exactly by any discrete-time process, even after time discretization. This inherent inexactness gives rise to this final error term.

The first two components are relatively easier to bound using well-established tools: mixing rate bounds of heat flow (Urakawa, 2006) and Girsanov transform (Chen et al., 2023). For the drift discretization error, recent techniques developed in the Euclidean setting (Benton et al., 2024) can also be adapted with modifications that account for the manifold curvature. However, the last component — the Brownian motion simulation error — represents the core challenge in the manifold setting, which fundamentally denies a direct extension of Euclidean analysis.

**Step I. Constructing auxiliary kernels via localization.** In view of this, we first introduce an intermediate random process that separates the drift discretization error from the BM simulation error. Constructing such a process, however, involves additional technicality. In particular, the frozen drift $\widehat{s}_{t_k}(Y_{t_k})$ is a vector in the tangent space $T_{Y_k}\mathcal{M}$, and is therefore only well-defined at the fixed point $Y_k$. This poses a compatibility issue: as Brownian motion evolves continuously on the manifold, it immediately departs from $Y_k$, rendering the frozen drift ill-defined. Careful geometric considerations are thus required to reconcile the piecewise-constant drift approximation with the intrinsic curvature of the manifold.

In our analysis, this is handled using localization by the construction of an auxiliary sequence of transition kernels $\mathsf{K}_k^{\mathsf{aux}}$. These kernels do not appear in the algorithm itself; they serve solely as an analytical tool to facilitate the proof. These kernels expose the behavior of the time-reverse SDE (2) when the estimated score $\widehat{s}_t$ is frozen to be a constant vector field in between discretization steps, meanwhile keeping the continuous Brownian motion.

Let $\eta : [0, \infty) \to [0, 1]$ be a smooth cutoff function, i.e., $\eta$ is decreasing, $\eta|_{[0,1]} \equiv 1$ and $\eta|_{[4,\infty)} \equiv 0$. Such a function can be chosen such that $|\eta'| + |\eta''| + |\eta'''| \leq 100$. Recall that the injective radius of $\mathcal{M}$ is lower bounded by $1/K$, and the curvature is upper bounded by $K$. Define

$$\omega := \frac{c_\omega}{Kd^4}, \qquad \eta_\omega(r) = \eta\left(\frac{4r^2}{\omega^2}\right), \quad r \geq 0, \tag{4}$$

where $c_\omega > 0$ is a small universal constant. We have $\eta_\omega|_{[0,\frac{\omega}{2}]} \equiv 1$ and $\eta_\omega|_{[\omega,\infty)} \equiv 0$. For $t > 0$, $x, y \in \mathcal{M}$, define the following vector field on $\mathcal{M}$:

$$\mathscr{S}_{t,x}(y) = (\mathrm{d}\exp_x)_{\log_x y} \left(\eta_\omega(\rho(x,y)) \cdot \widehat{s}_t(x)\right) \in T_y\mathcal{M}.$$

Intuitively speaking, $\mathscr{S}_{t,x}(\cdot)$ is the "constant" velocity field $\widehat{s}_t(x)$ in normal coordinates, which represents our idea of freezing the drift term for a time period. The $\mathrm{d}\exp_x$ in the formula is responsible for identifying $T_y\mathcal{M}$ with $T_x\mathcal{M}$. [2] On the other hand, the cut-off function $\eta_\omega$ is necessary to keep all our discussions restricted to the injective radius, so as to avoid pathologies of cut locus.

With this in mind, we are ready to define $\mathsf{K}_k^{\mathsf{aux}}$ as the transition kernel from time $t_k$ to $t_{k-1}$ of the reverse-time SDE

$$\mathrm{d}Y_\tau = \mathscr{S}_{t_k,Y_{t_k}}(Y_\tau)\mathrm{d}t + U_{Y_\tau} \circ \mathrm{d}W_t, \quad \tau = T - t, \quad \tau \in [t_{k-1}, t_k], \tag{5}$$

and in addition,

$$p_k^{\mathsf{aux}} = p_N \mathsf{K}_N^{\mathsf{aux}} \mathsf{K}_{N-1}^{\mathsf{aux}} \cdots \mathsf{K}_{k+1}^{\mathsf{aux}}, \quad k = N, N-1, \cdots, 0.$$

**Step II. Decomposing different sources of error.** We now decompose

$$\mathsf{TV}(p_0, q_0) \leq \mathsf{TV}(p_0, p_0^{\mathsf{aux}}) + \mathsf{TV}(p_0^{\mathsf{aux}}, q_0) \leq \sqrt{2\mathsf{KL}(p_0 \parallel p_0^{\mathsf{aux}})} + \mathsf{TV}(p_0^{\mathsf{aux}}, q_0),$$

where the last inequality used Pinsker's inequality. To control $\mathsf{KL}(p_0 \parallel p_0^{\mathsf{aux}})$, we further introduce the counterpart of $\mathscr{S}_{t,x}$ using the exact score function $\nabla \log p_t$:

$$\mathscr{S}_{t,x}^\star(y) = (\mathrm{d}\exp_x)_{\log_x y} \left(\eta_\omega(\rho(x,y)) \cdot \nabla \log p_t(x)\right) \in T_y\mathcal{M}.$$

We apply Girsanov's theorem (Hsu, 2002) to compare (5) with (2), in a way that is standard in recent literature (Chen et al., 2023; De Bortoli et al., 2022). Denote the path law of the solution of (2) by $\mathsf{Law}(Y)$, and the path law of the solution of (5) by $\mathsf{Law}(Y^{\mathsf{aux}})$. Girsanov's theorem asserts that the KL divergence $\mathsf{KL}(\mathsf{Law}(Y) \parallel \mathsf{Law}(Y^{\mathsf{aux}}))$ is upper bounded by the expectation of the squared norm of the difference between the drift terms in the two SDEs.[3] More concretely,

$$\mathsf{KL}\big(\mathsf{Law}(Y) \parallel \mathsf{Law}(Y^{\mathsf{aux}})\big) \leq \sum_{k=1}^N \int_{t_{k-1}}^{t_k} \mathbb{E}\left\|\nabla \log p_t(Y_t) - \mathscr{S}_{t_k,Y_{t_k}}(Y_t)\right\|^2 \mathrm{d}t.$$

---

[2] Generally speaking, it is more natural to use parallel transport to identify different tangent spaces. However, this would later lead to a more complicated treatment of the perturbed heat equation with variable drifts. We choose to use parallelism in normal coordinates instead for simplicity.

[3] In its classical form, Girsanov's theorem requires integrability such as Novikov's condition to hold. In our setting, this can be bypassed with a localization argument as in Chen et al. (2023).

Since $p_0$ and $p_0^{\mathsf{aux}}$ are marginals of $\mathsf{Law}(Y)$ and $\mathsf{Law}(Y^{\mathsf{aux}})$ respectively at time $t = t_0$, by post-processing inequality, we have

$$\mathsf{KL}(p_0 \parallel p_0^{\mathsf{aux}}) \leq \sum_{k=1}^{N} \int_{t_{k-1}}^{t_k} \mathbb{E} \left\| \nabla \log p_t(Y_t) - \mathscr{S}_{t_k, Y_{t_k}}(Y_t) \right\|^2 \mathrm{d}t$$

$$\leq 2 \underbrace{\sum_{k=1}^{N} \int_{t_{k-1}}^{t_k} \mathbb{E} \| \nabla \log p_t(Y_t) - \mathscr{S}_{t_k, Y_{t_k}}^{\star}(Y_t) \|^2 \mathrm{d}t}_{\text{drift discretization}} + 2 \underbrace{\sum_{k=1}^{N} \int_{t_{k-1}}^{t_k} \mathbb{E} \| \mathscr{S}_{t_k, Y_{t_k}}(Y_t) - \mathscr{S}_{t_k, Y_{t_k}}^{\star}(Y_t) \|^2 \mathrm{d}t}_{\text{score matching}} .$$

$$(6)$$

It remains to decompose $\mathsf{TV}(p_0^{\mathsf{aux}}, q_0)$. To isolate the initialization error, we introduce
$$q_0^{\star} = p_N \widehat{\mathsf{K}}_N \widehat{\mathsf{K}}_{N-1} \cdots \widehat{\mathsf{K}}_1.$$

By triangle inequality and post-processing inequality, we have
$$\mathsf{TV}(p_0^{\mathsf{aux}}, q_0) \leq \mathsf{TV}(p_0^{\mathsf{aux}}, q_0^{\star}) + \mathsf{TV}(q_0^{\star}, q_0) \leq \underbrace{\mathsf{TV}(p_0^{\mathsf{aux}}, q_0^{\star})}_{\text{BM simulation}} + \underbrace{\mathsf{TV}(p_N, q_N)}_{\text{initialization}} .$$

**Step III. Controlling initialization and score errors.** By our design, $q_N = \mu$, and $\mathsf{TV}(p_N, q_N) = \mathsf{TV}(p_N, \mu)$. This is known as the mixing rate of heat flow in total variation norm, and has well-established bounds, e.g., Urakawa (2006). The score-matching error, on the other hand, can be controlled with an analysis on the distortion on the Riemannian metric in normal coordinates. We compile the bounds into the following lemma.

**Lemma 1.** *There exists a universal constant $C > 0$, such that whenever $T \geq 1$, we have*

$$\mathsf{TV}(p_N, q_N) \leq \mathrm{e}^{C(K + d \log d)} \mathrm{e}^{-\frac{\lambda_1}{2}(T - \frac{1}{2})},$$

*and*

$$\sum_{k=1}^{N} \int_{t_{k-1}}^{t_k} \mathbb{E} \| \mathscr{S}_{t_k, Y_{t_k}}(Y_t) - \mathscr{S}_{t_k, Y_{t_k}}^{\star}(Y_t) \|^2 \mathrm{d}t \leq 2 \varepsilon_{\mathsf{score}}^2.$$

**Step IV. Controlling drift discretization error with Itô/Stratonovich calculus and Li-Yau estimates.** The drift discretization error defined in (6) has a similar form to the discretization error for the Euclidean setting (Benton et al., 2024), though additional complications arise due to non-constant $\mathscr{S}_{t_k, Y_{t_k}}^{\star}$. The idea is to study the time derivative of $\mathbb{E} \| \nabla \log p_t(Y_t) - \mathscr{S}_{t_k, Y_{t_k}}^{\star}(Y_t) \|^2$, which in view of $\partial_\tau \log p_t = -\frac{1}{2} \Delta_{\mathcal{M}} p_t$ (negative sign due to reverse time) involves space derivatives of $\log p_t$ up to third order. Fortunately, after applying Itô/Stratonovich calculus to simplify the expression, a key property in the proof of the Euclidean setting carries over: third-order derivatives of $\log p_t$ cancel out. The remaining first and second-order derivatives can be controlled by Li-Yau estimates on the log-gradient of the heat kernel. We obtain

**Lemma 2.** *Under the assumptions in Theorem 1, there is a universal constant $C > 0$ such that*

$$\sum_{k=1}^{N} \int_{t_{k-1}}^{t_k} \mathbb{E} \| \nabla \log p_t(Y_t) - \mathscr{S}_{t_k, Y_{t_k}}^{\star}(Y_t) \|^2 \mathrm{d}t \leq \frac{C d^6 K^8}{\delta^3} h^2 N.$$

**Step V. Controlling BM simulation error using parametrix estimates.** Our approach is inspired by the following consequence of post-processing inequality and Pinsker's inequality:

$$\mathsf{TV}(p_0^{\mathsf{aux}}, q_0^{\star}) \leq \sqrt{2 \mathsf{KL}(p_0^{\mathsf{aux}} \parallel q_0^{\star})} \leq \sqrt{2 \sum_{k=1}^{N} \mathsf{KL}(p_k^{\mathsf{aux}} \mathsf{K}_k^{\mathsf{aux}} \parallel p_k^{\mathsf{aux}} \widehat{\mathsf{K}}_k)}.$$

This leads us to compare the kernel $\mathsf{K}_k^{\mathsf{aux}}$ and $\widehat{\mathsf{K}}_k$. In normal coordinates, the Fokker-Planck equation shows that these two are the solutions of the heat equations with the Euclidean Laplacian and with the manifold Laplace-Beltrami operator. We utilize the Minakshisundaram-Pleijel parametrix theory (Berline et al., 2003) in geometric analysis for this comparison, and establish a quantitative bound in polynomially small radius and polynomially short time (cf. Lemma 20).

## 5 CONCLUSION

We developed a discrete-time theory for Riemannian diffusion models showing that a polynomial stepsize suffices for TV-accurate sampling under mild geometric conditions. In particular, our results show that choosing a stepsize polynomially small in manifold parameters achieves any prescribed TV target without exponential blow-ups in dimension or curvature. This complements prior Wasserstein-type guarantees which require exponentially many steps. Several important future directions remain open.

- *Sharper bounds.* For simplicity, we did not attempt to establish sharp bounds for the error terms in our analysis, and it is likely that the degree of the polynomial in the bound could be improved significantly by refining our analysis, and some of the polynomial dependencies can be improved to logarithmic ones (Benton et al., 2024; Li & Jiao, 2024).

- *Analysis of deterministic samplers.* We focused on DDPM-style stochastic samplers in our analysis. For practical purpose, it is also tempting to develop an analogous theory for DDIM-style deterministic samplers (Song et al., 2021; Li et al., 2024).

- *Conditional sampling.* Our theory was for unconditional diffusion models. Applications like solving inverse problems require conditional sampling, which calls for both new algorithm design and new theoretical analysis (Xu & Chi, 2024).

## 6 ACKNOWLEDGMENT

The work of X. Xu and Y. Chi is supported in part by Air Force Office of Scientific Research under FA9550-25-1-0060, and by National Science Foundation under ECCS-2126634/2537078. Z. Zhang, Y. Nakahira, and G. Qu are supported in part by NSF CAREER Award 2339112, NSF Award 2512805, NSF Award 2442948, and the Pennsylvania Infrastructure Technology Alliance. Any opinions, findings, and conclusions or recommendations expressed in this material are those of the author(s) and do not necessarily reflect the views of the United States Air Force.

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

## A    NUMERICAL EXPERIMENTS

In this section, we verify the results in Theorem 1 on compact manifolds by measuring the exit probability in the reverse steps of Algorithm 1, which is extensively used in the proof of Lemma 19 to ensure the convergence of Algorithm 1, and the TV distance between the target distribution (a Gaussian mixture) and the recovered distribution.

**Reset probability on $\mathbb{S}^2$ and $\mathbb{T}^2$** . We start by examining the total reset probability on the unit 2-sphere $\mathbb{S}^2$ and on the 2-torus $\mathbb{T}^2$. We run the backward process in Algorithm 1 with different stepsizes $h$ in each setting with $p_0$ being Gaussian mixture (see below for definition), and record the fraction of trials whose tangent update $\Delta_k = h\, s_{t_k}(Y_k) + \sqrt{h}\, G_k$ violates $\|\Delta_k\| \leq h^{1/4}$ at least once among all steps. On both manifolds, Figure 1a shows a clear linear trend of the logarithm of reset probability of against $h^{-1/2}$, which can be obtained by Gaussian tails. This confirms that the rejection sampling has no practical impact on the performance of Algorithm 1.

**High-dimensional torus.**    We extend this experiment to the $d$-dimensional flat torus $\mathbb{T}^d$ for different stepsizes. Figure 1b reports the logarithm of the reset probability versus $h^{-1/2}$ for $d \in \{2, 4, 8\}$. Increasing $d$ raises the baseline reset rate, yet the slope of the decay remains essentially unchanged—resets remain exponentially rare as $h \downarrow 0$, which aligns with our analysis.

**TV accuracy on $\mathbb{T}^d$ with warped Gaussian mixture.**    Finally, we assess the distributional accuracy in TV for a *warped Gaussian mixture* target on $\mathbb{T}^d$, $d \in \{1, 2, 3\}$. Here, the warped Gaussian distribution is defined to be the push-forward of the Gaussian distribution by the universal covering $\mathbb{R}^d \to \mathbb{T}^d$ given by $(x_1, \cdots, x_d) \mapsto (e^{i2\pi x_1}, \cdots, e^{i2\pi x_d})$, and warped Gaussian mixture is similarly the push-forward of Gaussian mixture in $\mathbb{R}^d$. The result is depicted in Figure 2, which confirms that the total variation decays fast with the increase of the number of steps.

## B    PRELIMINARIES

We first introduce some tools we use in the rest of the proof.

**Lemma 3** (Pinsker's inequality, Polyanskiy & Wu (2025))**.** *For any two probability distributions $p, q$ on $\mathcal{M}$, we have*

$$\mathsf{TV}(p, q) \leq \sqrt{2\mathsf{KL}(p \parallel q)}.$$

**Lemma 4.** *Let $v$ be a vector field on $\mathcal{M}$. In a local coordinate on $\mathcal{M}$, we have*

$$\partial_\alpha v_\beta = (\nabla_\alpha v)^\beta - \Gamma^\beta_{\alpha\gamma} v_\gamma.$$

*Here $\Gamma^\beta_{\alpha\gamma}$ is the Christoffel symbol, defined as*

$$\Gamma^\beta_{\alpha\gamma} = \frac{1}{2} g^{\beta\delta} (\partial_\alpha g_{\gamma\delta} + \partial_\gamma g_{\alpha\delta} - \partial_\delta g_{\alpha\gamma}).$$

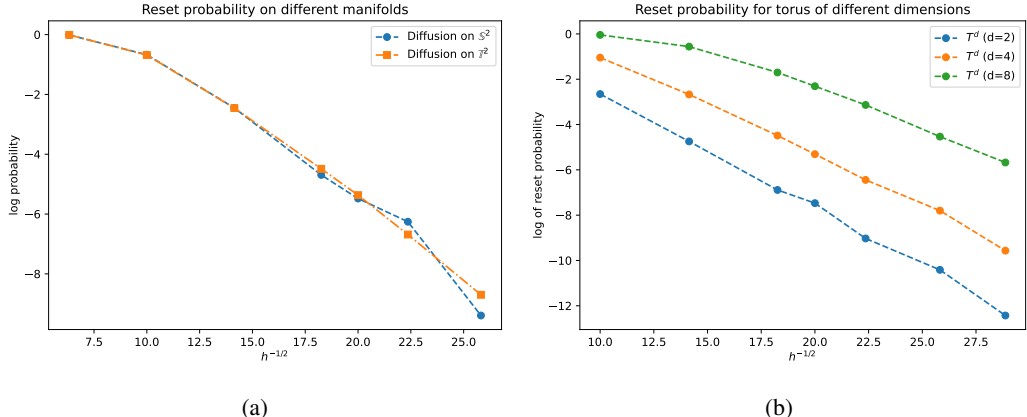

(a)  (b)

Figure 1: Reset probabilities on spheres and tori. In Figure 1a, we examine the relationship between $h^{-1/2}$ and the log of the reset probability of Algorithm 1 on both sphere $\mathbb{S}^2$ and torus $\mathbb{T}^2$ under the reset rules of Algorithm 1. In both cases, we see that the reset probability decays exponentially, confirming the conclusion in (43). In Figure 1b, we examine the same statistics on high-dimensional tori, and we find increasing $d$ only shifts the curves to the right but leaves the exponential decay rate in $h^{-1/2}$ unchanged.

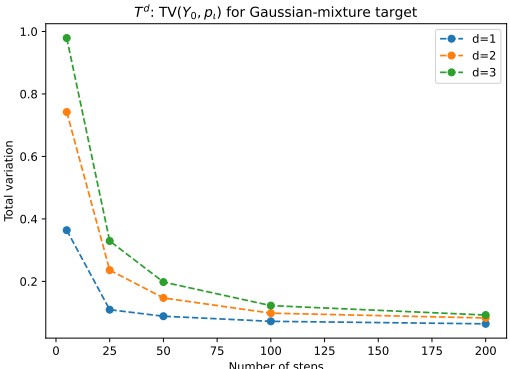

Figure 2: TV distance on $\mathbb{T}^d$ with a warped Gaussian-mixture target. The total variation is estimated with a kernel density estimator.

**Lemma 5** (Metric distortion in normal coordinates). *There exist coefficients $c, C > 0$ polynomial in $d$ and constant in other parameters, such that the following holds. Let $x \in \mathcal{M}$. In the normal coordinates $(\partial_\alpha)$ at $x$, for any $y \in \mathcal{M}$ such that $\rho(x, y) \le c/K$, we have*

$$\|g(y) - I\| \le CKd^2(x, y),$$
$$\|\partial_\alpha g_{\beta\gamma}\| \le CK\rho(x, y),$$
$$\|\partial_{\alpha\beta} g_{\gamma\xi}\| \le CK.$$

*Proof.* This is a quantitative version of the well-known Taylor expansion of $g$ in normal coordinates (cf. Berline et al. (2003, Proposition 1.28)):

$$g_{\alpha\beta}(\exp_x(u)) = \delta_{\alpha\beta} - \frac{1}{3} R_{\alpha\gamma\beta\xi}(x) u^\gamma u^\xi + O\big((\|\mathrm{Rm}\| + \|\nabla\mathrm{Rm}\|)\|u\|^3\big), \quad \|u\| \le c/K.$$

Let $(e_\alpha)_{\alpha=1}^d$ be an orthonormal basis of $T_x\mathcal{M}$; normal coordinates at $x$ are defined by identifying a point $\exp_x(z^\alpha e_\alpha)$ with its coordinate vector $(z^\alpha)$. Denote by $\partial_\alpha$ the coordinate vector fields.

**Step I: Representation by Jacobi fields.** Define the geodesic segment with *unit parameter* $s \in [0, 1]$:

$$\gamma(s) := \exp_x(sv), \qquad \gamma(0) = x, \quad \gamma(1) = y, \quad \dot{\gamma}(s) = \frac{d}{ds}\gamma(s).$$

Then $\nabla_s \dot{\gamma} = 0$ and $|\dot{\gamma}(s)| \equiv |v| = r$.

Fix an orthonormal basis $(e_\alpha)_{\alpha=1}^d$ of $T_x\mathcal{M}$ and parallel transport it along $\gamma$ to obtain an orthonormal frame $(E_\alpha(s))$ along $\gamma$:

$$\nabla_s E_\alpha(s) = 0, \qquad E_\alpha(0) = e_\alpha.$$

Let $J_\beta(s)$ be the Jacobi field along $\gamma$ corresponding to varying the initial point in direction $e_\beta$ in the normal coordinate chart, i.e.

$$J_\beta(s) := \mathrm{d}(\exp_x)_{sv}(se_\beta) \in T_{\gamma(s)}\mathcal{M}.$$

Equivalently, $J_\beta$ is the unique Jacobi field solving

$$\nabla_s^2 J_\beta + \mathrm{Rm}(J_\beta, \dot{\gamma})\dot{\gamma} = 0, \qquad J_\beta(0) = 0, \quad \nabla_s J_\beta(0) = e_\beta. \tag{7}$$

Note that in normal coordinates, $\partial_\beta|_x = e_\beta$ and the geodesic variation $\exp_x(s(v + \varepsilon e_\beta))$ yields (7).

Write $J_\beta(s)$ in the parallel frame:

$$J_\beta(s) = \sum_{\alpha=1}^d \mathsf{J}_{\alpha\beta}(s)\, E_\alpha(s),$$

and let $\mathsf{J}(s) \in \mathbb{R}^{d \times d}$ be the matrix with entries $\mathsf{J}_{\alpha\beta}(s)$. Since $E_\alpha$ is parallel, (7) becomes the *matrix Jacobi equation*

$$\mathsf{J}''(s) + \mathsf{R}(s)\, \mathsf{J}(s) = 0, \qquad \mathsf{J}(0) = 0, \quad \mathsf{J}'(0) = I, \tag{8}$$

where the curvature matrix $\mathsf{R}(s)$ is defined by

$$\big(\mathsf{R}(s)\, u\big)_\alpha := \Big\langle \mathrm{Rm}\Big( \sum_\mu u_\mu E_\mu(s), \dot{\gamma}(s)\Big)\dot{\gamma}(s), E_\alpha(s)\Big\rangle.$$

From $\|\mathrm{Rm}\| \le K$ and $|\dot{\gamma}| = r$ we have

$$\|\mathsf{R}(s)\| \le K\, |\dot{\gamma}(s)|^2 = Kr^2, \qquad s \in [0, 1]. \tag{9}$$

In normal coordinates, the coordinate vector fields at $y = \gamma(1)$ are

$$\partial_\beta|_y = \mathrm{d}(\exp_x)_v(e_\beta) = J_\beta(1).$$

Since the frame at $s = 1$ is orthonormal, the metric coefficients are

$$g_{\beta\gamma}(y) = \langle \partial_\beta|_y, \partial_\gamma|_y \rangle = \langle J_\beta(1), J_\gamma(1) \rangle = \sum_{\alpha=1}^d \mathsf{J}_{\alpha\beta}(1)\, \mathsf{J}_{\alpha\gamma}(1) = \big(\mathsf{J}(1)^\top \mathsf{J}(1)\big)_{\beta\gamma}.$$

Hence, as matrices,

$$g(y) = \mathsf{J}(1)^\top \mathsf{J}(1). \tag{10}$$

**Step II: Control of $\mathsf{J}(1) - I$ via Grönwall inequality.** From (8), integrating twice and using $\mathsf{J}(0) = 0$, $\mathsf{J}'(0) = I$, we get the exact Volterra equation

$$\mathsf{J}(s) = sI - \int_0^s (s - \tau)\, \mathsf{R}(\tau)\, \mathsf{J}(\tau)\, d\tau, \qquad s \in [0, 1]. \tag{11}$$

Taking operator norms and using (9) gives for $s \in [0, 1]$:

$$\|\mathsf{J}(s)\| \le s + Kr^2 \int_0^s (s - \tau)\, \|\mathsf{J}(\tau)\|\, d\tau.$$

A standard Grönwall argument yields

$$\|\mathsf{J}(s)\| \le Cs \quad \text{and} \quad \|\mathsf{J}'(s)\| \le C \qquad \text{for all } s \in [0, 1], \text{ provided } r \le c/\sqrt{K}. \tag{12}$$

Now subtract $sI$ in (11):

$$\mathsf{J}(s) - sI = -\int_0^s (s-\tau)\, \mathsf{R}(\tau)\, \mathsf{J}(\tau)\, d\tau.$$

Using (9) and (12),

$$\|\mathsf{J}(1) - I\| \le \int_0^1 (1-\tau)\, \|\mathsf{R}(\tau)\|\, \|\mathsf{J}(\tau)\|\, d\tau \le C\int_0^1 (1-\tau)\,(Kr^2)\,\tau\, d\tau \le CKr^2.$$

Combine with (10):

$$g(y) - I = \big(\mathsf{J}(1)^\top \mathsf{J}(1) - I\big) = \big(\mathsf{J}(1) - I\big)^\top + \big(\mathsf{J}(1) - I\big) + \big(\mathsf{J}(1) - I\big)^\top \big(\mathsf{J}(1) - I\big),$$

so

$$\|g(y) - I\| \le C\|\mathsf{J}(1) - I\| \le CKr^2. \tag{13}$$

**Step III: Control of first derivatives.** We first control $\partial_\alpha \mathsf{J}(1)$ as a function of the coordinate $v$. Let $v \mapsto \mathsf{J}_v(s)$ denote the Jacobi matrix for the geodesic $\gamma_v(s) = \exp_x(sv)$. Differentiate the ODE (8) w.r.t. $v^\alpha$:

$$\partial_\alpha \mathsf{J}'' + \mathsf{R}\,\partial_\alpha \mathsf{J} = -(\partial_\alpha \mathsf{R})\,\mathsf{J}, \qquad \partial_\alpha \mathsf{J}(0) = 0, \quad \partial_\alpha \mathsf{J}'(0) = 0. \tag{14}$$

*Bound on $\partial_\alpha \mathsf{R}$.* Recall $\mathsf{R}(s)$ represents the operator $u \mapsto \mathrm{Rm}(u, \dot\gamma)\dot\gamma$ in the parallel frame. Varying $v$ changes both $\gamma$ and $\dot\gamma$; the corresponding variation field $V_\alpha(s) := \partial_\alpha \gamma_v(s)$ along $\gamma$ is itself a Jacobi field with $V_\alpha(0) = 0$, $\nabla_s V_\alpha(0) = e_\alpha$, hence by the same estimate as (12)

$$\|V_\alpha(s)\| \le Cs, \qquad \|\nabla_s V_\alpha(s)\| \le C. \tag{15}$$

Using the product rule on $\mathrm{Rm}(\cdot, \dot\gamma)\dot\gamma$ and our Assumption 1, together with $|\dot\gamma| = r$ and (15), one obtains the uniform operator bound

$$\|\partial_\alpha \mathsf{R}(s)\| \le C\Big(\|\nabla \mathrm{Rm}\|\,\|V_\alpha(s)\|\,|\dot\gamma|^2 + \|\mathrm{Rm}\|\,|\dot\gamma|\,\|\partial_\alpha \dot\gamma(s)\|\Big) \le C\big(K \cdot s \cdot r^2 + K \cdot r \cdot 1\big) \le CKr, \tag{16}$$

for all $s \in [0,1]$ (since $s \le 1$). Here we used $\partial_\alpha \dot\gamma = \nabla_s V_\alpha$.

Now solve (14) by the same Duhamel principle: integrating twice with zero initial data gives

$$\partial_\alpha \mathsf{J}(s) = -\int_0^s (s-\tau)\Big(\mathsf{R}(\tau)\,\partial_\alpha \mathsf{J}(\tau) + (\partial_\alpha \mathsf{R})(\tau)\,\mathsf{J}(\tau)\Big)\, d\tau. \tag{17}$$

Using (9), (16), and (12), we obtain

$$\|\partial_\alpha \mathsf{J}(s)\| \le Kr^2 \int_0^s (s-\tau)\,\|\partial_\alpha \mathsf{J}(\tau)\|\, d\tau + CKr \int_0^s (s-\tau)\,\|\mathsf{J}(\tau)\|\, d\tau \le Kr^2 \int_0^s (s-\tau)\,\|\partial_\alpha \mathsf{J}(\tau)\|\, d\tau + CKr\, s^3.$$

Apply the same Grönwall comparison as before (now with a forcing term $CKr\, s^3$) to conclude, for $r \le c/\sqrt{K}$,

$$\|\partial_\alpha \mathsf{J}(1)\| \le CKr. \tag{18}$$

Finally differentiate $g = \mathsf{J}^\top \mathsf{J}$:

$$\partial_\alpha g = (\partial_\alpha \mathsf{J})^\top \mathsf{J} + \mathsf{J}^\top (\partial_\alpha \mathsf{J}),$$

so by (12) (at $s = 1$) and (18),

$$\|\partial_\alpha g(y)\| \le 2\|\partial_\alpha \mathsf{J}(1)\|\,\|\mathsf{J}(1)\| \le C(Kr) \cdot 1 \le CKr. \tag{19}$$

**Step IV: Control of second derivatives.** Differentiate (14) once more:

$$\partial_{\alpha\beta} \mathsf{J}'' + \mathsf{R}\,\partial_{\alpha\beta} \mathsf{J} = -(\partial_{\alpha\beta} \mathsf{R})\,\mathsf{J} - (\partial_\alpha \mathsf{R})\,\partial_\beta \mathsf{J} - (\partial_\beta \mathsf{R})\,\partial_\alpha \mathsf{J}, \tag{20}$$

with zero initial data at $s = 0$.

*Bound on $\partial_{\alpha\beta} \mathsf{R}$.* Under Assumption 1, $\partial_{\alpha\beta} \mathsf{R}$ can be bounded *uniformly by $CK$* on $[0,1]$ as follows: expanding the second parameter derivative of $\mathrm{Rm}(\cdot, \dot\gamma)\dot\gamma$ produces terms of the schematic form

$$(\nabla \mathrm{Rm})(V) \cdot \dot\gamma \cdot (\partial\dot\gamma), \qquad \mathrm{Rm}(\cdot, \partial\dot\gamma) \cdot (\partial\dot\gamma), \qquad (\nabla \mathrm{Rm})(\partial V) \cdot \dot\gamma \cdot \dot\gamma,$$

and also terms involving $\nabla_s(\partial V)$, all of which are controlled using $\|V\| \lesssim 1$, $\|\nabla_s V\| \lesssim 1$ and the fact that each appearance of $\dot{\gamma}$ contributes a factor $r$. Concretely, one shows (using (15) for both $V_\alpha, V_\beta$ and the same Jacobi estimates for their derivatives) that

$$\|\partial_{\alpha\beta}\mathsf{R}(s)\| \leq CK \qquad \text{for all } s \in [0,1]. \tag{21}$$

Now apply Duhamel's principle to (20) with zero initial data:

$$\partial_{\alpha\beta}\mathsf{J}(s) = -\int_0^s (s-\tau)\Big(\mathsf{R}\,\partial_{\alpha\beta}\mathsf{J} + (\partial_{\alpha\beta}\mathsf{R})\,\mathsf{J} + (\partial_\alpha\mathsf{R})\,\partial_\beta\mathsf{J} + (\partial_\beta\mathsf{R})\,\partial_\alpha\mathsf{J}\Big)(\tau)\,d\tau.$$

Take norms and use (9), (21), (12), (16), (18):

$$\|\partial_{\alpha\beta}\mathsf{J}(s)\| \leq Kr^2\int_0^s (s-\tau)\,\|\partial_{\alpha\beta}\mathsf{J}(\tau)\|\,d\tau + CK\int_0^s (s-\tau)\,\|\mathsf{J}(\tau)\|\,d\tau + C(Kr)(Kr)\int_0^s (s-\tau)\,d\tau.$$

Since $\|\mathsf{J}(\tau)\| \leq C\tau$, the second integral is bounded by $CKs^3$, and the third is bounded by $CK^2r^2s^2 \leq CKs^2$ provided $r \leq c/\sqrt{K}$. Thus for $s \leq 1$,

$$\|\partial_{\alpha\beta}\mathsf{J}(s)\| \leq Kr^2\int_0^s (s-\tau)\,\|\partial_{\alpha\beta}\mathsf{J}(\tau)\|\,d\tau + CK.$$

Applying Grönwall argument once more yields

$$\|\partial_{\alpha\beta}\mathsf{J}(1)\| \leq CK. \tag{22}$$

Finally differentiate $g = \mathsf{J}^\top \mathsf{J}$ twice:

$$\partial_{\alpha\beta}g = (\partial_{\alpha\beta}\mathsf{J})^\top \mathsf{J} + \mathsf{J}^\top(\partial_{\alpha\beta}\mathsf{J}) + (\partial_\alpha\mathsf{J})^\top(\partial_\beta\mathsf{J}) + (\partial_\beta\mathsf{J})^\top(\partial_\alpha\mathsf{J}).$$

Hence by (12), (18), (22) (and $K^2r^2 \leq CK$ for $r \leq c/\sqrt{K}$),

$$\|\partial_{\alpha\beta}g(y)\| \leq C\|\partial_{\alpha\beta}\mathsf{J}(1)\| \cdot \|\mathsf{J}(1)\| + C\|\partial_\alpha\mathsf{J}(1)\|\,\|\partial_\beta\mathsf{J}(1)\| \leq CK + C(Kr)^2 \leq CK. \tag{23}$$

This completes the proof. $\qquad\square$

**Lemma 6.** *Fix $x \in \mathcal{M}$. Define*

$$J(x,u) := \big|\det \mathrm{d}\exp_x(u)\big| = \sqrt{\det g_{ij}(\exp_x u)}.$$

*There exist universal constants $c, C > 0$, such that for $u \in T_x\mathcal{M}$ with $\|u\| \leq \frac{c}{Kd}$, we have the following bound on $J(x,u)$:*

$$\Big|J(x,u) - 1\Big| \leq CdK\|u\|^2. \tag{24}$$

*In particular, we have*

$$\frac{1}{2} \leq J(x,u) \leq 2, \quad \|u\| \leq \frac{c}{Kd}.$$

*Proof.* Work in normal coordinates at $x$ so that $\exp_x : B_{\mathrm{euc}}(0, 1/K) \subset T_x\mathcal{M} \to B_{\mathrm{geo}}(x, 1/K)$ is a diffeomorphism and $g_{ij}(0) = \delta_{ij}$, $\Gamma_{ij}^k(0) = 0$.

From Lemma 5, we know that $\|g(\exp_x u) - I\| \leq CK\|u\|^2$. In the region $\|u\| \leq \frac{c}{Kd}$, we have

$$\|g(\exp_x u) - I\| \leq \frac{c}{d}.$$

Therefore, by Taylor expansion of determinants, we know

$$\begin{aligned}
|\det g(\exp_x u) - 1| &= |\det(I + g(\exp_x u) - I) - 1| \\
&\leq C\,\mathrm{tr}(g(\exp_x u) - I) \\
&\leq Cd \cdot \|g(\exp_x u) - I\| \\
&\leq Cd \cdot CK\|u\|^2.
\end{aligned}$$

This concludes the proof by adjusting $C$ if necessary. $\qquad\square$

The metric distortion bound implies that geodesic is almost a straight line, in a sufficiently small normal neighborhood. The following quantitative bound shall be useful.

**Lemma 7** (Geodesics are almost straight in small balls). *There exist coefficients $c, C > 0$ polynomial in $d$ and constant in other parameters, such that the following holds. Fix any $x \in \mathcal{M}$ and let $0 < r \leq c/K$. Let $y, z \in B_x(r)$ and $\gamma$ be the unit-speed geodesic connecting $y$ to $z$. Write*

$$y(s) := \exp_x^{-1}(\gamma(s)) \in T_x \mathcal{M} \simeq \mathbb{R}^d$$

*for its representation in normal coordinates at $x$. Then:*

  *(i) (Almost constant velocity)*

$$\sup_{s \in [0, \ell]} \left| \dot{y}(s) - \dot{y}(0) \right| \leq CKr^2.$$

  *(ii) (Almost linear trajectory)*

$$\sup_{s \in [0, \ell]} \left| y(s) - y(0) - s\, \dot{y}(0) \right| \leq CKr^3.$$

*In words, in normal coordinates at $x$, any geodesic segment contained in $B_x(r)$ deviates from the Euclidean line segment connecting its endpoints by at most $O(Kr^3)$ in position and $O(Kr^2)$ in direction.*

*Proof.* Work in normal coordinates at $x$. By bounded geometry and the choice of $r$, the metric coefficients satisfy, in view of Lemma 5, that

$$\|g(y) - I\| \leq CK|y|^2, \qquad \|\partial g(y)\| \leq CK|y|, \qquad |y| \leq r,$$

which implies the Christoffel symbols obey

$$|\Gamma(y)| \leq CK|y| \leq CKr.$$

The coordinate representation $y(s)$ of the geodesic satisfies the geodesic equation

$$\ddot{y}^k(s) + \Gamma_{ij}^k(y(s))\, \dot{y}^i(s)\dot{y}^j(s) = 0.$$

Since $\gamma$ is unit–speed and $g(y)$ is uniformly equivalent to the Euclidean metric on $|y| \leq r$, we have $|\dot{y}(s)| \asymp 1$. Consequently,

$$|\ddot{y}(s)| \leq CKr \qquad \text{for all } s \in [0, \ell].$$

Integrating once gives

$$|\dot{y}(s) - \dot{y}(0)| \leq \int_0^s |\ddot{y}(u)|\, \mathrm{d}u \leq CKr\, s \leq CKr^2,$$

proving (i). Integrating again yields

$$|y(s) - y(0) - s\dot{y}(0)| \leq \int_0^s \int_0^u |\ddot{y}(w)|\, \mathrm{d}w\, \mathrm{d}u \leq CKr\, s^2 \leq CKr^3,$$

which proves (ii). $\qquad \square$

We spell out the constants in a few classical inequalities in geometric analysis.

**Lemma 8** (Schoen et al. (1994, Thm. 4.6)). *Let $(\mathcal{M}, g)$ be a complete Riemannian manifold with $\mathrm{Ric}(\mathcal{M}) \geq -K$ for some $K \geq 0$. Let $H(x, y, t)$ be the heat kernel, i.e., the fundamental solution of $(\Delta - \frac{\partial}{\partial t})u(x, t) = 0$. Then, for every $\delta_{\mathrm{Sch}} > 0$ and $\alpha > 1$,*

$$H(t, x, y) \leq C(\delta_{\mathrm{Sch}}, d, \alpha)\, V_x(\sqrt{t})^{-1/2} V_y(\sqrt{t})^{-1/2} \exp\left[ -\frac{r^2(x, y)}{(4 + \delta_{\mathrm{Sch}})t} + C_1\, \delta_{\mathrm{Sch}} Kt \right],$$

*where $V_x(R) = \mu(B_x(R))$, $C(\delta_{\mathrm{Sch}}, d, \alpha) = (1 + \delta_{\mathrm{Sch}})^{d\alpha} \exp\left(\frac{1 + \alpha}{\delta_{\mathrm{Sch}}}\right)$, and $C_1 = \frac{\alpha d}{\alpha - 1}$.*

To unleash the power of Lemma 8, we need the following lower bound on volume of geodesic balls assuming bounded geometry.

**Lemma 9** (Günther's comparison theorem). *Under Assumption 1, we have*

$$V_x(r) \geq \frac{(2\pi)^{d/2}}{\Gamma(d/2)} \int_0^r \left( \frac{\sin(t\sqrt{K})}{\sqrt{K}} \right)^{d-1} dt, \quad 0 \leq r \leq 1/K.$$

*Here $\Gamma$ is the Gamma function. In particular, when $r \leq c/K$ for some small universal constant $c > 0$, we have*

$$V_x(r) \geq \frac{\pi^{d/2}}{d\Gamma(d/2)} r^d \geq \frac{1}{d^{d/2}} r^d.$$

*Proof.* We observe that $\|\mathrm{Rm}\| \leq K$ implies that the sectional curvature is upper bounded by $K$, since by definition $\sec(u,v) = \mathrm{Rm}(u,v,u,v)$. The first inequality follows from the classical form of Günther's comparison theorem, see for example Gray (2003, Theorem 3.17). The second inequality follows from the elementary bound that $\sin(x) \geq \frac{1}{2}x$ for $x \in [0,c]$, where $c$ is a small universal constant, and the crude bound $\Gamma(x) \leq x^{x-1}$ for $x \geq 1$. □

A complementary lower bound to Lemma 8 is as follows.

**Lemma 10** (Li & Xu (2011, Thm. 1.5)). *Let $(\mathcal{M}, g)$ be complete, possibly with $\mathrm{Ric}(\mathcal{M}) \geq -K$. For the (Neumann) heat kernel $H(x,y,t)$ and all $x, y \in \mathcal{M}$, $t > 0$,*

$$H(t,x,y) \geq (4\pi t)^{-d/2} \frac{(2Kt)^{d/2}}{(e^{2Kt} - 2Kt - 1)^{d/4}} \exp\left[ -\frac{\rho(x,y)^2}{4t} \left( 1 + \frac{Kt \coth(Kt) - 1}{Kt} \right) \right],$$

$$H(t,x,y) \geq (4\pi t)^{-d/2} \exp\left[ -\frac{\rho(x,y)^2}{4t} \left( 1 + \tfrac{1}{3}Kt \right) - \frac{d}{4}Kt \right]. \tag{25}$$

The above bounds for heat kernel translates seamlessly to $p_t$, since $p_t$ is a convolution of $p_0$ with $H(t, \cdot, \cdot)$. We formalize this in the following lemma.

**Lemma 11.** *We have*

$$\inf_{x,y \in \mathcal{M}} H(t,x,y) \leq \inf_{x \in \mathcal{M}} p_t(x) \leq \sup_{x \in \mathcal{M}} p_t(x) \leq \sup_{x,y \in \mathcal{M}} H(t,x,y).$$

*Proof.* This follows from taking infimum and supremum respectively in the formula (Duhamel principle)

$$p_t(y) = \int_{\mathcal{M}} p_0(x) H(t,x,y) \mu(dx).$$

□

The following lemma compiles a few follow-ups of Li-Yau estimates (Hamilton, 1993; Han & Zhang, 2016) with constants made explicit.

**Lemma 12.** *Under Assumption 1, we have Han-Zhang's inequality*

$$\frac{\nabla^2 p_t}{p_t} \preceq C_{\mathsf{HZ}} \left( \frac{1}{t} + K \right) \left( 1 + \log \frac{\sup p_{t/2}}{p_t} \right).$$

*On the other hand, we also have Hamilton's Harnack inequality*

$$\nabla^2 \log p_t = \frac{\nabla^2 p_t}{p_t} - \frac{(\nabla p_t)(\nabla p_t)^\top}{p_t^2} \succeq -\frac{1}{2t}g - C_{\mathsf{Ham}} \left( 1 + \log \frac{\sup p_{t/2}}{p_t} \right) g$$

*and*

$$\|\nabla \log p_t\|^2 = \frac{\|\nabla p_t\|^2}{p_t^2} \leq C \left( \frac{1}{t} + K \right) \log \frac{\sup p_{t/2}}{p_t}.$$

*Here $C > 0$ is a universal constant, $C_{\mathsf{HZ}} = CdK$, $C_{\mathsf{Ham}} = CdK^2$.*

*Proof.* The last inequality follows from Hamilton (1993, Theorem 1.1). The proof of the rest two inequalities require tracing the proofs of Hamilton (1993); Han & Zhang (2016). The details would be too tedious to reproduce here, so we leave pointers to relevant proofs for interested readers.

To prove the second inequality, we trace the proof of Hamilton (1993, Theorem 4.3) to see that if $A > 0$ is such that

$$\frac{\Delta_{\mathcal{M}} p_t}{p_t} \leq \frac{A}{t} \left( d + \log \frac{\sup p_{t/2}}{p_t} \right)$$

and

$$\frac{\|\nabla p_t\|^2}{p_t^2} \leq \frac{A}{t} \left( d + \log \frac{\sup p_{t/2}}{p_t} \right),$$

then

$$\frac{\nabla^2 p_t}{p_t} - \frac{(\nabla p_t)(\nabla p_t)^\top}{p_t^2} \succeq -\frac{1}{2t} g - CA(\|\mathrm{Rm}\| + \|\nabla \mathrm{Rm}\|)g.$$

Here $C > 0$ is an absolute constant. By Assumption 1, this implies

$$C_{\mathsf{Ham}} \leq CKC_{\mathsf{HZ}}.$$

Tracing the proof the main theorem in Han & Zhang (2016, Page 9), we can see

$$C_{\mathsf{HZ}} \leq C_{\mathsf{LY}}(1 + K) = CdK,$$

where $C_{\mathsf{LY}}$ is the maximum of the coefficients before $t^{-1}$ and $K$ in Li & Yau (1986, Theorem 1.2). This was explicitly defined as $Cd$ therein, by setting $\alpha = 2$ there. This completes the proof. $\square$

**Lemma 13.** *For any three points $x, y, z \in \mathcal{M}$, we have*

$$\frac{\rho(x,z)^2}{1-t} + \frac{\rho(z,y)^2}{t} \geq \rho(x,y)^2, \qquad \forall t \in (0,1).$$

*The equality is attainable at some point $z_\star$ on the minimum-length geodesic from $x$ to $y$. Moreover, if $x, y, z$ are within $\iota \leq 1/\operatorname{poly}(d,K)$ distance to each other, then the function*

$$\psi(z) := \frac{\rho(x,z)^2}{1-t} + \frac{\rho(z,y)^2}{t} - \rho(x,y)^2$$

*is $\frac{1 - Cd^2 K^2 \iota}{t(1-t)}$-strongly convex in the normal coordinates at $x$ (or $y$), where $C > 0$ is a universal constant.*

*Proof.* The first inequality follows from Cauchy-Schwarz inequality and triangle inequality:

$$(1 - t + t) \left( \frac{\rho(x,z)^2}{1-t} + \frac{\rho(z,y)^2}{t} \right) \geq (\rho(x,z) + \rho(z,y))^2 \geq \rho(x,y)^2.$$

Let $\gamma : [0,1] \to \mathcal{M}$ be the constant-speed, minimum-length geodesic from $x$ to $y$. It is then straightforward to check that

$$\frac{\rho(x,z_\star)^2}{1-t} + \frac{\rho(z_\star,y)^2}{t} = \rho(x,y)^2, \quad z_\star := \gamma(\lambda_\star),$$

where $\lambda_\star \in (0,1)$ solves the quadratic equation

$$\frac{\lambda^2}{1-t} + \frac{(1-\lambda)^2}{t} = 1.$$

Proving the strong convexity requires Lemma 5 and Lemma 6, which implies that $\rho(x,z)^2$ and $\rho(z,y)^2$ are both $(1 - Cd^2 K^2 \iota)$-strongly convex in the normal coordinates. The desired conclusion then follows from

$$\frac{1}{1-t} + \frac{1}{t} = \frac{1}{t(1-t)}.$$

The proof is completed. $\square$

## C  Initialization error

We require the following result from Urakawa (2006, Proposition 2.6).

**Lemma 14.** *Denote $H(t, x, y)$ the heat kernel on $\mathcal{M}$. Assume*

$$A := \sup_{t \leq 1} \sup_{x \in \mathcal{M}} t^{d/2} H(t, x, x).$$

*Then for any probability distribution $p_0$, its evolution along heat flow $\partial_t p_t = \frac{1}{2}\Delta_\mathcal{M}$ satisfies*

$$\mathsf{TV}(p_t, \mu) \leq \sqrt{A}\, \mathrm{e}^{-\frac{1}{2\lambda_1}(t - \frac{1}{2})}, \quad t \geq 1.$$

We combine this with the Li-Yau upper bound (Lemma 8) to obtain

**Lemma 15** (First part of Lemma 1). *Under Assumption 1, there exists a universal constant $C > 0$ such that*

$$\mathsf{TV}(p_N, \mu) \leq \mathrm{e}^{C(K + d\log(Kd))} \mathrm{e}^{-\frac{1}{2\lambda_1}(T - \frac{1}{2})}.$$

*Proof.* Plug the bound in Lemma 9 into Lemma 8 and use the fact that $\sup_{x,y} H(t, x, y)$ is decreasing in $t$ (by convolution inequality), we obtain

$$A \leq (Cd/K)^d \mathrm{e}^{CK}$$
$$\leq \exp(C'K + C'd\log(Kd)),$$

for some universal constants $C, C' > 0$. Note that we absorbed $d\log K$ into $K + d\log d$. The desired claim follows. □

## D  Score matching error

We now prove the second inequality in Lemma 1.

**Lemma 16.** *Under the same assumptions as in Theorem 1, we have*

$$\sum_{k=1}^{N} \int_{t_{k-1}}^{t_k} \mathbb{E}\|\mathscr{S}_{t_k, Y_{t_k}}(Y_t) - \mathscr{S}^\star_{t_k, Y_{t_k}}(Y_t)\|^2 \mathrm{d}t \leq 2\varepsilon^2_{\mathsf{score}}.$$

*Proof.* This is relatively straightforward. Notice that in normal coordinates, by Lemma 5, we have

$$\|\mathscr{S}_{t_k, Y_{t_k}}(Y_t) - \mathscr{S}^\star_{t_k, Y_{t_k}}(Y_t)\|^2 = g_{\alpha\beta}(Y_t)(\widehat{s}_t - \nabla \log p_t)^\alpha (\widehat{s}_{t_k}(Y_{t_k}) - \nabla \log p_{t_k}(Y_{t_k}))^\beta$$
$$\leq \|g(Y_t)\| \cdot \|\widehat{s}_{t_k}(Y_{t_k}) - \nabla \log p_{t_k}(Y_{t_k})\|^2$$
$$\leq 2\|\widehat{s}_{t_k}(Y_{t_k}) - \nabla \log p_{t_k}(Y_{t_k})\|^2$$

for $\rho(Y_t, Y_{t_k}) \leq c/K$, and is 0 otherwise due to our cutoff $\eta_\omega$. Therefore

$$\mathbb{E}\|\mathscr{S}_{t_k, Y_{t_k}}(Y_t) - \mathscr{S}^\star_{t_k, Y_{t_k}}(Y_t)\|^2 \leq 2\mathbb{E}\|\widehat{s}_{t_k}(Y_{t_k}) - \nabla \log p_{t_k}(Y_{t_k})\|^2,$$

and consequently,

$$\sum_{k=1}^{N} \int_{t_{k-1}}^{t_k} \mathbb{E}\|\mathscr{S}_{t_k, Y_{t_k}}(Y_t) - \mathscr{S}^\star_{t_k, Y_{t_k}}(Y_t)\|^2 \mathrm{d}t$$
$$\leq 2\sum_{k=1}^{N}(t_k - t_{k-1})\mathbb{E}\|\widehat{s}_{t_k}(Y_{t_k}) - \nabla \log p_{t_k}(Y_{t_k})\|^2 = 2\varepsilon^2_{\mathsf{score}}.$$

This proves the claim, as desired. □

# E DISCRETIZATION ERROR

**Lemma 17.** *Under the same assumptions as in Theorem 1 and assuming (3) without loss of generality, there is a universal constant $C > 0$ such that for $t_k - h \leq t \leq t_k$, we have*

$$\mathbb{E}\|\nabla \log p_t(Y_t) - \mathscr{S}^\star_{t_k, Y_{t_k}}(Y_t)\|^2 \leq \frac{Cd^6 K^8}{t^3}(t_k - t).$$

*Proof.* For convenience, set the reverse time

$$\tau := t_k - t.$$

The main challenge is that Li-Yau estimates provide sharp uniform control up to second-order derivatives of $\log p_t$, but a naïve calculation of the difference $\nabla \log p_t(Y_t) - \mathscr{S}^\star_{t_k, Y_{t_k}}(Y_t)$ involves third-order derivatives. More precisely, a straightforward Taylor expansion will introduce a factor of $\partial_\tau \nabla \log p_t$, which, by reverse-time heat equation $\partial_\tau p_t = -\frac{1}{2}\Delta_{\mathcal{M}} p_t$, contains third-order derivatives of $p_t$. We bypass this difficulty by making use of Itô's calculus to show that third-order derivatives cancel out; this is inspired by Benton et al. (2024), where a similar strategy was employed to the Euclidean setting.

**Step I. Applying Itô/Stratonovish formula.** We first compute $\partial_\tau \nabla \log p_t$. Since the forward heat equation is $\partial_t p_t = \frac{1}{2}\Delta_{\mathcal{M}} p_t$, we have

$$\partial_\tau \nabla \log p_t = -\partial_t \nabla \log p_t = -\nabla\left(\frac{\partial_t p_t}{p_t}\right) = -\frac{1}{2}\nabla\left(\frac{\Delta_{\mathcal{M}} p_t}{p_t}\right).$$

Use the manifold quotient rule and the identity $\Delta_{\mathcal{M}} \log p_t = \frac{\Delta_{\mathcal{M}} p_t}{p_t} - \|\nabla \log p_t\|^2$ to rewrite

$$\frac{\Delta_{\mathcal{M}} p_t}{p_t} = \Delta_{\mathcal{M}} \log p_t + \|\nabla \log p_t\|^2.$$

Therefore, we have

$$\partial_\tau \nabla \log p_t = -\frac{1}{2}\nabla\left(\frac{\Delta_{\mathcal{M}} p_t}{p_t}\right) = -\frac{1}{2}\nabla \Delta_{\mathcal{M}} \log p_t - \frac{1}{2}\nabla\|\nabla \log p_t\|^2$$

$$= -\frac{1}{2}\nabla \Delta_{\mathcal{M}} \log p_t - \nabla^2 \log p_t \cdot \nabla \log p_t.$$

On the other hand, it is straightforward to calculate

$$\nabla^2 \log p_t = \frac{\nabla^2 p_t}{p_t} - \frac{(\nabla p_t)(\nabla p_t)^\top}{p_t^2}.$$

Now, from Itô's formula, we know

$$\mathrm{d}\nabla \log p_t(Y_t) = (\partial_\tau \nabla \log p_t)(Y_t)\mathrm{d}\tau + \nabla^2 \log p_t(Y_t)(\nabla \log p_t(Y_t)\mathrm{d}\tau + U_{Y_t} \circ \mathrm{d}W_t) + \frac{1}{2}\Delta_{\mathcal{M}}\nabla \log p_t(Y_t)\mathrm{d}\tau$$

$$= -\frac{1}{2}\nabla \Delta_{\mathcal{M}} \log p_t \mathrm{d}\tau - \nabla^2 \log p_t \cdot \nabla \log p_t \mathrm{d}\tau + \nabla^2 \log p_t \cdot \nabla \log p_t \mathrm{d}\tau$$

$$+ \nabla^2 \log p_t \cdot U_{Y_t} \circ \mathrm{d}W_t + \frac{1}{2}\Delta_{\mathcal{M}}\nabla \log p_t \mathrm{d}\tau$$

$$= \frac{1}{2}(\Delta_{\mathcal{M}}\nabla - \nabla \Delta_{\mathcal{M}}) \log p_t \mathrm{d}\tau + \nabla^2 \log p_t \cdot U_{Y_t} \circ \mathrm{d}W_t$$

$$= \frac{1}{2}\mathrm{Ric}^\sharp(\nabla \log p_t, \cdot)\mathrm{d}\tau + \nabla^2 \log p_t \cdot U_{Y_t} \circ \mathrm{d}W_t,$$

where the last line follows from Bochner's identity $(\Delta_{\mathcal{M}}\nabla - \nabla \Delta_{\mathcal{M}})f = \mathrm{Ric}^\sharp(\nabla f, \cdot)$, and $\mathrm{Ric}^\sharp$ denotes the $(1,1)$-tensor obtained by raising one index in Ricci curvature. Notice here the cancellation of third-order derivatives.

On the other hand,

$$\mathrm{d}\mathscr{S}^\star_{t_k, Y_{t_k}}(Y_t) = \nabla \mathscr{S}^\star_{t_k, Y_{t_k}}(Y_t) \cdot (\nabla \log p_t \mathrm{d}\tau + U_{Y_t} \circ \mathrm{d}W_t) + \frac{1}{2}\Delta_{\mathcal{M}}\mathscr{S}^\star_{t_k, Y_{t_k}}(Y_t)\mathrm{d}\tau.$$

In normal coordinates, $\mathscr{S}^\star_{t_k, Y_{t_k}}$ is a constant vector field inside $B(0, \omega/3)$, therefore we have (cf. Lemma 4):

$$\nabla_\alpha \mathscr{S}^\star_{t_k, Y_{t_k}}(Y_t)^\beta = \Gamma^\beta_{\alpha\gamma} \mathscr{S}^\star_{t_k, Y_{t_k}}(Y_t)^\gamma = \Gamma^\beta_{\alpha\gamma} \nabla^\gamma \log p_{t_k}(Y_{t_k}), \quad \rho(Y_t, Y_{t_k}) \le \omega/3,$$

and similarly, when $\rho(Y_t, Y_{t_k}) \le \omega/3$, we have

$$\Delta_\mathcal{M} \mathscr{S}^\star_{t_k, Y_{t_k}}(Y_t)^\alpha = -\mathrm{Ric}^\alpha_{\ \beta} \nabla^\beta \log p_{t_k}(Y_{t_k}) - g^{\beta\gamma} \left( \partial_\gamma \Gamma^\alpha_{\beta\xi} + \Gamma^\alpha_{\gamma\zeta} \Gamma^\zeta_{\beta\xi} + \Gamma^\alpha_{\zeta\xi} \Gamma^\zeta_{\beta\gamma} \right) \nabla^\xi \log p_{t_k}(Y_{t_k}).$$

**Step II. Bounding the coefficients.** Combine the above formulas with the estimates given in Lemma 5, we obtain for some universal constant $C > 0$:

$$\left\| \mathrm{Ric}^\sharp(\nabla \log p_t, \cdot) \right\| \le CK \|\nabla \log p_t(Y_t)\|,$$

$$\left\| \nabla \mathscr{S}^\star_{t_k, Y_{t_k}} \right\| \le CK \|\nabla \log p_{t_k}(Y_{t_k})\|, \qquad \rho(Y_t, Y_{t_k}) \le \omega/2,$$

$$\left\| \Delta_\mathcal{M} \mathscr{S}^\star_{t_k, Y_{t_k}} \right\| \le CK^2 \|\nabla \log p_{t_k}(Y_{t_k})\|, \qquad \rho(Y_t, Y_{t_k}) \le \omega/2.$$

Outside the geodesic ball $B_{Y_{t_k}}(\omega/2)$, the field $\mathscr{S}^\star_{t_k, Y_{t_k}}$ is non-zero only inside $B_{Y_{t_k}}(\omega)$. Between these two balls, we have to take into account the radial derivative of the cutoff function $\eta_\omega$, whose first order derivative is bounded by $C\omega^{-1}$ and second order derivative by $C\omega^{-2}$ by our construction of $\eta_\omega$ (recall that $|\eta'| + |\eta''| \le 100$). Apply Lemma 4 and Lemma 5 again, this time we bound

$$\left\| \nabla \mathscr{S}^\star_{t_k, Y_{t_k}} \right\| \le CK\omega^{-1} \|\nabla \log p_{t_k}(Y_{t_k})\|,$$

$$\left\| \Delta_\mathcal{M} \mathscr{S}^\star_{t_k, Y_{t_k}} \right\| \le CK^2 \omega^{-2} \|\nabla \log p_{t_k}(Y_{t_k})\|.$$

We now apply Itô's formula on manifold (i.e., take expectation and invoke the martingale property in (1)) and collect the above bounds to obtain (cf. Benton et al. (2024))

$$\left| \frac{\mathrm{d}}{\mathrm{d}\tau} \mathbb{E} \|\nabla \log p_t(Y_t) - \mathscr{S}^\star_{t_k, Y_{t_k}}(Y_t)\|^2 \right|$$

$$\le CK^2 \left( \mathbb{E} \|\nabla \log p_t(Y_t)\|^2 + \mathbb{E} \|\nabla \log p_{t_k}(Y_{t_k})\|^2 + \mathbb{E} \|\nabla^2 \log p_t(Y_t)\|^2 \right)$$

$$\quad + CK^2 \omega^{-2} \mathbb{E} \left[ (\|\nabla \log p_t(Y_t)\|^2 + \|\nabla \log p_{t_k}(Y_{t_k})\|^2 + \|\nabla^2 \log p_t(Y_t)\|^2) \mathbf{1}_{\rho(Y_t, Y_{t_k}) > \omega/3} \right]$$

$$\le CK^2 \left( \mathbb{E} \|\nabla \log p_t(Y_t)\|^2 + \mathbb{E} \|\nabla \log p_{t_k}(Y_{t_k})\|^2 + \mathbb{E} \|\nabla^2 \log p_t(Y_t)\|^2 \right)$$

$$\quad + CK^2 \omega^{-2} \sqrt{\mathbb{E} \|\nabla \log p_t(Y_t)\|^4 + \mathbb{E} \|\nabla \log p_{t_k}(Y_{t_k})\|^4 + \mathbb{E} \|\nabla^2 \log p_t(Y_t)\|^4} \sqrt{\mathbb{P}(\rho(Y_t, Y_{t_k}) > \omega/3)}$$

$$\le CK^2 d \left( \frac{1}{t} + dK^2 \right)^2 \sup_{t \le s \le t_k} \sqrt{\mathbb{E} \log^4 \frac{\sup p_{s/2}}{p_s(Y_s)}} \cdot \left( 1 + \omega^{-2} \sqrt{\mathbb{P}(\rho(Y_t, Y_{t_k}) > \omega/3)} \right). \qquad (26)$$

Here the last line follows from Lemma 12.

**Step III. Controlling expectations via Chebyshev and Li-Yau estimates.** To bound $\mathbb{E} \log^4 \frac{\sup p_{s/2}}{p_s(Y_s)}$, we note that

$$\mathbb{E} \left( \frac{1}{p_t(Y_t)} \right) = \int \frac{1}{p_t} p_t \mathrm{d}\mu = \int \mathrm{d}\mu = 1.$$

By Chebyshev's inequality, we have

$$\mathbb{P} \left( \frac{1}{p_t(Y_t)} \ge \lambda \right) \le \lambda^{-1}, \quad \lambda > 0,$$

and then

$$\mathbb{P} \left( \log^4 \frac{1}{p_t(Y_t)} \ge \lambda \right) \le e^{-\sqrt[4]{\lambda}}, \quad \lambda \ge 0.$$

Integrate with respect to $\lambda$, we see

$$\mathbb{E}\log^4\frac{1}{p_t(Y_t)} \leq C.$$

We then apply Li-Yau's estimate (Lemma 8) combined with Lemma 9, Lemma 11 to obtain $\sup\log p_{s/2} \leq \sup\log H(s/2,x,y) \lesssim d\log\frac{d}{s}+Ks$, where the first inequality follows from $p_{s/2}$ being the convolution of $p_0$ with $H(s/2,x,y)$. These together shows

$$\mathbb{E}\log^4\frac{\sup p_{s/2}}{p_s(Y_s)} \leq \left(Cd\log\frac{d}{s}+CKs\right)^4 \leq \frac{Cd^5K^4}{\delta}.$$

Here we used $\log\frac{d}{s}\leq C\left(\frac{d}{s}\right)^{1/4}$, and $s\geq t\geq\delta$.

**Step IV. Controlling exit probability via stopping time.** It remains to bound the probability $\mathbb{P}(\rho(Y_t,Y_{t_k})>\omega/3)$. This would follow from a stopping time argument. We claim that given $t_k-t\leq h$, we have

$$\mathbb{P}(\rho(Y_t,Y_{t_k})>\omega/3) \leq \exp\left(-\frac{c\omega^2}{t_k-t}\right) \leq \omega^4. \tag{27}$$

where the last inequality follows from (3). Plug this back into the desired conclusion of the lemma is proved.

We now prove (27). Let $\sigma$ be the largest $t\leq t_k$ such that $\rho(Y_t,Y_{t_k})>\omega/3$. We have

$$\mathbb{P}(\rho(Y_t,Y_{t_k})>\omega/3)\leq\mathbb{P}(\sigma\geq t).$$

In the interval $[\sigma,t_k]$, $Y_t$ stays in the geodesic ball $B_{Y_{t_k}}(\omega/3)$, and follows the SDE (2). In normal coordinates, this can be spelled out explicitly:

$$dY_t^\alpha = \nabla^\alpha\log p_t(Y_t)dt + A_\beta^\alpha(Y_t)\circ dW_t^\beta$$
$$= \left(\nabla^\alpha\log p_t(Y_t)+\frac{1}{2}(\partial_\gamma A_\beta^\alpha)A^{\beta\gamma}\right)dt + A_\beta^\alpha dW_t^\beta, \tag{28}$$

where $A$ is the square root of the matrix representing the coefficients of the Laplace-Beltrami operator

$$\frac{1}{\sqrt{\det g}}\partial_\alpha(\sqrt{\det g}\cdot g^{\alpha\beta}\partial_\beta).$$

It can be checked with the help of Lemma 5 that $\|A-I\|\leq CdK\omega^2$, and $\|\partial_\alpha A\|\leq Cd^2K\omega$; we omit the computation that has a similar pattern as many of the previous arguments. Furthermore, we have $\|\nabla\log p_t(Y_t)\|\lesssim(\delta^{-1}+K)\log\frac{\sup p_{t/2}}{p_t}$ by the same argument via Lemma 12 as before. This time we combine the uniform bound provided by Lemma 10 with Lemma 8 to conclude $\log\frac{\sup p_{t/2}}{p_t}\lesssim(\delta^{-1}+K+d\log d)^2\operatorname{Diam}(\mathcal{M})^2\lesssim(\delta^{-1}+K+d\log d)^2K^2$ by Assumption 1. This shows

$$\sup\|\nabla\log p_t\|\lesssim(\delta^{-1}+K+d\log d)^3K^2. \tag{29}$$

Therefore, in view of Lemma 5 to convert the above bound to normal coordinates, and together with the aforementioned bound for $A$ and $\partial_\alpha A$, we see that the drift term up to time $\sigma$ will not exceed

$$\sup\left\|\nabla^\alpha\log p_t(Y_t)+\frac{1}{2}(\partial_\gamma A_\beta^\alpha)A^{\beta\gamma}\right\|\cdot(t_k-\sigma)\leq C(\delta^{-1}+K+d\log d)^3K^2(t_k-\sigma)\leq\frac{\omega}{12}, \tag{30}$$

where the last inequality used (3). On the other hand, the bound on $A$ implies that the quadratic variation of the martingale part does not exceed

$$\int_\sigma^{t_k}A_\gamma^\alpha A_\beta^\gamma dt\preceq 2(t_k-\sigma)I.$$

By Burkholder-Davis-Gundy inequality (Revuz & Yor, 2013), the tail of $\int_\sigma^{t_k}A_\beta^\alpha dW_t^\beta$ is $O(1)$-subgaussian, thus we have

$$\mathbb{P}\left(\sigma\geq t,\left\|\int_\sigma^{t_k}A_\beta^\alpha dW_t^\beta\right\|>\frac{\omega}{12}\right)\leq\exp\left(\frac{-c(\omega-2\sqrt{\rho(t_k-t)})^2}{t_k-t}\right).$$

In view of (3), combine this with (30) and (28), we have proved (27) as claimed. $\qquad\square$

**Lemma 18.** *Under the same assumptions as in Theorem 1 and assuming (3) without loss of generality, the discretization error obeys the following upper bound:*

$$\sum_{k=1}^{N} \int_{t_{k-1}}^{t_k} \mathbb{E} \|\nabla \log p_t(Y_t) - \mathscr{S}_{t_k, Y_{t_k}}^\star(Y_t)\|^2 \mathrm{d}t \le \frac{Cd^6 K^8}{\delta^3} h^2 N,$$

*where $C > 0$ is a universal constant.*

*Proof.* This follows directly from Lemma 17. $\qquad\square$

## F  BROWNIAN MOTION SIMULATION ERROR

In this section, we handle the Brownian motion simulation error using the machinery of Minakshisundaram-Pleijel parametrix. A complete introduction to this heavy machinery would require establish a whole system of notations and lemmas in geometric analysis, which is unduly burdensome. We instead refer the interested reader to Berline et al. (2003) for a comprehensive treatment, and point to results there whenever needed.

### F.1  OVERVIEW

Our aim is to prove the following lemma.

**Lemma 19.** *Under the same assumptions as in Theorem 1, and assuming (3) without loss of generality, we have*

$$\mathsf{TV}(p_0^{\mathsf{aux}}, q_0^\star) \le \sqrt{hT}\,\mathrm{poly}(d, K, \delta^{-1}).$$

To better explain the idea of the proof, we ignore the rejection sampling procedure in the construction of $\widehat{\mathsf{K}}_k$ temporarily. Our starting point is the observation that by Fokker-Planck equation, $\widehat{\mathsf{K}}_k$ is the heat kernel associated to the Euclidean Laplacian with drift $\mathscr{S}_{t_k, Y_{t_k}}$, in normal coordinates. On the other hand, $\mathsf{K}_k^{\mathsf{aux}}$ is also a heat kernel with the same drift, but associated to the manifold Laplace-Beltrami operator. The following lemma shows that the two solutions coincide up to first order in time, at least in a polynomially small neighborhood of initial point and in a polynomially short time.

**Lemma 20.** *Let $F^{\mathcal{H}}(t, x, y)$ be the (generalized) heat kernel for the operator $\mathcal{H} = \frac{1}{2}\Delta_{\mathcal{M}} + \langle \mathscr{S}_{t_k, Y_{t_k}}, \nabla \rangle$. Define the Euclidean density*

$$\varphi_t(u; x) := \frac{1}{(2\pi t)^{d/2}} \exp\left( -\frac{\|u - \mathscr{S}_{t_k, Y_{t_k}}(x)t\|^2}{2t} \right), \quad u \in T_x \mathcal{M},$$

*and let $\Phi(t, x, y)$ be the density of the push-forward by $\exp_x$ of $\eta_\omega \varphi_t(\cdot; x)$ with respect to the volume measure, where $\eta_\omega$ is the cutoff function defined in (4). Then there exists polynomial $\mathrm{poly}(d, K)$ with universally constant coefficients, such that for all $0 < t \le \frac{1}{\mathrm{poly}(d, K, \delta^{-1})}$ and for all $\rho(y, Y_{t_k}) \le t^{5/12}$, we have*

$$\left| \frac{F^{\mathcal{H}}(t, Y_{t_k}, y)}{\Phi(t, Y_{t_k}, y)} - 1 \right| \le \mathrm{poly}(d, K, \delta^{-1})t.$$

With Lemma 20 in hand, it is tempting to calculate the KL error with the following heuristic:

$$\begin{aligned}
\mathsf{KL}(p_k^{\mathsf{aux}} \mathsf{K}_k^{\mathsf{aux}} \,\|\, p_k^{\mathsf{aux}} \widehat{\mathsf{K}}_k) &\lesssim -\mathbb{E} \int F^{\mathcal{H}}(h, Y_{t_k}, \cdot) \log \frac{\Phi(h, Y_{t_k}, \cdot)}{F^{\mathcal{H}}(h, Y_{t_k}, \cdot)} \\
&\lesssim \mathbb{E} \int F^{\mathcal{H}}(h, Y_{t_k}, \cdot) \left( \frac{F^{\mathcal{H}}(h, Y_{t_k}, \cdot)}{\Phi(h, Y_{t_k}, \cdot)} - 1 \right)^2 \\
&\lesssim \mathrm{poly}(d, K, \delta^{-1})h^2,
\end{aligned}$$

where we ignore the fact that Lemma 20 holds only in a small neighborhood; the first line is post-processing inequality, and the second line stems from the fact that for two distributions $p, q$, we have

$$\int p \log \frac{q}{p} = \int p \log \left( 1 + \frac{q - p}{p} \right)$$

$$\geq \int p \left( \frac{q-p}{p} - C \frac{(q-p)^2}{p^2} \right)$$

$$= -C \int p \left( \frac{q}{p} - 1 \right)^2,$$

given $\frac{q}{p} - 1$ is sufficiently small, where the last line follows from $\int p = \int q = 1$. From this, we conclude that the accumulated error along $N$ steps is bounded by $\mathrm{poly}(d, K)h^2 N = \mathrm{poly}(d, K)hT$, and the desired bound follows from Pinsker's inequality.

Apart from Lemma 20, the above computation is the essence of this proof. The rest of this section is mainly devoted to proving Lemma 20, and then formalizing the above computation by handling exceptional events of exiting the polynomially small neighborhood.

### F.2 PROOF OF LEMMA 20: A PARAMETRIX ESTIMATE

We begin the proof of Lemma 20. For simplicity, denote by $v^{\alpha}$ the normal coordinate representation of $\widehat{s}_{t_k}(Y_{t_k})$. Naturally, our initial test solution is the drifted heat kernel, as simulated by our discretized process:

$$\varphi_t(u) := \frac{1}{(2\pi t)^{d/2}} \exp \left( -\frac{\|u - vt\|^2}{2t} \right), \quad u \in \mathbb{R}^d.$$

Before we compare this with the manifold heat kernel, there is one subtlety we need to keep in mind. The density $\varphi_t$ is with respect to the *Lebesgue* measure on $T_x \mathcal{M}$, not with respect to the *volume* on $\mathcal{M}$. We compute and define the corresponding density on $\mathcal{M}$ as follows:

$$\Phi(t, x, y) = \varphi_t(\log_x y)\sqrt{\Delta(x, y)}, \quad \Delta(x, y) := \frac{|\det \mathrm{d} \log_x y|^2}{\det g(y)}, \quad y \in B_x(\omega).$$

Here all quantities are computed in normal coordinates. The factor $\Delta(x, y)$ is known as the van Vleck-Morette determinant. From Lemma 5 and Lemma 6, we know that

$$\frac{1}{2} \leq \Delta(x, y) \leq 2, \quad \text{if } \rho(x, y) \leq \frac{c}{Kd}. \tag{31}$$

We consider the generalized Laplacian

$$\mathcal{H} := \frac{1}{2}\Delta_{\mathcal{M}} + \langle \mathscr{S}_{t_k, Y_{t_k}}, \nabla \rangle.$$

As in Lemma 20, denote by $F^{\mathcal{H}}$ the heat kernel of $\mathcal{H}$ at time $t_k - t_{k-1}$. We also propose an approximation of $F^{\mathcal{H}}$ by

$$\Psi(t, x, y) := G(t, x, y) \exp(\psi(x, y))\sqrt{\Delta(x, y)}, \quad G(t, x, y) := \frac{1}{(2\pi t)^{d/2}} \exp \left( -\frac{d^2(x, y)}{2t} \right),$$

where for any two point $x, y \in \mathcal{M}$, letting $\gamma : [0, 1] \to$ be a constant-speed geodesic connecting $x$ to $y$, we define

$$\psi(x, y) := \int_0^1 \left\langle \mathscr{S}_{t_k, Y_{t_k}}(\gamma(s)), \frac{\mathrm{d}}{\mathrm{d}s}\gamma(s) \right\rangle_g \mathrm{d}s.$$

The auxiliary function $\Psi$ bridges $F^{\mathcal{H}}$ and $\Phi$ in the following sense. On the one hand, we relate $\Phi$ and $\Psi$ with the following lemma:

**Lemma 21.** *There exists a polynomial* $\mathrm{poly}(d, K)$ *with universally constant coefficients such that the following holds. For any* $0 < r \leq \frac{1}{\mathrm{poly}(d, K, \delta^{-1})}$, $0 < t \leq \frac{1}{\mathrm{poly}(d, K, \delta^{-1})}$ *and for all* $x, y \in B_{Y_{t_k}}(r)$, *we have*

$$\left| \frac{\Phi(t, x, y)}{\Psi(t, x, y)} - 1 \right| \leq \mathrm{poly}(d, K, \delta^{-1})(r^3 + t).$$

On the other hand, we have the following asymptotic expansion:

$$F^{\mathcal{H}}(t, x, y) = \Psi(t, x, y) \cdot \left( 1 + \sum_{i=1}^{\infty} t^i u_i(x, y) \right), \quad t \to 0^+,$$

where $u_i$ are smooth functions that can be computed explicitly via a recursive formula (Berline et al., 2003). We will not need the formula here, but instead require $u_1$ and the remainder terms to be bounded properly. Such bounds have been well-established, which we wrap up into the following lemma. Recall the cutoff function $\eta_\omega$ with radius $\omega$ defined in (4). It is clear we can replace $\omega$ with any $\iota > 0$ to define a cutoff $\eta_\iota$ of radius $\iota$.

**Lemma 22** (adapted from Berline et al. (2003))**.** *Fix a positive $\iota \leq 1/\operatorname{poly}(d, K)$. There exists a smooth function $u_1(x, y)$ on $\mathcal{M} \times \mathcal{M}$ such that*

$$\|u_1\|_\infty + \|\nabla_y u_1\|_\infty \leq \operatorname{poly}(d, K, \delta^{-1}),$$

*and for all $0 < t \leq 1/\operatorname{poly}(d, K)$, $y \in B_x(\iota)$, we have*

$$\left| (\partial_t - \mathcal{H}) \left[ \eta_\iota(\rho(x, y)) \Psi(t, x, y)(1 + t u_1(x, y)) \right] \right| \leq r_\eta(t, x, y) + r_\psi(t, x, y), \tag{32}$$

*where*

$$r_\eta(t, x, y) \leq \frac{1}{\iota t} \operatorname{poly}(d, K) \mathbb{1}_{\frac{\iota}{2} \leq \rho(x, y) \leq \iota} G(t, x, y), \tag{33a}$$

$$r_\psi(t, x, y) \leq t \cdot \operatorname{poly}(d, K, \delta^{-1}) \mathbb{1}_{\rho(x, y) \leq \iota} G(t, x, y). \tag{33b}$$

*Proof.* The inequality on $u_1$ follows from Theorem 2.26 in Berline et al. (2003), with $\mathcal{H}$ the same as our $\mathcal{H}$ and therefore $F = \langle \mathscr{S}_{t_k, Y_{t_k}}, \nabla \rangle$. Note that all the coefficients in $\mathcal{H}$ are bounded in $C^2$ by $\operatorname{poly}(d, K)(1 + \|\mathscr{S}_{t_k, Y_{t_k}}\|_{C^2(\mathcal{M})})$, which is further bounded by $\operatorname{poly}(d, K, \delta^{-1})$ as we will show momentarily. In fact, by Lemma 5 and Assumption 1, (A3), we have

$$\|\mathscr{S}_{t_k, Y_{t_k}}\|_{C^2(\mathcal{M})} = \|\mathscr{S}_{t_k, Y_{t_k}}\|_\infty + \|\nabla \mathscr{S}_{t_k, Y_{t_k}}\|_\infty + \|\nabla^2 \mathscr{S}_{t_k, Y_{t_k}}\|_\infty \leq \operatorname{poly}(d, K)\, \omega^{-2}(1 + \|\nabla \log p_{t_k}(Y_{t_k})\|),$$

and then we control $\|\nabla \log p_{t_k}(Y_{t_k})\| \leq \operatorname{poly}(d, K, \delta^{-1})$ via (29), yielding the claimed bound (recall that $\omega^{-1} = \operatorname{poly}(d, K)$ by definition).

We proceed to prove (32). We follow the proof of Theorem 2.29, item (iii) in Berline et al. (2003), and choose the cutoff function $\psi$ there to be $\eta_\iota$ as defined in (4), and with the differential operator $B$ defined in Berline et al. (2003), we have

$$\left| (\partial_t - \mathcal{H}) \left[ \Psi(t, x, y)(1 + t u_1(x, y)) \right] \right| \leq \underbrace{\frac{1}{\iota t} \operatorname{poly}(d)\, G(t, x, y)}_{=: r_\eta,\ \nabla \eta_\iota \text{ related terms}} + \underbrace{t \cdot G(t, x, y) \cdot |(B_y u_2)(x, y)|}_{=: r_\psi}.$$

Here $B_y$ can be viewed as a coordinate-transformed version of $\mathcal{H}$, applied to the variable $y$ (precise definition can be found in the reference), and $u_2$ is the second order term in the expansion. Similar to the argument we used to bound $\|u_1\|$, in virtue of Theorem 2.26 in Berline et al. (2003), we have

$$\|\mathscr{B}_x u_2\|_{L^\infty(\mu)} \leq \operatorname{poly}(d, K, \delta^{-1}).$$

The claimed bound follows from combining the above inequalities. $\square$

We now state the Volterra series representation of heat kernel.

**Lemma 23** (Volterra series, Theorem 2.23 in Berline et al. (2003))**.** *Fix a $\iota > 0$. Let*

$$\Psi_1(t, x, y) := \eta_\iota(\rho(x, y)) \Psi(t, x, y)(1 + t u_1(x, y)),$$
$$r_1(t, x, y) := (\partial_t - \mathcal{H})\, \Psi_1(t, x, y).$$

*Define the time-space convolution operator $*$ as*

$$(f * g)(t, x, y) = \int_0^t \int f(t - s, x, z) g(s, z, y) \mu(\mathrm{d}z) \mathrm{d}s.$$

*Then we have*

$$F^{\mathcal{H}} = \Psi_1 + \sum_{k=1}^{\infty} (-1)^k\, \Psi_1 * r_1^{*k}, \quad \text{where} \quad r_1^{*k} := \underbrace{r_1 * \cdots * r_1}_{k \text{ times}},$$

*on any domain such that the series on the right hand side converges absolutely uniformly.*

**Lemma 24** (Iterative bounds for Volterra series). *There exists a polynomial* $\mathrm{poly}(d, K, \delta^{-1})$ *with universally constant coefficients such that the following holds. Assume* $0 < t \leq 1/\mathrm{poly}(d, K.\delta^{-1})$, *take* $\iota = 4t^{5/12}d$ *in the definition of* $\Psi$. *Then we have, for all* $\rho(x, y) \leq t^{5/12} = \iota/(4d)$, *that*

$$\sum_{k=1}^{\infty} \left| \Psi_1 * r_1^{*k} \right| (t, x, y) \leq t \cdot \mathrm{poly}(d, K, \delta^{-1}) \, G(t, x, y).$$

*Proof.* Recall Lemma 22, and denote by $P$ the polynomial factor $\mathrm{poly}(d, K)$ therein. Denote by $\lambda\Delta^k$ the dilated standard simplex

$$\lambda\Delta^k = \{(s_1, \cdots, s_{k+1}) : s_i \geq 0, \sum_{i=1}^{k+1} s_i = \lambda\}, \quad \lambda > 0.$$

Fix some $k \geq 1$. Set $z_0 = x$ and $z_k = y$, we have

$$\left| \Psi_1 * r_1^{*k} \right| (t, x, y)$$

$$\leq t^k P^k \int_{t\Delta^{k-1}} \mathrm{d}s \int_{\mathcal{M}^{k-1}} \left( \prod_{i=1}^{k} G(s_i, z_{i-1}, z_i) \Big( \mathbb{1}_{\rho(z_{i-1}, z_i) \leq \iota} + \frac{1}{\iota s_i} \mathbb{1}_{\rho(z_{i-1}, z_i) > \iota/2} \Big) \right) \mu^{\otimes(k-1)}(\mathrm{d}z).$$
(34)

We split the integral in (34) into a *local* part and an *outlier* part. Define a small "local" region

$$\mathcal{R} := \big\{ (z_1, \cdots, z_{k-1}) : \rho(z_i, x) \leq 2\iota, \; \rho(z_{i-1}, z_i) \leq \iota/2, \; i = 1, \cdots, k \big\}.$$

We further define

$$I_{\mathsf{loc}}(s) := \int_{\mathcal{R}} \left( \prod_{i=1}^{k} G(s_i, z_{i-1}, z_i) \right) \mu^{\otimes(k-1)}(\mathrm{d}z),$$

$$I_{\mathsf{out}}(s) := \int_{\mathcal{R}^c} \left( \prod_{i=1}^{k} G(s_i, z_{i-1}, z_i) \mathbb{1}_{\rho(z_{i-1}, z_i) \leq \iota} \Big( 1 + \frac{1}{\iota s_i} \mathbb{1}_{\rho(z_{i-1}, z_i) > \iota/2} \Big) \right) \mu^{\otimes(k-1)}(\mathrm{d}z).$$

It is clear that

$$\left| \Psi_1 * r_1^{*k} \right| (t, x, y) \leq t^k P^k \int_{t\Delta^{k-1}} (I_{\mathsf{loc}}(s) + I_{\mathsf{out}}(s)) \, \mathrm{d}s.$$
(35)

We will establish bounds for $I_{\mathsf{loc}}$ and $I_{\mathsf{out}}$ respectively.

**Bounding the local integral.** For ease of understanding, we begin by computing the first integral in $I_{\mathsf{loc}}$ with respect to $z_1$. Extracting the factors containing $z_1$, we need to calculate

$$\int_{\{\rho(z_1, x) \leq 2\iota\}} \frac{1}{(2\pi s_1)^{d/2}} \frac{1}{(2\pi s_2)^{d/2}} \exp\left( -\frac{\rho(x, z_1)^2}{2s_1} - \frac{\rho(z_1, z_2)^2}{2s_2} \right) \mu(\mathrm{d}z_1).$$
(36)

To proceed, we will invoke Lemma 13. Let $z_\star$ be a minimizer of

$$V(z) = \frac{\rho(x, z)^2}{s_1(s_1 + s_2)^{-1}} + \frac{\rho(z, z_2)^2}{s_2(s_1 + s_2)^{-1}} - \rho(x, z_2)^2.$$

By Lemma 13, $V(z_\star) = 0$. Since $V(z) > 0$ whenever $\rho(x, z) \geq \rho(x, z_2)$, we know that $z_\star \in B_x(2\iota)$. Moreover, Lemma 13 and Lemma 5 together imply

$$V(z) \geq (1 - Cd^2K^2\iota) \cdot \frac{(s_1 + s_2)^2}{s_1 s_2} \rho(z, z_\star)^2, \quad \forall z \in B_x(4\iota).$$

Here, the second inequality follows from strong convexity given by Lemma 13 and a comparison of geometric distance and Euclidean distance in normal coordinates fueled by Lemma 5. Denote for the moment that

$$\theta := 1 - Cd^2K^2\iota.$$

Plugging this back into (36), we obtain

$$\int_{\{\rho(z_1,x)\le 2\iota\}} \frac{1}{(2\pi s_1)^{d/2}} \frac{1}{(2\pi s_2)^{d/2}} \exp\left(-\frac{\rho(x,z_1)^2}{2s_1} - \frac{\rho(z_1,z_2)^2}{2s_2}\right)\mu(\mathrm{d}z)$$

$$= \int_{\{\rho(z_1,x)\le 2\iota\}} \frac{1}{(2\pi s_1)^{d/2}} \frac{1}{(2\pi s_2)^{d/2}} \exp\left(-\frac{V(z_1)}{2(s_1+s_2)}\right)\mu(\mathrm{d}z_1)$$

$$\le \int_{\{\rho(z_1,x)\le 2\iota\}} \frac{1}{(2\pi(t-s))^{d/2}} \frac{1}{(2\pi s)^{d/2}} \exp\left(-\frac{\theta(s_1+s_2)\rho(z_1,z_\star)^2}{2s_1 s_2} - \frac{\rho(x,z_2)^2}{2(s_1+s_2)}\right)\mu(\mathrm{d}z_1)$$

$$\le 4\exp\left(-\frac{\rho(x,z_2)^2}{2(s_1+s_2)}\right)\cdot \int \frac{1}{(2\pi s_1)^{d/2}} \frac{1}{(2\pi s_2)^{d/2}} \exp\left(-\frac{\theta(s_1+s_2)\|Z\|^2}{2s_1 s_2}\right)\mathrm{d}Z$$

$$= 4(2\pi(s_1+s_2))^{d/2} G(s_1+s_2,x,z_2)\cdot \frac{1}{(2\pi s_1)^{d/2}} \frac{1}{(2\pi s_2)^{d/2}} \left(2\pi\cdot\frac{s_1 s_2}{\theta(s_1+s_2)}\right)^{d/2}$$

$$\le 4\theta^{-d/2} G(s_1+s_2,x,z_2),$$

where the third-to-last line follows from change of variable to normal coordinates at $z_\star$ and from using (31) to bound the determinant; the penultimate line follows from Gaussian integration. Now, we note that

$$\theta^{-d/2} = (1 - Cd^2 K^2\iota)^{-d/2} \le \exp(2Cd^3 K^2\iota) \le 2,$$

give $\iota \le \frac{1}{100 Cd^3 K^2}$. Putting these pieces together, we proved

$$\int_{\{\rho(z_1,x)\le 2\iota\}} \frac{1}{(2\pi s_1)^{d/2}} \frac{1}{(2\pi s_2)^{d/2}} \exp\left(-\frac{\rho(x,z_1)^2}{2s_1} - \frac{\rho(z_1,z_2)^2}{2s_2}\right)\mu(\mathrm{d}z_1) \le 8G(s_1+s_2,x,z_2).$$

Iterate the above argument for the integration over $z_2,\cdots,z_{k-1}$ to obtain

$$I_{\mathsf{loc}}(s) \le 8^k \cdot G\left(\sum_{i=1}^k s_i, x, y\right) = 8^k \cdot G(t,x,y). \tag{37}$$

**Bounding the outlier integral.** Next, we show how to control $I_{\mathsf{out}}$. We first write

$$\prod_{i=1}^k G(s_i, z_{i-1}, z_i) = \frac{1}{\prod_{i=1}^k (2\pi s_i)^{d/2}} \exp\left(-\sum_{i=1}^k \frac{\rho(z_{i-1},z_i)^2}{2s_i}\right). \tag{38}$$

We claim that for any $z \in \mathcal{R}^c$, we have

$$T := \prod_{i=1}^k \exp\left(-\sum_{i=1}^k \frac{\rho(z_{i-1},z_i)^2}{2s_i(d+1)}\right)\mathbb{1}_{\rho(z_{i-1},z_i)\le\iota}\left(1 + \frac{1}{\iota s_i}\mathbb{1}_{\rho(z_{i-1},z_i)>\iota/2}\right) \le \exp\left(-\frac{\iota^2}{16td}\right). \tag{39}$$

The claim is proved at the end of this proof. It is tempting to plug this back into (38), and argue that when $t$ is polynomially small, the integrand in $I_{\mathsf{out}}$ becomes exponentially small. However, this would not work since it does not resolve the singular factors $\prod_{i=1}^k s_i^{-d/2}$ in the integrand. For this purpose, we need the following crucial "freezing" trick, which follows trivially from $1 = \frac{1}{1+d^{-1}} + \frac{1}{d+1}$:

$$\prod_{i=1}^k G(s_i, z_{i-1}, z_i)\mathbb{1}_{\rho(z_{i-1},z_i)\le\iota}\left(1 + \frac{1}{\iota s_i}\mathbb{1}_{\rho(z_{i-1},z_i)>\iota/2}\right)$$

$$= \left(\prod_{i=1}^k (1+d^{-1})^{d/2} G\big((1+d^{-1})s_i, z_{i-1}, z_i\big)\mathbb{1}_{\rho(z_{i-1},z_i)\le\iota}\right) T.$$

The idea is to keep the Gaussian behavior to resolve the $s_i^{-d/2}$ factors, and only single out a very small proportion to demonstrate exponential smallness. Plug (39) into the above identity to obtain

$$I_{\mathsf{out}}(s) \le \exp\left(-\frac{\iota^2}{16td}\right)\cdot \int_{\mathcal{R}^c}\left(\prod_{i=1}^k (1+d^{-1})^{d/2} G\big((1+d^{-1})s_i, z_{i-1}, z_i\big)\mathbb{1}_{\rho(z_{i-1},z_i)\le\iota}\right)\mu^{\otimes(k-1)}(\mathrm{d}z).$$

Integrate successively for each variable, convert to normal coordinates, and apply Lemma 5, Eqn. (31), and Gaussian integration as we did in bounding $I_{\text{loc}}$, we obtain

$$\int_{\mathcal{R}^c}\left(\prod_{i=1}^k G\big((1+d^{-1})s_i, z_{i-1}, z_i\big)\mathbb{1}_{\rho(z_{i-1},z_i)\le\iota}\right)\mu^{\otimes(k-1)}(\mathrm{d}z) \le 8^k(1+d^{-1})^{dk/2}(ct)^{-d/2} \le 32^k(ct)^{-d/2}.$$

Therefore

$$I_{\text{out}}(s) \le 32^k(ct)^{-d/2}\cdot\exp\left(-\frac{\iota^2}{16td}\right) \le 32^k G(t,x,y), \tag{40}$$

where in the last inequality we used the assumption $\rho(x,y) \le t^{5/12} = \iota/(4d)$ and $t \le 1/\operatorname{poly}(d,K)$, so that $\exp(-\iota^2/(32td)) \le \exp(-\rho(x,y)^2/(2t))$ and $\exp(-\iota^2/(32td)) = \exp(-\frac{1}{2}dt^{-1/6}) \le (2\pi t)^{-d/2}$.

**Putting things together.** We plug the bounds (37) and (40) into (35) to obtain

$$\left|\Psi_1 * r_1^{*k}\right|(t,x,y)| \le t^k P^k \int_{t\Delta^{k-1}}(8^k + 32^k)G(t,x,y)\mathrm{d}s \le \frac{40}{(k-1)!}(40t)^{2k-1}P^k G(t,x,y).$$

The desired conclusion of Lemma 24 follows from the above inequality by summing over $k$ and taking $t \le 1/\operatorname{poly}(d,K)$.

**Proof of Claim (39).** For $z \in \mathcal{R}^c$, let

$$J = \{i : \rho(z_{i-1}, z_i) > \iota/2\}.$$

By definition of $\mathcal{R}^c$, either $J$ is nonempty, or there is $i_0$ such that $\rho(x, z_{i_0}) > 2\iota$. For $i \in J$, we note that

$$\exp\left(-\frac{\rho(z_{i-1},z_i)^2}{2s_i d}\right)\left(1 + \frac{1}{\iota s_i}\right) \le \exp\left(-\frac{\iota^2}{8s_i d}\right)\left(1 + \frac{1}{\iota s_i}\right) \le \exp\left(-\frac{\iota^2}{16td}\right), \tag{41}$$

where the last inequality follows from $s_i \le t$, $\iota = 4t^{5/12}d$, and that $t \le 1/\operatorname{poly}(d,K)$. When $J$ is nonempty, we readily deduce (39) as all the other factors are $\le 1$.

When $J$ is empty, let $i_0$ be such that $\rho(x, z_{i_0}) > 2\iota$. We apply Cauchy-Schwarz to obtain

$$\sum_{i=1}^k \frac{\rho(z_{i-1},z_i)^2}{s_i} \ge \frac{1}{\sum_{i=1}^k s_i}\left(\sum_{i=1}^k \rho(z_{i-1},z_i)\right)^2 = \frac{1}{t}\left(\sum_{i=1}^k \rho(z_{i-1},z_i)\right)^2.$$

Then, by triangle inequalities, we have

$$\sum_{i=1}^{i_0}\rho(z_{i-1},z_i) \ge \rho(z_0, z_{i_0}) = \rho(x, z_{i_0}) > 2\iota, \quad \text{therefore}\quad \sum_{i=1}^k \frac{\rho(z_{i-1},z_i)^2}{s_i} \ge \frac{2\iota^2}{t}.$$

The desired claim (39) follows immediately, given that $J$ is empty. $\qquad\square$

We now have all the ingredients to prove Lemma 20.

*Proof of Lemma 20.* This follows immediately from Lemma 21, Lemma 23 and Lemma 24. Note that in applying Lemma 21, we used $r \le t^{5/12}$, thus $r^3 \le t^{5/4} \le t$. $\qquad\square$

### F.3 Proof of Lemma 21

By definition, we can compute

$$\frac{\Phi(t,x,y)}{\Psi(t,x,y)} = \exp\left((\log_x y)\cdot v - \psi(x,y)\right)\exp\left(-\|v\|^2 t/2\right).$$

Note that $\|v\| \le C\|\nabla \log p_{t_k}(Y_{t_k})\|$ by Lemma 5, which in turn is bounded by $\mathrm{poly}(d, K, \delta^{-1})$ by (29). When $t \le \frac{1}{\mathrm{poly}(d,K,\delta^{-1})} \le \frac{1}{4\|v\|^2}$, we have $|\exp(-\|v\|^2 t/2)-1| \le \|v\|^2 t \le \mathrm{poly}(d, K, \delta^{-1})t$. Therefore, it suffices to show

$$|(\log_x y) \cdot v - \psi(x,y)| \le \mathrm{poly}(d, K, \delta^{-1})r^3.$$

Recall the definition of $\psi$. Note that since $r \le \frac{1}{\mathrm{poly}(d,K,\delta^{-1})}$, when the polynomial $\mathrm{poly}(d, K, \delta^{-1})$ is sufficiently large, the geodesic $\gamma$ from $x$ to $y$ is unique and is inside $B_{Y_{t_k}}(r)$. In the normal coordinate on $Y_{t_k}$ within radius $r$, the vector field $\mathscr{S}_{t_k, Y_{t_k}}$ is represented by the constant vector $v$. We also recognize that in normal coordinate, $\gamma(1) - \gamma(0) = \log_x y$. We thus have

$$\begin{aligned}
|(\log_x y) \cdot v - \psi(x,y)| &= \left| \delta_{\alpha\beta} v^\alpha(\gamma^\beta(1) - \gamma^\beta(0)) - \int_0^1 g_{\alpha\beta}(\gamma(s))v^\alpha \frac{\mathrm{d}}{\mathrm{d}s}\gamma^\beta(s)\mathrm{d}s \right| \\
&= \left| \int_0^1 (g_{\alpha\beta}(\gamma(s)) - \delta_{\alpha\beta})v^\alpha \frac{\mathrm{d}}{\mathrm{d}s}\gamma^\beta(s)\mathrm{d}s \right| \\
&\le C \int_0^1 \|g(\gamma(s)) - I\| \cdot \|v\| \cdot \|\frac{\mathrm{d}}{\mathrm{d}s}\gamma(s)\|\mathrm{d}s \\
&\le C\|v\| \cdot \rho(x,y) \int_0^1 CK(s\rho(x,y))^2 \mathrm{d}s \\
&\le \mathrm{poly}(d, K, \delta^{-1})\rho(x,y)^3,
\end{aligned}$$

as desired. Here the penultimate line follows from Lemma 5.

### F.4 PROOF OF LEMMA 19: HANDLING EXCEPTIONAL EVENTS

*Proof of Lemma 19.* Recall that $p_0^{\mathsf{aux}} = p_N \mathsf{K}_N^{\mathsf{aux}} \mathsf{K}_{N-1}^{\mathsf{aux}} \cdots \mathsf{K}_1^{\mathsf{aux}}$ and $q_0^\star = p_N \widehat{\mathsf{K}}_N \widehat{\mathsf{K}}_{N-1} \cdots \widehat{\mathsf{K}}_1$. We need to compare the kernel $\mathsf{K}_k^{\mathsf{aux}}$ with $\widehat{\mathsf{K}}_k$, $k = 1, \cdots, N$. To apply Lemma 20, we define two auxiliary kernels $\widetilde{\mathsf{K}}_k^{\mathsf{aux}}, \widetilde{\mathsf{K}}_k$ that are "localized" version of $\mathsf{K}_k^{\mathsf{aux}}$ and $\widehat{\mathsf{K}}_k$. We show the auxiliary kernels are close to $\mathsf{K}_k^{\mathsf{aux}}$ and $\widehat{\mathsf{K}}_k$ respectively in total variation, and establish bound on $\mathsf{KL}(\widetilde{\mathsf{K}}_k^{\mathsf{aux}} \| \widetilde{\mathsf{K}}_k)$. Denote

$$\mathcal{R}_x \coloneqq \{y \in \mathcal{M} : \rho(x,y) \le h^{5/12}\}, \quad R_x^c \coloneqq \mathcal{M} \setminus \mathcal{R}_x.$$

Recall the notation $\Phi$ in the proof of Lemma 24. To distinguish the kernels at different step, we denote $\Phi_k$ as the corresponding $\Phi$ at step $k$. Define $\widetilde{\mathsf{K}}_k^{\mathsf{aux}}, \widetilde{\mathsf{K}}_k$ by

$$\widetilde{\mathsf{K}}_k^{\mathsf{aux}}(x, \mathrm{d}y) = \mathsf{K}_k^{\mathsf{aux}}(x, \mathrm{d}y)\mathbb{1}_{\mathcal{R}_x}(y) + \frac{\mathsf{K}_k^{\mathsf{aux}}(x, \mathcal{R}_x^c)}{\mu(\mathcal{R}_x^c)}\mu(\mathrm{d}y)\mathbb{1}_{\mathcal{R}_x^c},$$

$$\widetilde{\mathsf{K}}_k(x, \mathrm{d}y) = \Phi_k(h, x, y)\mathbb{1}_{\mathcal{R}_x}(y) + \frac{\int_{\mathcal{R}_x^c} \Phi_k(h, x, z)\mu(\mathrm{d}z)}{\mu(\mathcal{R}_x^c)}\mu(\mathrm{d}y)\mathbb{1}_{\mathcal{R}_x^c},$$

By converting to normal coordinate and invoking Gaussian integration in the same way as in the proof of Lemma 24, we obtain

$$\exp\left(-\frac{2}{h^{1/6}}\right) \le \int_{\mathcal{R}_x^c} \Phi_k(h, x, z)\mu(\mathrm{d}z) \le \exp\left(-\frac{1}{16h^{1/6}}\right). \tag{42}$$

When $h \le 1/\mathrm{poly}(d, K, \delta^{-1})$, it is apparent (e.g., follows from Gromov's volume comparison theorem) that $\mu(\mathcal{R}^x) \le 1/2$, thus

$$\frac{1}{2} \le \mu(\mathcal{R}_x^c) \le \mu(\mathcal{M}) = 1.$$

We observe that $\widehat{\mathsf{K}}_k$ differs from $\widetilde{\mathsf{K}}_k$ by a rejection sampling with radius $h^{1/4}$. With the above bounds and the same Gaussian integration technique, we see that the probability of rejection is bounded by

$$\mathbb{P}(\text{rejection at step } k) \le \exp\left(-\frac{(h^{1/4})^2}{16h}\right) \le \exp\left(-\frac{1}{16h^{1/2}}\right). \tag{43}$$

Summing up, We readily obtain

$$\mathsf{TV}(\widehat{\mathsf{K}}_k, \widetilde{\mathsf{K}}_k) \le \exp\left(-\frac{1}{16h^{1/6}}\right). \tag{44}$$

On the other hand, by using the stopping time argument as in the proof of (27), we have

$$\mathsf{K}_k^{\mathsf{aux}}(x, \mathcal{R}_x^c) \le \exp\left(-\frac{1}{16h^{1/6}}\right). \tag{45}$$

Therefore, the following TV bound is obvious:

$$\mathsf{TV}(\mathsf{K}_k^{\mathsf{aux}}, \widetilde{\mathsf{K}}_k^{\mathsf{aux}}) \le \exp\left(-\frac{1}{16h^{1/6}}\right), \qquad . \tag{46}$$

Now we compute $\mathsf{KL}\big(\widetilde{\mathsf{K}}_k^{\mathsf{aux}}(x, \cdot) \,\|\, \widetilde{\mathsf{K}}_k(x, \cdot)\big)$. By definition, we have

$$\mathsf{KL}\big(\widetilde{\mathsf{K}}_k^{\mathsf{aux}}(x, \cdot) \,\|\, \widetilde{\mathsf{K}}_k(x, \cdot)\big) = \underbrace{\int_{\mathcal{R}_x} \left(\log \frac{\widetilde{\mathsf{K}}_k^{\mathsf{aux}}(x, \mathrm{d}y)}{\widetilde{\mathsf{K}}_k(x, \mathrm{d}y)}\right) \widetilde{\mathsf{K}}_k^{\mathsf{aux}}(x, \mathrm{d}y)}_{=:T_1} + \underbrace{\left(\log \frac{\mathsf{K}_k^{\mathsf{aux}}(x, \mathcal{R}_x^c)}{\int_{\mathcal{R}_x^c} \Phi_k(h, x, z)\mu(\mathrm{d}z)}\right) \widetilde{\mathsf{K}}_k(x, \mathcal{R}_x^c)}_{=:T_2}.$$

We control the two terms separately.

**Controlling $T_1$.** We invoke Lemma 20 to see

$$\left| \frac{\widetilde{\mathsf{K}}_k^{\mathsf{aux}}(x, \mathrm{d}y)}{\widetilde{\mathsf{K}}_k(x, \mathrm{d}y)} - 1 \right| \le \mathrm{poly}(d, K, \delta^{-1})h.$$

Therefore, we use the elementary fact that $\log(1 + x) \ge x - 2x^2$ for $x \in [-1/2, 1/2]$ to obtain

$$\log \frac{\widetilde{\mathsf{K}}_k^{\mathsf{aux}}(x, \mathrm{d}y)}{\widetilde{\mathsf{K}}_k(x, \mathrm{d}y)} = -\log \frac{\widetilde{\mathsf{K}}_k(x, \mathrm{d}y)}{\widetilde{\mathsf{K}}_k^{\mathsf{aux}}(x, \mathrm{d}y)}$$

$$\le 1 - \frac{\widetilde{\mathsf{K}}_k(x, \mathrm{d}y)}{\widetilde{\mathsf{K}}_k^{\mathsf{aux}}(x, \mathrm{d}y)} + 2\left(\frac{\widetilde{\mathsf{K}}_k(x, \mathrm{d}y)}{\widetilde{\mathsf{K}}_k^{\mathsf{aux}}(x, \mathrm{d}y)} - 1\right)^2$$

$$\le 1 - \frac{\widetilde{\mathsf{K}}_k(x, \mathrm{d}y)}{\widetilde{\mathsf{K}}_k^{\mathsf{aux}}(x, \mathrm{d}y)} + \mathrm{poly}(d, K, \delta^{-1})h^2,$$

provided $h \le 1/\mathrm{poly}(d, K, \delta^{-1})$. Integrate with respect to $\widetilde{\mathsf{K}}_k^{\mathsf{aux}}(x, \mathrm{d}y)$ over $y \in \mathcal{R}_x$ to obtain

$$T_1 \le \widetilde{\mathsf{K}}_k^{\mathsf{aux}}(x, \mathcal{R}_x) - \widetilde{\mathsf{K}}_k(x, \mathcal{R}_x) + \mathrm{poly}(d, K, \delta^{-1})h^2$$

$$\le 2\exp\big(-\frac{1}{16h^{1/6}}\big) + \mathrm{poly}(d, K, \delta^{-1})h^2$$

$$\le \mathrm{poly}(d, K, \delta^{-1})h^2,$$

where the second line follows from (42) and (45), and the last line follows from $h \le 1/\mathrm{poly}(d, K, \delta^{-1})$ so that the exponential term is sufficiently small.

**Controlling $T_2$.** This is strightforward given (45) and (42). We obtain in the same way as above that

$$T_2 \le \exp\big(-\frac{1}{32h^{1/6}}\big) \le \mathrm{poly}(d, K, \delta^{-1}) \le h^2.$$

Summarizing the above, we have shown that

$$\mathsf{KL}\big(\widetilde{\mathsf{K}}_k^{\mathsf{aux}}(x, \cdot) \,\|\, \widetilde{\mathsf{K}}_k(x, \cdot)\big) \le \mathrm{poly}(d, K, \delta^{-1})h^2.$$

Accumulate the error over all $N$ steps using post-processing inequality and apply Pinsker's inequality, we obtain

$$\mathsf{TV}(p_0^{\mathsf{aux}} \,\|\, q_0^\star) \le \sqrt{\mathrm{poly}(d, K, \delta^{-1})h^2 N} \le \sqrt{hT}\,\mathrm{poly}(d, K, \delta^{-1}),$$

since $hN = T - \delta \le T$, as claimed. $\qquad\square$

## G  PROOF OF MAIN RESULTS

*Proof of Lemma 1.* This follows from combining Lemma 15 and Lemma 16. ☐

*Proof of Lemma 2.* This follows from Lemma 18 and our choice of schedule $hN = T - \delta \leq T$. ☐

*Proof of Theorem 1.* This follows from Lemma 1, Lemma 2, and Lemma 19. ☐

