# POLYNOMIAL CONVERGENCE OF RIEMANNIAN DIFFUSION MODELS

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

$$y_{k-1} = y_k + h\left[-b_{t_k}(y_k) + s_\theta(t_k, y_k)\right] + \sqrt{h}g_k, \quad g_k \sim \mathcal{N}(0, I_d).$$

In our driftless setting, this reduces to

$$y_{k-1} = y_k + h s_\theta(t_k, y_k) + \sqrt{h}g_k, \quad g_k \sim \mathcal{N}(0, I_d).$$

## 2.2 GEOMETRY AND NOTATION

We assume some familiarity with Riemannian geometry, and make use of standard notation. Please refer to Jost (2017); Petersen (2006) for a more in-depth treatment. In particular, we use $\alpha, \beta, \xi, \zeta$, etc., to index *coordinate representation* of tensors, and assume Einstein's summation convention. Let $(\mathcal{M}, g)$ be a connected, compact $d$-dimensional Riemannian manifold, with geodesic distance $d(\cdot, \cdot)$ and volume measure $\mu$. We assume $\mu(\mathcal{M}) = 1$. The Levi–Civita connection is denoted by $\nabla$, and the Laplace–Beltrami operator by

$$\Delta_{\mathcal{M}} f := \nabla_\alpha \nabla^\alpha f.$$

We use $T_x\mathcal{M}$ for the tangent space at $x$ and use $\exp_x : T_x\mathcal{M} \to \mathcal{M}$ for the exponential map and $\log_x$ for its local inverse on the normal neighborhood of $x$. The injectivity radius of $\mathcal{M}$ is assumed to be lower bounded by some $\rho \in (0, 1)$. Furthermore, we define *geodesic diameter* of $(M, g)$ is

$$\mathrm{Diam}(M) := \sup_{x,y\in M} d(x, y),$$

where $d(\cdot, \cdot)$ is the geodesic distance induced by $g$. We further denote $\mathrm{Rm}$ as the Riemannian curvature tensor. Geodesic ball centered at $x$ with radius $r$ is denoted $B_x(r)$.

## 2.3 HEAT FLOW, BROWNIAN MOTION, AND DIFFUSION ON $\mathcal{M}$

We also recall the setup for SDE and diffusion processes on Riemannian manifolds introduced in De Bortoli et al. (2022); Cheng et al. (2023). Let $(W_t)_{t\geq 0}$ be a standard Brownian motion in $\mathbb{R}^d$ and $U_x : \mathbb{R}^d \to T_x\mathcal{M}$ any orthonormal frame at $x$. The *Geometric Brownian motion* solves

$$\mathrm{d}X_t = U_{X_t} \circ \mathrm{d}W_t,$$

where $\circ$ denotes Stratonovich integral, and its transition density $p_t(x, y)$ with respect to $\mu$ solves the heat equation

$$\partial_t p_t(\cdot, y) = \frac{1}{2}\Delta_{\mathcal{M}} p_t(\cdot, y).$$

Equivalently, Brownian motion can be defined abstractly as the solution to the martingale problem for the operator $\frac{1}{2}\Delta_{\mathcal{M}}$. Concretely, for any $f \in C^\infty([0, \infty) \times \mathcal{M})$, the process

$$M_t^f := f(t, X_t) - f(0, X_0) - \int_0^t \left(\partial_s + \frac{1}{2}\Delta_{\mathcal{M}}\right) f(s, X_s)\mathrm{d}s$$

---

**Algorithm 1** Riemannian Score-Based Generative Models (RSGM)

---

1: Manifold $(\mathcal{M}, g)$; score $s_\theta(x,t)$; early stopping time $\delta > 0$; reverse time grid $\delta = t_0 < t_1 < \cdots < t_N = T$; step size $h = t_k - t_{k-1}$; initial $x_N \sim \mu$ (uniform distribution);
2: **for** $k \in \{N, \ldots, 1, 0\}$ **do**
3:     Choose an orthonormal frame $U_k$ at $Y_k$, which is a linear map from $\mathbb{R}^d$ to $T_{Y_k}\mathcal{M}$.
4:     $g_k \sim \mathcal{N}(0, I_d)$ in $\mathbb{R}^d$;    $G_k \leftarrow U_k g_k \in T_{Y_k}\mathcal{M}$.
5:     $b_k \leftarrow s_\theta(Y_k, t_k) \in T_{Y_k}\mathcal{M}$
6:     $\Delta_k \leftarrow hb_k + \sqrt{h}\, G_k \in T_{Y_k}\mathcal{M}$
7:     **if** $\|\Delta_k\| \leq h^{1/4}$ **then**
8:         $Y_{k-1} \leftarrow \exp_{Y_k}(\Delta_k)$
9:     **else**
10:        $Y_{k-1} \sim \mu$
11: **return** $Y_0$

---

is a martingale with respect to the natural filtration of $X$. More generally, a forward diffusion process with drift is given by

$$\mathrm{d}X_t \;=\; b_t(X_t)\,\mathrm{d}t + U_{X_t} \circ \mathrm{d}W_t,$$

with Fokker–Planck equation $\partial_t p_t = -\nabla(b_t p_t) + \frac{1}{2}\Delta_\mathcal{M} p_t$. Note that in this setting, the following process is a martingale for smooth $f$:

$$M_t^f := f(t, X_t) - f(0, X_0) - \int_0^t \left(\partial_s f + \langle b_t, \nabla f\rangle + \frac{1}{2}\Delta_\mathcal{M} f\right)(s, X_s)\mathrm{d}s. \tag{1}$$

Let $p_t$ denote the density of $X_t$ w.r.t. $\mu$, and define the *score* $s_t := \nabla \log p_t$. The time-reversal identity on manifolds yields a reverse SDE:

$$\mathrm{d}\widetilde{X}_t = (-b_t(\widetilde{X}_t) + \nabla \log p_t(\widetilde{X}_t))\mathrm{d}t + U_{\widetilde{X}_t} \circ \mathrm{d}W_t,$$

In our algorithm, the score $\nabla \log p_t$ is approximated by a trained neural network $s_\theta(t, x)$.

On compact manifolds, $-\Delta_\mathcal{M}$ admits a spectral gap $\lambda_1 > 0$. Any initial distribution mixes to the uniform distribution $\mu$ along the heat flow with rate $\mathrm{e}^{-\lambda_1 t}$. We use the *total variation* (TV) and the Kullback-Leibler distance to measure two distributions $p, q$:

$$\mathsf{TV}(p, q) = \int_\mathcal{M} |\mathrm{d}p - \mathrm{d}q|, \quad \mathsf{KL}(p \,\|\, q) = \int_\mathcal{M} \left(\log \frac{\mathrm{d}p}{\mathrm{d}q}\right)\mathrm{d}p.$$

## 3 MAIN RESULT

In this section, for completeness, we first introduce the RSGM algorithm in De Bortoli et al. (2022) with an injective radius guardrail. Then, we offer a polynomial convergence guarantee in Theorem 1. Recall that the forward diffusion on a Riemannian manifold $(\mathcal{M}, g)$ is

$$\mathrm{d}X_t \;=\; U_{X_t} \circ \mathrm{d}W_t, \qquad X_0 \sim p_0.$$

The time-reversal identity yields the *reverse-time SDE*

$$\mathrm{d}Y_t \;=\; s_t(Y_t)\,\mathrm{d}t \;+\; U_{Y_t} \circ \mathrm{d}W_t, \qquad Y_T \sim p_T. \tag{2}$$

In Algorithm 1, we provide the outline of discretized diffusion process on Riemannian manifold. In each reverse step $k \in \{N, \ldots, 1, 0\}$, we select an orthonormal frame $U_k$ at $Y_k$ to map Euclidean noise to the tangent space at $Y_k$. At line 3, we sample Gaussian noise $g_k$ and lift it to the tangent space $T_{Y_k}\mathcal{M}$, obtaining $G_k \in T_{Y_k}\mathcal{M}$. In line 5-6, we compute the reverse drift $b_k = s_\theta(Y_k, t_k)$ and propose a tangent update $\Delta_k = hb_k + \sqrt{h}G_k$. If $\Delta_k$ falls within the injective radius, we proceed to project it back to $\mathcal{M}$ with the exponential map $\exp_{Y_k}$. Otherwise, it triggers a reset $Y_{k-1} \sim \mu$. The algorithm terminates at $k = 0$ and returns the final iterate $Y_0$. Therefore, we ensure every update is well-defined in normal coordinates during the diffusion process while preserving a simple mixture kernel. We first formalize the assumptions needed for the convergence guarantee.

**Assumption 1** (Regularity). *Let $(\mathcal{M}, g)$ be a connected, compact $d$-dimensional Riemannian manifold. We assume the following conditions on $\mathcal{M}$:*

*(A1)* ***Positive injectivity radius:*** *there exists some $\rho \in (0, 1)$ such that the injective radius $\geq \rho$.*

*(A2)* ***Uniform curvature bounds:*** *there exist constants $K \geq 1$ such that*

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

**Step I. Constructing auxiliary kernels via localization.** The overarch of our proof is to decompose the total error into four components:

$$\text{error} = (\text{initialization error}) + (\text{score error}) + (\text{drift discretization error}) + (\text{BM simulation error}).$$

More concretely:

- **Initialization error** arises from initializing $Y_N$ with $\mu$ instead of the true marginal $p_N$;

- **Score error** arises from imperfect score estimation;

- **Drift discretization** error arises from approximating the continuous-time drift $s_\theta(t, Y_t)$ by its "time-frozen" counterpart $s_\theta(t_k, Y_k)$;

- **Brownian motion (BM) simulation error** is a distinctive feature of the manifold setting. Unlike in Euclidean space – where the transition kernel of Brownian motion over $[t_k, t_{k-1}]$ is exactly Gaussian with variance $(t_k - t_{k-1})$ – the transition kernel of manifold-valued Brownian motion cannot be simulated exactly by any discrete-time process, even after time discretization. This inherent inexactness gives rise to the final error term.

The first two components are relatively easy to bound using well-established tools: mixing rate bounds of heat flow (Urakawa, 2006) and Girsanov transform (Chen et al., 2023). For the drift discretization error, recent techniques developed in the Euclidean setting (Benton et al., 2024) can also be adapted with modifications that account for the manifold curvature. However, the last component – the Brownian motion simulation error – represents the core challenge in the manifold setting, which fundamentally denies a direct extension of Euclidean analysis.

In view of this, we introduce an intermediate random process that separates the drift discretization error from the BM simulation error. Constructing such a process, however, involves additional technicality. In particular, the frozen drift $s_\theta(t_k, Y_k)$ is a vector in the tangent space $T_{Y_k}\mathcal{M}$, and is therefore only well-defined at the fixed point $Y_k$. This poses a compatibility issue: as Brownian motion evolves continuously on the manifold, it immediately departs from $Y_k$, rendering the frozen

drift ill-defined. Careful geometric considerations are thus required to reconcile the piecewise-constant drift approximation with the intrinsic curvature of the manifold.

In our analysis, this is handled using localization by the construction of an auxiliary sequence of transition kernels $\mathsf{K}_k^{\mathsf{aux}}$. These kernels do not appear in the algorithm itself; they serve solely as an analytical tool to facilitate the proof. These kernels expose the behavior of the time-reverse SDE (2) when the estimated score $s_\theta$ is frozen to be a constant vector field in between discretization steps, meanwhile keeping the continuous Brownian motion.

Let $\eta : [0, \infty) \to [0, 1]$ be a smooth cutoff function, i.e., $\eta$ is decreasing, $\eta|_{[0,1]} \equiv 1$ and $\eta|_{[2,\infty)} \equiv 0$. Such a function can be chosen such that $|\eta'| + |\eta''| + |\eta'''| \le 100$.

Recall that $\rho$ is the injective radius of $\mathcal{M}$, and $K$ is the curvature bound. Define

$$\omega := \frac{c_\omega}{d^4} \min\left(\rho, \frac{1}{K}\right), \qquad \eta_\omega(x) = \eta\left(\frac{16\|x\|^2}{\omega^2}\right), \quad x \in \mathbb{R}^d, \tag{4}$$

where $c_\omega > 0$ is a small universal constant. We have $\eta_\omega|_{B(0,\omega/4)} \equiv 1$ and $\eta_\omega|_{\mathbb{R}^d \setminus B(0,\omega/2)} \equiv 0$. For $t > 0$, $x, y \in \mathcal{M}$, define the following vector field on $\mathcal{M}$:[2]

$$\mathscr{S}_{t,x}(y) = (\mathrm{d}\exp_x)_{\log_x y}\left(\eta_\omega(\log_x y) \cdot s_\theta(t, x)\right) \in T_y\mathcal{M}.$$

Intuitively speaking, $\mathscr{S}_{t,x}(\cdot)$ is the "constant" velocity field $s_\theta(t, x)$ in normal coordinates, which represents our idea of freezing the drift term for a time period. The $\mathrm{d}\exp_x$ in the formula is responsible for identifying $T_y\mathcal{M}$ with $T_x\mathcal{M}$.[3] On the other hand, the cut-off function $\eta_\omega$ is necessary to keep all our discussions restricted to the injective radius, so as to avoid pathologies of cut locus.

With this in mind, we are ready to define $\mathsf{K}_k^{\mathsf{aux}}$ as the transition kernel from time $t_k$ to $t_{k-1}$ of the reverse-time SDE

$$\mathrm{d}Y_t = \mathscr{S}_{t_k, Y_{t_k}}(Y_t)\mathrm{d}t + \mathrm{d}W_t, \quad t \in [t_{k-1}, t_k].$$

Then we define

$$p_k^{\mathsf{aux}} = p_N \mathsf{K}_N^{\mathsf{aux}} \mathsf{K}_{N-1}^{\mathsf{aux}} \cdots \mathsf{K}_{k+1}^{\mathsf{aux}}, \quad k = N, N-1, \cdots, 0.$$

**Step II. Decomposing different sources of error.** We now decompose

$$\mathsf{TV}(p_0, q_0) \le \mathsf{TV}(p_0, p_0^{\mathsf{aux}}) + \mathsf{TV}(p_0^{\mathsf{aux}}, q_0) \le \sqrt{2\mathsf{KL}(p_0 \parallel p_0^{\mathsf{aux}})} + \mathsf{TV}(p_0^{\mathsf{aux}}, q_0),$$

where the last inequality used Pinsker's inequality. To control $\mathsf{KL}(p_0 \parallel p_0^{\mathsf{aux}})$, we further introduce the counterpart of $\mathscr{S}_{t,x}$ using the exact score function $\nabla \log p_t$:

$$\mathscr{S}_{t,x}^\star(y) = (\mathrm{d}\exp_x)_{\log_x y}\left(\eta_\omega(\log_x y) \cdot \nabla \log p_t(x)\right) \in T_y\mathcal{M}.$$

A standard application of Girsanov (Chen et al., 2023; De Bortoli et al., 2022) then implies

$$\mathsf{KL}(p_0 \parallel p_0^{\mathsf{aux}}) \le \sum_{k=1}^N \int_{t_{k-1}}^{t_k} \mathbb{E}\left\|\nabla \log p_t(Y_t) - \mathscr{S}_{t_k, Y_{t_k}}(Y_t)\right\|^2 \mathrm{d}t$$

$$\le 2\underbrace{\sum_{k=1}^N \int_{t_{k-1}}^{t_k} \mathbb{E}\|\nabla \log p_t(Y_t) - \mathscr{S}_{t_k, Y_{t_k}}^\star(Y_t)\|^2 \mathrm{d}t}_{\text{drift discretization}} + 2\underbrace{\sum_{k=1}^N \int_{t_{k-1}}^{t_k} \mathbb{E}\|\mathscr{S}_{t_k, Y_{t_k}}(Y_t) - \mathscr{S}_{t_k, Y_{t_k}}^\star(Y_t)\|^2 \mathrm{d}t}_{\text{score matching}}.$$

$$\tag{5}$$

It remains to decompose $\mathsf{TV}(p_0^{\mathsf{aux}}, q_0)$. To isolate the initialization error, we introduce

$$q_0^\star = p_N \widehat{\mathsf{K}}_N \widehat{\mathsf{K}}_{N-1} \cdots \widehat{\mathsf{K}}_1.$$

By triangle inequality and post-processing inequality, we have

$$\mathsf{TV}(p_0^{\mathsf{aux}}, q_0) \le \mathsf{TV}(p_0^{\mathsf{aux}}, q_0^\star) + \mathsf{TV}(q_0^\star, q_0) \le \underbrace{\mathsf{TV}(p_0^{\mathsf{aux}}, q_0^\star)}_{\text{BM simulation}} + \underbrace{\mathsf{TV}(p_N, q_N)}_{\text{initialization}}.$$

---

[2]More precisely, the formula is obviously well-defined for $d(x, y) < \omega$. On the boundary and outside of this radius, we can smoothly extend $\eta_\omega(\log_x y) = 0$ (even though $\log_x y$ might be undefined) and therefore $\mathscr{S}_{t,x}(y) = 0$. Thus the definition extends smoothly to the whole manifold.

[3]Generally speaking, it is more natural to use parallel transport to identify different tangent spaces. However, this would later lead to a much more complicated treatment of perturbed heat equation with variable drifts. We choose to use $\mathrm{d}\exp$ here for simplicity.

**Step III. Controlling initialization and score errors.** By our design, $q_N = \mu$, and $\mathsf{TV}(p_N, q_N) = \mathsf{TV}(p_N, \mu)$. This is known as the mixing rate of heat flow in total variation norm, and has well-established bounds (e.g., Urakawa (2006)). The score-matching error, on the other hand, can be controlled with an analysis on the distortion on the Riemannian metric in normal coordinates. We compile the bounds into the following lemma.

**Lemma 1.** *There exists a universal constant $C > 0$, such that whenever $T \geq 1$, we have*

$$\mathsf{TV}(p_N, q_N) \leq \mathrm{e}^{C(K + d \log d)} \mathrm{e}^{-\frac{\lambda_1}{2}(T - \frac{1}{2})},$$

$$\sum_{k=1}^{N} \int_{t_{k-1}}^{t_k} \mathbb{E}\|\mathscr{S}_{t_k, Y_{t_k}}(Y_t) - \mathscr{S}^{\star}_{t_k, Y_{t_k}}(Y_t)\|^2 \mathrm{d}t \leq 2\varepsilon_{\mathsf{score}}^2.$$

**Step IV. Controlling drift discretization error with Itô/Stratonovich calculus and Li-Yau estimates.** The drift discretization error defined in (5) has a similar form to the discretization error for Euclidean setting (Benton et al., 2024), though additional complication arises due to non-constant $\mathscr{S}^{\star}_{t_k, Y_{t_k}}$. The idea is to study the time derivative of $\mathbb{E}\|\nabla \log p_t(Y_t) - \mathscr{S}^{\star}_{t_k, Y_{t_k}}(Y_t)\|^2$, which in view of $\partial_t \log p_t = -\frac{1}{2}\Delta_{\mathcal{M}} p_t$ (negative sign due to reverse time) involves space derivatives of $\log p_t$ up to third order. Fortunately, after applying Itô/Stratonovich calculus to simplify the expression, a key property in the proof of the Euclidean setting carries over: third-order derivatives of $\log p_t$ cancel out. The remaining first and second-order derivatives can be controlled by Li-Yau estimates on the log-gradient of the heat equation. We obtain

**Lemma 2.** *Under the assumptions as in Theorem 1, there is a universal constant $C > 0$ such that*

$$\sum_{k=1}^{N} \int_{t_{k-1}}^{t_k} \mathbb{E}\|\nabla \log p_t(Y_t) - \mathscr{S}^{\star}_{t_k, Y_{t_k}}(Y_t)\|^2 \mathrm{d}t \leq \frac{Cd^6 K^8}{\delta^3} h^2 N.$$

**Step V. Controlling BM simulation error using parametrix estimates.** Our approach is inspired by the following consequence of post-processing inequality and Pinsker's inequality:

$$\mathsf{TV}(p_0^{\mathsf{aux}}, q_0^{\star}) \leq \sqrt{2\mathsf{KL}(p_0^{\mathsf{aux}} \| q_0^{\star})} \leq \sqrt{2\sum_{k=1}^{N} \mathsf{KL}(p_k^{\mathsf{aux}}\mathsf{K}_k^{\mathsf{aux}} \| p_k^{\mathsf{aux}}\widehat{\mathsf{K}}_k)}.$$

This leads us to compare the kernel $\mathsf{K}_k^{\mathsf{aux}}$ and $\widehat{\mathsf{K}}_k$. In normal coordinates, Fokker-Planck equation shows that these two are the solutions of the heat equations with the Euclidean Laplacian and with the manifold Laplace-Beltrami operator. We utilize the Minakshisundaram-Pleijel parametrix theory (Berline et al., 2003) in geometric analysis for this comparison, and established a quantitative bound in polynomially small radius and short-time.

## 5 CONCLUSION

We developed a discrete-time theory for Riemannian diffusion models showing that a polynomial stepsize suffices for TV-accurate sampling under mild geometric conditions. Our main bound decomposes the total variation error into (i) exponential mixing at the spectral gap of $-\Delta_{\mathcal{M}}$, (ii) a term driven by score-estimation error, and (iii) a discretization error that scales as $\sqrt{h}\,\mathrm{poly}(d, K, \rho^{-1}, \delta^{-1})$. In particular, choosing stepsize $h$ polynomially small in manifold parameters achieves any prescribed TV target without exponential blow-ups in dimension or curvature. This bound gives an alternative to prior Wasserstein-type guarantees by providing a strong distributional closeness criterion aligned with generative modeling goals.

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

## A  PRELIMINARIES

We first introduce some tools we use in the rest of the proof.

**Lemma 3** (Pinsker's inequality, Polyanskiy & Wu (2025))**.** *For any two probability distributions $p, q$ on $\mathcal{M}$, we have*

$$\mathsf{TV}(p, q) \leq \sqrt{2\mathsf{KL}(p \parallel q)}.$$

**Lemma 4.** *Let $v$ be a vector field on $\mathcal{M}$. In a local coordinate on $\mathcal{M}$, we have*

$$\partial_\alpha v_\beta = (\nabla_\alpha v)^\beta - \Gamma_{\alpha\gamma}^\beta v_\gamma.$$

*Here $\Gamma_{\alpha\gamma}^\beta$ is the Christoffel symbol, defined as*

$$\Gamma_{\alpha\gamma}^\beta = \frac{1}{2} g^{\beta\delta} (\partial_\alpha g_{\gamma\delta} + \partial_\gamma g_{\alpha\delta} - \partial_\delta g_{\alpha\gamma}).$$

**Lemma 5** (Bounded Christoffel in normal coordinates)**.** *There exist universal constants $c, C > 0$ such that the following holds. Let $x \in \mathcal{M}$. In the normal coordinates $(\partial_\alpha)$ at $x$, for any $y \in \mathcal{M}$ such that $d(x, y) \leq c \min(\rho, 1/K)$, we have*

$$\|g(y) - I\| \leq CKd(x, y),$$
$$\|\partial_\alpha g_{\beta\gamma}\| \leq CKd(x, y),$$
$$\|\partial_{\alpha\beta} g_{\gamma\xi}\| \leq CK.$$

*Proof.* This follows from the well-known Taylor expansion of $g$ in normal coordinates (cf. Berline et al. (2003, Proposition 1.28)):

$$g_{\alpha\beta}(\exp_x(u)) = \delta_{\alpha\beta} - \frac{1}{3} R_{\alpha\gamma\beta\xi}(x) u^\gamma u^\xi + O\big((\|\mathrm{Rm}\| + \|\nabla\mathrm{Rm}\|)\|u\|^3\big), \quad \|u\| \leq c\min(\rho, 1/K).$$

We sketch a formal proof here for sake of convenience. Denote $f_\theta(t) = \exp_x(t\theta)$, $\theta \in T_x\mathcal{M}$, which is the constant-speed geodesic starting from $x$ with initial velocity $\theta$. Consider the Jacobi field $E_\alpha(t, \theta) = (\mathrm{d}\exp_x)_{t\theta}(e_\alpha)$, which obeys the Jacobi equation

$$\frac{D^2}{\mathrm{d}t^2}E_\alpha(t, \theta) + \mathrm{Rm}(E_\alpha(t, \theta), \dot{f}_\theta)\dot{f}_\theta = 0, \quad E_\alpha(0, \theta) = 0, \quad \frac{D}{\mathrm{d}t}E_\alpha(0, \theta) = e_\alpha,$$

where, by convention, $\frac{D}{\mathrm{d}t}$ denotes covariant derivative. Applying our curvature bound assumption, we obtain

$$\left\|\frac{D^2}{\mathrm{d}t^2}E_\alpha(t, \theta)\right\| \leq K\|\theta\|^2\|E_\alpha(t, \theta)\|.$$

Move everything to the base point $x$ by parallel transport, we are left with an ODE inequality. Use Rauch comparison for this inequality to get a bound for $\|\tau_{\exp_x(t\theta)\to x}E_\alpha(t, \theta) - te_\alpha\|$, where $\tau$ is the parallel transport map. Note that parallel transport preserves inner product, thus

$$g_{\alpha\beta}(y) = \langle E_\alpha(1, \log_x y), E_\beta(1, \log_x y)\rangle = \langle \tau_{y\to x}E_\alpha(1, \log_x y), \tau_{y\to x}E_\beta(1, \log_x y)\rangle.$$

The desired bound on $\|g(y) - I\|$ would follow from this and the previous bounds on $\|\tau_{\exp_x(t\theta)\to x}E_\alpha(t, \theta) - te_\alpha\|$. Taking derivatives yield the bounds on $\|\partial_\alpha g_{\beta\gamma}\|$ and $\|\partial_{\alpha\beta}g_{\gamma\xi}\|$ in a similar but more complicated way. $\square$

**Lemma 6.** *Fix $x \in \mathcal{M}$ and let $\rho > 0$ be the injectivity radius at $x$.

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

{d(x,z)^2}{1-t} + \frac{d(z,y)^2}{t} \geq d(x,y)^2, \qquad \forall t \in (0,1).$$

*Moreover, if $x, y, z$ are within $\iota \leq 1/\mathrm{poly}(d, K, \rho^{-1})$ distance to each other, then the function*

$$\psi(z) := \frac{d(x,z)^2}{1-t} + \frac{d(z,y)^2}{t} - d(x,y)^2$$

*is $\frac{1-Cd^2K^2\iota}{t(1-t)}$-strongly convex in the normal coordinates at $x$ (or $y$, $z$), where $C > 0$ is a universal constant.*

*Proof.* The first inequality follows from Cauchy-Schwarz inequality and triangle inequality:

$$(1-t+t) \left( \frac{d(x,z)^2}{1-t} + \frac{d(z,y)^2}{t} \right) \geq (d(x,z) + d(z,y))^2 \geq d(x,y)^2.$$

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

and similarly, when $d(Y_t, Y_{t_k}) \leq \omega/3$, we have

$$\Delta_{\mathcal{M}}\mathscr{S}^\star_{t_k, Y_{t_k}}(Y_t)^\alpha = -\mathrm{Ric}^\alpha{}_\beta\nabla^\beta \log p_{t_k}(Y_{t_k}) - g^{\beta\gamma}\left(\partial_\gamma\Gamma^\alpha_{\beta\xi} + \Gamma^\alpha_{\gamma\zeta}\Gamma^\zeta_{\beta\xi} + \Gamma^\alpha_{\zeta\xi}\Gamma^\zeta_{\beta\gamma}\right)\nabla^\xi \log p_{t_k}(Y_{t_k}).$$

**Step II. Bounding the coefficients.** Combine the above formulas with the estimates given in Lemma 5, we obtain for some universal constant $C > 0$:

$$\left\|\mathrm{Ric}^\sharp(\nabla \log p_t, \cdot)\right\| \leq CK\|\nabla \log p_t(Y_t)\|,$$

$$\left\|\nabla\mathscr{S}^\star_{t_k, Y_{t_k}}\right\| \leq CK\|\nabla \log p_{t_k}(Y_{t_k})\|, \qquad d(Y_t, Y_{t_k}) \leq \omega/3,$$

$$\left\|\Delta_{\mathcal{M}} \mathscr{S}^{\star}_{t_k, Y_{t_k}}\right\| \leq C K^2 \|\nabla \log p_{t_k}(Y_{t_k})\|, \qquad d(Y_t, Y_{t_k}) \leq \omega/3.$$

Outside the geodesic ball $B_{Y_{t_k}}(\omega/3)$, the field $\mathscr{S}^{\star}_{t_k, Y_{t_k}}$ only inside $B_{Y_{t_k}}(\omega)$. Between these two balls, we have to take into account the radial derivative of the cutoff function $\eta_\omega$, whose first order derivative is bounded by $C\omega^{-1}$ and second order derivative by $C\omega^{-2}$ by our construction (recall that $|\eta'| + |\eta''| \leq 100$). Apply Lemma 4 and Lemma 5 again, this time we bound

$$\left\|\nabla \mathscr{S}^{\star}_{t_k, Y_{t_k}}\right\| \leq C K \omega^{-1} \|\nabla \log p_{t_k}(Y_{t_k})\|,$$

$$\left\|\Delta_{\mathcal{M}} \mathscr{S}^{\star}_{t_k, Y_{t_k}}\right\| \leq C K^2 \omega^{-2} \|\nabla \log p_{t_k}(Y_{t_k})\|.$$

We now apply Itô's formula on manifold (i.e., take expectation and invoke the martingale property in (1)) and collect the above bounds to obtain (cf. Benton et al. (2024))

$$\left|\frac{\mathrm{d}}{\mathrm{d}\tau} \mathbb{E}\|\nabla \log p_t(Y_t) - \mathscr{S}^{\star}_{t_k, Y_{t_k}}(Y_t)\|^2\right|$$

$$\leq C K^2 \left(\mathbb{E}\|\nabla \log p_t(Y_t)\|^2 + \mathbb{E}\|\nabla \log p_{t_k}(Y_{t_k})\|^2 + \mathbb{E}\|\nabla^2 \log p_t(Y_t)\|^2\right)$$

$$+ C K^2 \omega^{-2} \mathbb{E}\left[\left(\|\nabla \log p_t(Y_t)\|^2 + \|\nabla \log p_{t_k}(Y_{t_k})\|^2 + \|\nabla^2 \log p_t(Y_t)\|^2\right) \mathbf{1}_{d(Y_t, Y_{t_k}) > \omega/3}\right]$$

$$\leq C K^2 \left(\mathbb{E}\|\nabla \log p_t(Y_t)\|^2 + \mathbb{E}\|\nabla \log p_{t_k}(Y_{t_k})\|^2 + \mathbb{E}\|\nabla^2 \log p_t(Y_t)\|^2\right)$$

$$+ C K^2 \omega^{-2} \sqrt{\mathbb{E}\|\nabla \log p_t(Y_t)\|^4 + \mathbb{E}\|\nabla \log p_{t_k}(Y_{t_k})\|^4 + \mathbb{E}\|\nabla^2 \log p_t(Y_t)\|^4} \sqrt{\mathbb{P}(d(Y_t, Y_{t_k}) > \omega/3)}$$

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

 some $\iota \leq 1/\mathrm{poly}(d, K, \rho^{-1}) \leq \omega$, such that for all*
$$0 < t < \frac{\iota}{\mathrm{poly}(d, K, \rho^{-1})(1 + \|s_\theta(t_k, Y_{t_k})\|^2)},$$

*we have*
$$\left| \frac{F^{\mathscr{S}}(t, Y_{t_k}, y)}{\Phi(t, Y_{t_k}, y)} - 1 \right| \leq \mathrm{poly}(d, K, \rho^{-1})t.$$

With Lemma 19 in hand, it is tempting to calculate the KL error
$$\mathsf{KL}(p_k^{\mathsf{aux}} \mathsf{K}_k^{\mathsf{aux}} \,\|\, p_k^{\mathsf{aux}} \widehat{\mathsf{K}}_k) \lesssim -\mathbb{E} \int F^{\mathscr{S}}(h, Y_{t_k}, \cdot) \log \frac{\Phi(h, Y_{t_k}, \cdot)}{F^{\mathscr{S}}(h, Y_{t_k}, \cdot)}$$
$$\lesssim \mathbb{E} \int F^{\mathscr{S}}(h, Y_{t_k}, \cdot) \left( \frac{F^{\mathscr{S}}(h, Y_{t_k}, \cdot)}{\Phi(h, Y_{t_k}, \cdot)} - 1 \right)^2$$
$$\lesssim \mathrm{poly}(d, K, \rho^{-1})h^2,$$

where we ignore the fact that Lemma 19 holds only in a small neighborhood; the first line is post-processing inequality, and the second line follows from the fact that for two distributions $p, q$, we have
$$\int p \log \frac{q}{p} = \int p \log \left( 1 + \frac{q - p}{p} \right)$$
$$\geq \int p \left( \frac{q - p}{p} - C\frac{(q - p)^2}{p^2} \right)$$
$$= -C \int p \left( \frac{q}{p} - 1 \right)^2,$$

given $\frac{q}{p} - 1$ is sufficiently small, where the last line follows from $\int p = \int q = 1$. From this, we conclude that the accumulated error along $N$ steps is bounded by $\mathrm{poly}(d, K, \rho^{-1})h^2 N = \mathrm{poly}(d, K, \rho^{-1})hT$, and the desired bound follows from Pinsker's inequality.

Apart from Lemma 19, the above computation is the essence of this proof. The rest of this section is devoted to formalizing this computation, and in particular, in handling exceptional events of exiting the polynomially small neighborhood.

### E.2 A PARAMETRIX ESTIMATE

We present the proof of Lemma 19. For simplicity, denote by $v^\alpha$ the normal coordinate representation of $s_\theta(t_k, Y_{t_k})$. Naturally, our initial test solution is the drifted heat kernel, as simulated by our discretized process:

$$\varphi_t(u) := \frac{1}{(2\pi t)^{d/2}} \exp\left(-\frac{\|u - vt\|^2}{2t}\right), \quad u \in \mathbb{R}^d.$$

Before we compare this with the manifold heat kernel, there is one subtlety we need to keep in mind. The density $\varphi_t$ is with respect to the *Lebesgue* measure on $T_x\mathcal{M}$, not with respect to the *volume* on $\mathcal{M}$. In Berline et al. (2003), this is reflected by the distinction between the spaces $\Gamma(\mathcal{M}, \mathscr{E})$ and $\Gamma(\mathcal{M}, \mathscr{E} \otimes |\Lambda|^{1/2})$. We define and compute the corresponding density on $\mathcal{M}$ as follows:

$$\Phi(t, x, y) = \varphi_t(\log_x y)\sqrt{\Delta(x,y)}, \quad \Delta(x,y) := \frac{|\det d\log_x y|^2}{\det g(y)}, \quad y \in B_x(\rho).$$

Here all quantities are computed in normal coordinates. The factor $\Delta(x, y)$ is known as the van Vleck-Morette determinant. From Lemma 5 and Lemma 6, we know that

$$\frac{1}{2} \le \Delta(x, y) \le 2, \quad \text{if } d(x, y) \le \frac{c}{d}\min(\rho, 1/K). \tag{12}$$

We consider the generalized Laplacian

$$\mathcal{H} := \frac{1}{2}\Delta_\mathcal{M} + \langle \mathscr{S}_{t_k, Y_{t_k}}, \nabla \rangle.$$

As in Lemma 19, denote by $F^{\mathscr{S}}$ the heat kernel of $\mathcal{H}$ at time $t_k - t_{k-1}$. The essence of the proof is a quantitative version of the following asymptotic expansion:

$$F^{\mathscr{S}}(t, x, y) = \Phi(t, x, y)(y) \cdot \left(1 + \sum_{i \ge 1} t^i u_i(x, y)\right), \quad \theta \to 0^+,$$

where $u_i$ are smooth functions that can be computed explicitly via a recursive formula. We will not need this formula here, but instead require $u_1$ and the remainder terms to be bounded properly. Such bounds have been well-established, which we wrap up into the following lemma.

**Lemma 20** (adapted from Berline et al. (2003)). *There exists a smooth function $u_1(x, y)$ on $\mathcal{M} \times \mathcal{M}$ such that*

$$\|u_1\|_{L^\infty(\mu)} + \|\nabla_y u_1\|_{L^\infty(\mu)} \le \operatorname{poly}(d, K, \rho^{-1})\|\mathscr{S}_{t_k, Y_{t_k}}\|_{L^\infty(\mathcal{M})},$$

*and for all $0 < t \le 1/\operatorname{poly}(d, K, \rho^{-1})$, $y \in B_x(\rho/3)$, we have*

$$\left|(\partial_t - H)\left[\Phi(t, x, y)(1 + tu_1(x, y))\right]\right| \le \operatorname{poly}(d, K, \rho^{-1})\|\mathscr{S}_{t_k, Y_{t_k}}\|_{L^\infty(\mathcal{M})}^2 \Phi(t, x, y). \tag{13}$$

*Proof.* The first inequality follows from Theorem 2.26 in Berline et al. (2003), with $H$ the same as our $H$ and therefore $F = \langle \mathscr{S}_{t_k, Y_{t_k}}, \nabla \rangle$; all these coefficients are bounded by $\operatorname{poly}(d, K, \rho^{-1})\|\mathscr{S}_{t_k, Y_{t_k}}\|_{L^\infty(\mathcal{M})}$. The second inequality follows from the proof of Theorem 2.29, item (iii) in Berline et al. (2003). In particular, the role of the cutoff function $\psi$ there is played by our $\eta_\omega$, and with the notations there, we have

$$\left|(\partial_t - H)\left[\Phi(t, x, y)(1 + tu_1(x, y))\right]\right| \le \underbrace{\operatorname{poly}(\omega^{-1})\Phi(t, x, y)}_{\partial_y \eta_\omega \text{ related terms}} + t \cdot \Phi(t, x, y) \cdot |(\mathscr{B}_y u_2)(x, y)|.$$

Here $\mathscr{B}_y$ can be viewed as a coordinate-transformed version of $H$, applied to the variable $y$ (precise definition can be found in the reference), and $u_2$ is the second order term in the expansion. Again, using Theorem 2.26 there, we can prove

$$\|\mathscr{B}_y u_2\|_{L^\infty(\mu)} \le \operatorname{poly}(d, K, \rho^{-1})\|\mathscr{S}_{t_k, Y_{t_k}}\|_{L^\infty(\mathcal{M})}^2.$$

The claimed bound follows from combining the above inequalities. $\square$

**Lemma 21** (Volterra series, Theorem 2.23 in Berline et al. (2003)). *Let*

$$\Phi_1(t, x, y) := \Phi(t, x, y)(1 + tu_1(x, y)),$$
$$r_1(t, x, y) := (\partial_t - H)\, \Phi_1(t, x, y).$$

*Define the time-space convolution operator*

$$f * g(t, x, y) = \int_0^t \int f(t - s, x, z)g(s, z, y)\mu(\mathrm{d}z)\mathrm{d}s.$$

*Then we have*

$$F^{\mathscr{S}} = H_1 - H_1 * r_1 + H_1 * r_1 * r_1 - H_1 * r_1 * r_1 * r_1 \pm \cdots,$$

*on any domain such that the series in the right hand side converges uniformly.*

**Lemma 22** (Iterative bounds for Volterra series). *There exists a radius $\iota = 1/\operatorname{poly}(d, K, \rho^{-1}) \leq \omega$ with universally constant coefficients such that the following holds. Assume $d(x, y) \leq \iota$ and*

$$0 < t \leq \frac{1}{\operatorname{poly}(d, K, \rho^{-1})(1 + \|\mathscr{S}_{t_k, Y_{t_k}}\|^2_{L^\infty(\mathcal{M})})}.$$

*Then we have*

$$|H_1 * r_1| + |H_1 * r_1 * r_1| + |H_1 * r_1 * r_1 * r_1| + \cdots \leq \operatorname{poly}(d, K, \rho^{-1})\|\mathscr{S}_{t_k, x}\|^2_{L^\infty(\mathcal{M})} t \cdot \Phi.$$

*Proof.* As in Berline et al. (2003), we first apply a cutoff function to localize to a normal neighborhood at $x$ of radius $\iota \leq \omega$. One may check that the bounds in Lemma 20 still holds, as the only change involved is replacing $\rho$ with $\operatorname{poly}(d, K, \rho^{-1})$.

We then estimate

$$|H_1 * r_1(t, x, y)|$$

$$= \left| \int_0^t \int_{B_x(\iota)} H_t(t - s, x, z)r_1(t, z, y)\mu(\mathrm{d}z)\mathrm{d}s \right|$$

$$\leq \operatorname{poly}(d, K, \rho^{-1})\|\mathscr{S}_{t_k, x}\|^2_{L^\infty(\mathcal{M})} \int_0^t \int_{B_x(\iota)} \Phi(t - s, x, z)\Phi(s, z, y)\mu(\mathrm{d}z)\mathrm{d}s$$

$$\leq \operatorname{poly}(d, K, \rho^{-1})\|\mathscr{S}_{t_k, x}\|^2_{L^\infty(\mathcal{M})} \int_0^t \int_{B_x(\iota)} \frac{1}{(2\pi(t - s))^{d/2}} \frac{1}{(2\pi s)^{d/2}} \mathrm{e}^{-\frac{d(x,z)^2}{2(t-s)} - \frac{d(z,y)^2}{2s}} \mu(\mathrm{d}z)\mathrm{d}s,$$

$$\tag{14}$$

where the second line follows from Lemma 20, and the last line follows from the definition of $\Phi$ and the bound (12).

To proceed, we make use of Lemma 12. Let $z_\star$ be a minimizer of

$$\psi(z) = \frac{d(x, z)^2}{1 - st^{-1}} + \frac{d(z, y)^2}{st^{-1}} - d(x, y)^2.$$

By obvious coercivity, when $t \leq c\iota^2$ for some small universal constant $c > 0$, we have $z_\star \in B_x(4\iota)$. By Lemma 12 and Lemma 5, we have

$$\psi(z_\star) \geq 0, \quad \psi(z) \geq (1 - Cd^2K^2\iota)\frac{t^2}{s(t - s)}d(z, z_\star)^2.$$

Here the second inequality follows from strong convexity given by Lemma 12 and a comparison of geometric distance and Euclidean distance in normal coordinates fueled by Lemma 5. Denote for the moment that

$$\theta := 1 - Cd^2K^2\iota.$$

Plug this back into (14), we obtain

$$|H_1 * r_1(t, x, y)|$$

$$\leq \text{poly}(d, K, \rho^{-1}) \|\mathscr{S}_{t_k,x}\|_{L^\infty(\mathcal{M})}^2 \int_0^t \int_{B_x(\iota)} \frac{1}{(2\pi(t-s))^{d/2}} \frac{1}{(2\pi s)^{d/2}} \mathrm{e}^{-\frac{\theta t d(z,z_\star)^2}{2s(t-s)} - \frac{d(x,y)^2}{2t}} \mu(\mathrm{d}z) \mathrm{d}s$$

$$\leq \text{poly}(d, K, \rho^{-1}) \|\mathscr{S}_{t_k,x}\|_{L^\infty(\mathcal{M})}^2 \mathrm{e}^{-\frac{d(x,y)^2}{2t}} \int_0^t \int_{B(0,\iota)} \frac{1}{(2\pi(t-s))^{d/2}} \frac{1}{(2\pi s)^{d/2}} \mathrm{e}^{-\frac{\theta t \|Z\|^2}{2s(t-s)}} \mathrm{d}Z \mathrm{d}s$$

$$\leq \text{poly}(d, K, \rho^{-1}) \|\mathscr{S}_{t_k,x}\|_{L^\infty(\mathcal{M})}^2 \frac{1}{(2\pi t)^{d/2}} \mathrm{e}^{-\frac{d(x,y)^2}{2t}} \theta^{-d/2} \int_0^t \mathrm{d}s$$

$$\leq \text{poly}(d, K, \rho^{-1}) \|\mathscr{S}_{t_k,x}\|_{L^\infty(\mathcal{M})}^2 \theta^{-d/2} \Phi(t,x,y) \cdot t,$$

where the third line follows from change of variable to normal coordinates at $z_\star$ and use (12) to bound the determinant, the last line uses again the definition of $\Phi$ and (12), and the penultimate line follows from Gaussian integration

$$\int \frac{(\theta t)^{d/2}}{(2\pi s(t-s))^{d/2}} \mathrm{e}^{-\frac{\theta t \|Z\|^2}{2s(t-s)}} \mathrm{d}Z = 1.$$

Now, we note that

$$\theta^{-d/2} = (1 - Cd^2 K^2 \iota)^{-d/2} \leq \exp(2Cd^3 K^2 \iota) \leq 2,$$

give $\iota \leq \frac{1}{100 C d^3 K^2}$. Putting these pieces together, we proved

$$|H_1 * r_1| \leq t \cdot \underbrace{\text{poly}(d, K, \rho^{-1}) \|\mathscr{S}_{t_k,x}\|_{L^\infty(\mathcal{M})}^2}_{=:\kappa} \Phi.$$

Iterate this above process to see that the $n$-th convolution is bounded by

$$|H_1 * r_1^{(j)}| \leq (\kappa t)^j \Phi.$$

Therefore,

$$\left| \sum_{j=1}^\infty (-1)^{j-1} H_1 * r_1^{(j)} \right| \leq \sum_{j=1}^\infty (\kappa t)^j \Phi \leq \frac{\kappa t}{1 - \kappa t} \Phi \leq 2t\Phi,$$

given $t \leq 1/(4\kappa)$. Recalling the definition of $\kappa$, the lemma is proved. $\qquad\square$

We now have all the ingredients to prove Lemma 19.

*Proof of Lemma 19.* This follows immediately from Lemma 21 and Lemma 22. $\qquad\square$

### E.3 HANDLING EXCEPTIONAL EVENTS

*Proof of Lemma 18.* Denote

$$u(t,x,y) = \frac{1}{t} \left( \frac{\Phi(t,x,y)}{F\mathscr{S}(t,x,y)} - 1 \right), \quad d(x,y) \leq \iota.$$

Then Lemma 19 implies

$$|u| \leq \text{poly}(d, K, \rho^{-1}), \quad d(x,y) < \iota, \ 0 < t < \frac{\iota}{\text{poly}(d, K, \rho^{-1})(1 + \|s_\theta(t_k, Y_{t_k})\|^2)}.$$

As a first simplification, we invoke (A3) in Assumption 1 to see

$$\|s_\theta(t_k, Y_{t_k})\| \leq \text{poly}(d, K, \rho^{-1}, \delta^{-1})(1 + \|\nabla \log p_{t_k}(Y_{t_k})\|).$$

Then we can control $\|\nabla \log p_{t_k}(Y_{t_k})\|$ as in the proof of (11), resulting in

$$\|s_\theta(t_k, Y_{t_k})\| \leq \text{poly}(d, K, \rho^{-1}, \delta^{-1}).$$

Therefore, the conclusion of Lemma 19 simplifies to

$$|u| \leq \text{poly}(d, K, \rho^{-1}), \quad d(x,y) < \iota, \ 0 < t < \frac{\iota}{\text{poly}(d, K, \rho^{-1}, \delta^{-1})}.$$

Now, notice that the integration of $F^{\mathscr{S}}(h, Y_{t_k}, \cdot)$ outside $B_{Y_{t_k}}(\iota)$ is bounded by Lemma 7 and Lemma 8:

$$\int_{\mathcal{M} \setminus B_{Y_{t_k}}(\iota)} F^{\mathscr{S}}(h, Y_{t_k}, y) \mu(\mathrm{d}y) \leq \exp\left(-\frac{c\iota^2}{h}\right), \tag{15}$$

and similarly

$$\int_{B_{Y_{t_k}}(\iota)} \Phi(h, Y_{t_k}, y) \mu(\mathrm{d}y) \geq 1 - \exp\left(-\frac{c\iota^2}{h}\right).$$

We may extend $u(h, \cdot, \cdot)$ to a bounded function with value in $[-\frac{1}{h}, h\exp(C\iota^2/h))$, defined on the whole $\mathcal{M}$ obeying the following two restrictions:

$$\int_{\mathcal{M}} u(h, Y_{t_k}, y) F^{\mathscr{S}}(h, Y_{t_k}, y) \mu(\mathrm{d}y) = 0, \tag{16}$$

$$\int_{\mathcal{M} \setminus B_{Y_{t_k}}(3\iota)} h u(h, Y_{t_k}, y) F^{\mathscr{S}}(h, Y_{t_k}, y) \mu(\mathrm{d}y) \leq \exp\left(-\frac{c\iota^2}{h}\right). \tag{17}$$

The construction of such an extension is routine, for example, can be a constant function of order $h\exp(C\iota^2/h)$ outside $B_{Y_{t_k}}(2\iota)$, and tuned in $B_{Y_{t_k}}(2\iota) \setminus B_{Y_{t_k}}(\iota)$ to fulfill the restrictions.

With this extension, we find that

$$\widetilde{\mathsf{K}}_k := \widetilde{\Phi}(h, x, y) := (h u(h, x, y) + 1) F^{\mathscr{S}}(h, x, y)$$

is a kernel on $\mathcal{M}$, by (16). Moreover, (15) and (17) together imply

$$\mathsf{TV}(p_{Y_{t_k}} \widehat{\mathsf{K}}_k,\ p_{Y_{t_k}} \widetilde{\mathsf{K}}_k) \leq \exp\left(-\frac{c\iota^2}{h}\right).$$

Therefore, by post-processing inequality,

$$\mathsf{TV}(p_N \widetilde{\mathsf{K}}_N \cdots \widetilde{\mathsf{K}}_1,\ q_0^\star) \leq N\exp\left(-\frac{c\iota^2}{h}\right) \leq N\exp(-c'h^{-1/2}) \leq h, \tag{18}$$

by choosing $\iota = \frac{1}{10} h^{1/4}$ and $h \leq \frac{1}{\mathrm{poly}(d, K, \rho^{-1}, \delta^{-1})}$.

On the other hand, the KL computation following Lemma 19, together with the bound $u \leq h\exp(C\iota^2/h)$ so that $\log \frac{\widetilde{\Phi}(h,x,y)}{F^{\mathscr{S}}(h,x,y)} \leq C\iota^2/h$, show that

$$\begin{aligned}
\mathsf{KL}(p_0^{\mathsf{aux}} \parallel p_N \widetilde{\mathsf{K}}_N \cdots \widetilde{\mathsf{K}}_1) &\leq \sum_{k=1}^{N} \mathrm{poly}(d, K, \rho^{-1}, \delta^{-1}) h^2 + N^2 \frac{C\iota^2}{h} \exp\left(\frac{-c\iota^2}{h}\right) \\
&\leq \mathrm{poly}(d, K, \rho^{-1}, \delta^{-1}) h^2 N \\
&\leq \mathrm{poly}(d, K, \rho^{-1}, \delta^{-1}) hT,
\end{aligned}$$

where the penultimate line follows from choosing $h \leq \frac{\iota^2}{\mathrm{poly}(d, K, \rho^{-1}, \delta^{-1})}$, and the last line follows from the choice of schedule $hN = T - \delta \leq T$. The final TV bound follows from combining the above estimates and Pinsker's inequality. $\qquad \square$

# F    PROOF OF MAIN RESULTS

*Proof of Lemma 1.* This follows from combining Lemma 14 and Lemma 15. $\qquad \square$

*Proof of Lemma 2.* This follows from Lemma 17 and our choice of schedule $hN = T - \delta \leq T$. $\qquad \square$

*Proof of Theorem 1.* This follows from Lemma 1, Lemma 2, and Lemma 18. $\qquad \square$

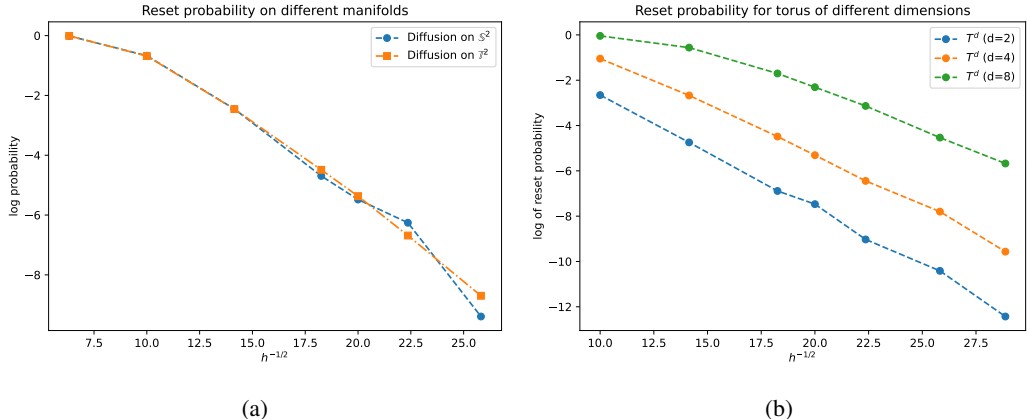

(a)                       (b)

Figure 1: Reset probabilities on spheres and tori. In Figure 1a, we examine the relationship between $h^{-1/2}$ and the log of the reset probability of Algorithm 1 on both sphere $\mathbb{S}^2$ and torus $\mathbb{T}^2$ under the reset rules of Algorithm 1. In both cases, we see that the reset probability decays exponentially, confirming the conclusion in (18). In Figure 1b, we examine the same statistics on high-dimensional tori, and we find increasing $d$ only shifts the curves to the right but leaves the exponential decay rate in $h^{-1/2}$ unchanged.

# G  SIMULATIONS

In this section, we verify the results in Theorem 1 on compact manifolds by measuring the exit probability in the reverse steps of Algorithm 1 and the TV distance between the target distribution (a Gaussian mixture) and the recovered distribution.

**Reset probability on $\mathbb{S}^2$ and $\mathbb{T}^2$**   . We start by examining the total reset probability on the unit 2-sphere $\mathbb{S}^2$ and on the 2-torus $\mathbb{T}^2$. We run the backward process in Algorithm 1 with different stepsizes $h$ in each setting with $p_0$ being Gaussian mixture (see below for definition), and record the fraction of trials whose tangent update $\Delta_k = h\, s_{t_k}(Y_k) + \sqrt{h}\, G_k$ violates $\|\Delta_k\| \leq h^{1/4}$ at least once among all steps. On both manifolds, Figure 1a shows a clear linear trend of the logarithm of reset probability of against $h^{-1/2}$, consistent with our exponentially small tails.

**High-dimensional torus.**   We extend this experiment to the $d$-dimensional flat torus $\mathbb{T}^d$ for different stepsizes. Figure 1b reports the logarithm of the reset probability versus $h^{-1/2}$ for $d \in \{2, 4, 8\}$. Increasing $d$ raises the baseline reset rate, yet the slope of the decay remains essentially unchanged—resets remain exponentially rare as $h \downarrow 0$, which aligns with our analysis.

**TV accuracy on $\mathbb{T}^d$ with warped Gaussian mixture.**   Finally, we assess distributional accuracy in TV for a *warped Gaussian mixture* target on $\mathbb{T}^d$, $d \in \{1, 2, 3\}$. Here the warped Gaussian distribution is defined to be the push-forward of the Gaussian distribution under the universal covering $\mathbb{R}^d \to \mathbb{T}^d$ given by $(x_1, \cdots, x_d) \mapsto (\mathrm{e}^{i2\pi x_1}, \cdots, \mathrm{e}^{i2\pi x_d})$, and warped Gaussian mixture is similarly the push-forward of Gaussian mixture in $\mathbb{R}^d$. The result is depicted in Figure 2, which confirms that the total variation decays fast with the increase of the number of steps.

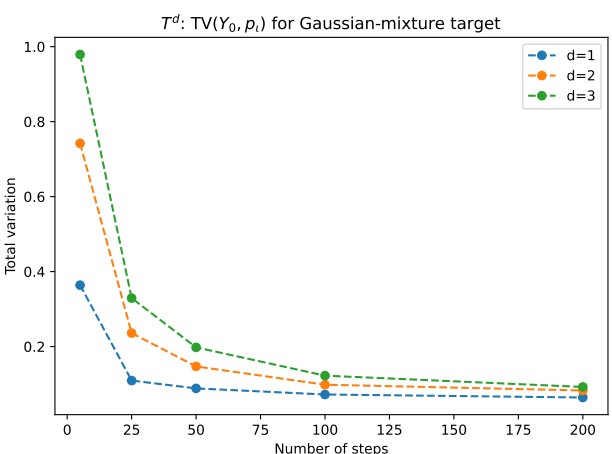

Figure 2: TV distance on $\mathbb{T}^d$ with a warped Gaussian-mixture target. The total variation is estimated with a kernel density estimator.