# OpenReview forum: "Polynomial Convergence of Riemannian Diffusion Models"
_ICLR.cc/2026/Conference — ICLR 2026 Poster_

### Official Review · Reviewer_qEnZ · 2025-10-28

**Soundness:** 3
**Presentation:** 3
**Contribution:** 2
**Rating:** 4
**Confidence:** 3

**Summary:**

This paper extends the Riemannian diffusion model, studying the convergence behavior under the L2 accurate score estimate. The paper shows that a polynomially small stepsize is sufficient to guarantee small sampling error in TV distance with only mild curvature assumptions. For technical contribution, the proof uses Li-Yau gradient bounds for the heat kernel, a localization scheme for the drift, and Minakshisundaram–Pleijel parametrix for controlling deviations from the heat flow and its discretized proxy.

**Strengths:**

- The paper presents the first polynomial TV convergence results for Riemannian diffusion models to the best of my knowledge. While it extends the theoretical results in Riemannian diffusion models, the paper proves TV convergence instead of Wasserstein convergence under mild geometric conditions.

- The paper provides analytical techniques to prove the polynomial convergence: Li-Yau estimate for the heat kernel, localization of the drift fields, and parametrix estimates. These tools may be meaningful for bridging diffusion theory and geometry and could be used for developing a sampling method for Riemannian diffusion models.

**Weaknesses:**

- I'm unsure what the impact of the paper is and how it contributes to the field. While the paper gives a theoretical guarantee for the Riemannian diffusion model, the empirical results from De Bortoli et al already validated that it worked in diverse settings. I agree that the paper provided new tools to use for future study in this field, but it is unclear how important it is to show polynomial TV convergence for the already working model. If the paper presented a recipe for stepsize (with empirical results) or an advanced sampling method, it should have been a strong signal for acceptance. I would appreciate it if the authors could give an explanation of the importance of the work.

- While the theoretical contributions are important, I believe empirical results, at least on small toy settings, are necessary. I cannot say that the details of the proof are all correct due to an insufficient understanding of differential geometry. Empirical results should have complemented this, giving strong evidence.

- Is the result only applicable to the Riemannian diffusion model from De Bortoli's design? For example, there are several designs of diffusion models, such as [Chin-Wei Huang et al., 2022], [Lou et al., 2023], Riemannian Flow Matching [Chen and Lipman, 2024], or Riemannian Diffusion Mixture [Jo et al., 2024], and I wonder if the convergence results or its argument could be applied to different designs.

[Chin-Wei Huang et al., 2022] Riemannian Diffusion Models, NeurIPS 2022
[Lou et al., 2023] Scaling Riemannian Diffusion Models, NeurIPS 2023
[Chen and Lipman] Flow Matching on General Geometries, ICLR 2024
[Jo et al., 2024] Generative Modeling on Manifolds Through Mixture of Riemannian Diffusion Processes, ICML 2024

**Questions:**

Please address the questions raised in the weakness section

---

> ### Author Response · Authors · 2025-11-25
>
> Thank you for your careful review and for recognizing our theoretical advances in improving the iteration complexity of RSGM from exponential to polynomial. We would like to address your concerns below.
>
> - As you pointed out, we focused on theoretical analysis based on the algorithm of De Bortoli's design. Our result marks a transition of understanding for such algorithms: from knowing that it must converge in exponentially many steps (practically intractable) to knowing such in polynomially many steps (practically efficient). While we did not design new algorithm or propose advanced sampling method, our work provides the theoretical grounding needed to explain the benign performance observed in De Bortoli et al. (2022). Such understandings may provide crucial insights to the development of improved sampling algorithms, as demonstrated by some recent accelerated algorithms for Euclidean diffusion models (Li and Jiao, 2024) and discrete diffusion models [R1]. To better illustrate our results, we ran experiments to plot certain ''anatomy'' of the algorithm in Appendix, like the reset probability (Figure 1 in our revision) and the total variation convergence.
> - Our analysis can be extended to other stochastic samplers based on simulating the reverse-time Brownian SDE on a Riemannian manifold. This covers Riemannian Diffusion Models (Huang et al., 2022) and their “Scaling RDM” variant (Lou et al., 2023). These algorithms, except for their ODE variant, have similar reverse diffusion SDE and score matching objective as that of RSGM (for example, see Theorem 4 in Huang et al. (2022) and the the remarks thereafter). For the “Riemannian Diffusion Mixture” (Jo and Hwang, 2024), the proposed model is also a Brownian SDE with a bridge‑type drift, which again fits into our analysis framework. Finally, “Flow Matching on General Geometries” (Chen and Lipman, 2024) trains a deterministic probability‑flow ODE rather than the reverse Brownian SDE; our Brownian‑simulation error bound are specific to SDE sampling and do not apply directly. An analogous TV bound for the ODE sampler requires analyzing the ODE discretization error, which requires additional techniques that are out of our current scope.
>
> We hope the above have answered your questions and provided some more context. Please let us know if you have further questions or suggestions.
>
> [R1] Ren, Yinuo, et al. "Fast solvers for discrete diffusion models: Theory and applications of high-order algorithms." arXiv preprint arXiv:2502.00234 (2025).

---

### Official Review · Reviewer_5iY7 · 2025-10-31

**Soundness:** 3
**Presentation:** 3
**Contribution:** 3
**Rating:** 6
**Confidence:** 2

**Summary:**

The paper “Polynomial Convergence of Riemannian Diffusion Models” presents a significant advancement over the work of De Bortoli et al. (2022), which introduced generative diffusion processes on Riemannian manifolds. This framework, known as the Riemannian Score-Based Generative Model (RSGM), is motivated by the observation that many datasets, such as those arising in climate science and high-energy physics, naturally lie on Riemannian-like manifolds. De Bortoli et al. demonstrated that, in such settings, performing diffusion directly on the underlying manifold can substantially improve generative performance. However, their approach suffered from a severe limitation: achieving a small divergence between the model and target distributions required an exponentially small discretization step size, leading to prohibitive computational costs. The current paper overcomes this issue by proving that, under mild assumptions on the manifold structure, one can instead use a polynomially small step size to achieve comparable accuracy. This result dramatically improves the practical feasibility of RSGMs and paves the way for their broader application in real-world generative modeling tasks.

**Strengths:**

The work has several notable strengths:

1. The presentation is clear and pedagogical, making the material accessible even to readers without a strong background in differential geometry.

2. The discussion of related work is thorough and well-integrated, situating the paper within a coherent and well-defined research landscape.

3. The analytical results appear sound, and the way of deriving the bound over the total variation, as a measure of the distance between two distributions, could be employed in characterizing other contexts, such as classifier-free-guidance and other diffusion routines where an exact theory is lacking and sources of errors are similar.

**Weaknesses:**

The results are primarily analytical and the central goal of the work is to derive new performance bounds for RSGMs. I hence appreciate the total analytical vocation of the manuscript. On the other hand, one might think that the total absence of experimental validation could appear as a weakness of the work. De Bortoli et al. (2022) compare RSGM with other manifold-based diffusive routines on manifold supported datasets, like climate science spherical data. They quantify the quality of the performance in terms of the log-likelihood achieved by the model. I was wondering whether the authors of the current paper ever evaluated the idea of running similar experiments with both RSGM and other procedures (e.g. Mixture of Kent, Moser Flow etc.) and show that RSGM can reach a comparable degree of the performance still employing a polynomial number of steps. Such scaling could be analyzed as a function of relevant control parameters, including manifold size and early stopping time. Another possible, and possibly equivalent, experiment would instead imply verifying that Riemann supported real datasets verify the assumptions of the current analysis.

In addition to addressing the questions outlined in the following section of this review, I would appreciate if the authors could argue around this point more in detail.

**Questions:**

I will now proceed to ask questions and raise issues that I would like the authors to address in order to both finalize my personal judgment over the paper and improve the manuscript.

**Questions:**

1. Why do I re-start from the uniform distribution if the diffusion step from $Y_k$ exits the injective radius ? What happens if one restarts from $Y_k$ and samples new noise? How often would the reset take place in practical applications?
2. In the continuous-time variance-exploding prescription, a common choice for the prior distribution in backward diffusion is a wide Gaussian distribution, with zero mean and variance equal to the horizon $T$. Are there other prior distributions, alternative to the uniform one, that can reduce the initialization error? Can Urakawa (2006) be generalized to other initializations?
3. Recent studies in this field, such as Ventura et al. (2025) "Manifolds, Random Matrices and Spectral Gaps: The geometric phases of generative diffusion", Wang et al. (2025) "Diffusion Models Learn Low-Dimensional Distributions via Subspace Clustering", Stanczuck et al. (2024) "Diffusion Models Encode the Intrinsic Dimension of Data Manifolds" show that, when data are supported by a low-dimensional manifold, the geometry of this manifold  progressively emerges in the backward diffusion in the Euclidean space, without the need of constraining the trajectory on the manifold itself. I would ask to the authors whether it would be possible to study the time dependence (that I guess translates into studying the $\delta$ dependence) of the TV and see whether its upper bound becomes tighter at smaller times, when the geometry of the target manifold already emerges even when diffusing in the Euclidean space.

**Readability issues:**

1. [101] What does BM stand for?
2. [157-158] Here and across all the paper there is a conflict between the dimension of the ambient Euclidean space and the dimension of the manifold used in the subsequent proofs.
3. [270-271] You should probably state that M has dimension d, since it appears implicitly in A3.
4. Both in [306-307] and the very beginning in [088] the authors may introduce a bit better the role of the error constant 𝜀, as a quantity employed to tune the horizon time $T$ and from that the discretization stepsize.
5. [395-396] May be useful to uniform the nomenclature for ℐ with respect to the previous expression in [394-395].

**Typos**:
1. [073] “exponential”
2. [362] “time frozen”
3. [324-325] "poly-logarithmic"

I take advantage for thanking the authors for the nice job and I wait for their response.

---

> ### Author Response · Authors · 2025-11-25
>
> Thank you for your thoughtful and thorough review! We are happy for your recognition of our contributions that improves previously intractable exponential complexity bound to a polynomial one, which improves the practical feasibility of RSGMs, and would like to address your concerns below.
>
> ***Regarding the Weaknesses part:***
> - Since this paper focuses on the analysis of RSGM and aims to provide a guarantee of efficiency for it, we would like to restrict discussion of potential experiments to the RSGM algorithm. We added numerical experiments depicting the reset probability and total variation distance of our algorithm on synthetic data in the revision (Appendix G) to illustrate our theory.
>
> - Verifying the assumption on the manifold geometry (Assumption 1) is usually straightforward in real-world applications, including [R1, R2]. On the other hand, our assumption on score function is commonly adopted in theoretical analysis of diffusion models (Chen et al., 2023; Li et al., 2024; Benton et al., 2024), even though its empirical correctness is still under active investigation even for Euclidean diffusion models [R3-R5]).
>
> ***Regarding your questions:***
>
> - The rejection sampling step is purely technical and has no practical consequence. The restarting distribution is an arbitrary choice, and restarting from $Y_k$ works equally well for our analysis and leads to the same theoretical guarantee. In our proof (cf. Eqn (18)), we have shown that the reset probability is exponentially small (of order $\\exp(-\\Omega(1/\\sqrt{h}))$), which is negligible for practical purpose. In the revised draft, we have added simulations of the proposed algorithm on spheres and tori, which confirm the claim mentioned above.
>
> - Our choice of uniform distribution is actually an analogue of the Gaussian distribution in the Euclidean setting: they are both the limiting distribution of the standard heat equation, i.e., the variance-exploding SDE with zero drift and identity diffusion matrix. More generally, our analysis also extends to other choice of the ``prior distribution in backward diffusion'', as long as it can be represented as the limiting distribution of a SDE of the form $\\mathrm{d}x_t = f_t(x_t) \mathrm{d}t + \\mathrm{d} W_t$, $t \\to \\infty$, under mild regularity assumptions on $f_t$. In particular, this includes all distributions with density of the form $\\mathrm{e}^{-V(x)}$ where $V$ satisfies isoperimetry properties, which can be sampled efficiently with Langevin dynamics [R6].
>
> - While it is an exciting surging direction to study the adaptivity to manifold for diffusion processes in the ambient Euclidean space, the current theoretical understanding of them when the data is hard-constrained on the manifold and the distributional closeness is measured by $f$-divergence supported on the manifold. Therefore, our work complements these studies in a parallel direction. Regarding your question about whether the bound gets better at small time, unfortunately, we have not observed such phenomena in our analysis. Nevertheless, we agree that it is an interesting problem to study what the adaptivity of Euclidean diffusion to low-dimensional structure implies for Riemannian diffusion models.
>
> In addition, we thank the reviewer for pointing out the readability issue. We have fixed these issues as per your suggestion. We hope the above have answered your questions and provided some more context. Please let us know if you have further questions or suggestions.
>
> [R1] Mathieu, Emile, and Maximilian Nickel. "Riemannian continuous normalizing flows." Advances in neural information processing systems 33 (2020): 2503-2515.
> [R2] Watson, Joseph L., et al. "De novo design of protein structure and function with RFdiffusion." Nature 620.7976 (2023): 1089-1100.
> [R3] Gatmiry, Khashayar, Jonathan Kelner, and Holden Lee. "Learning mixtures of gaussians using diffusion models." arXiv preprint arXiv:2404.18869 (2024).
> [R4] Ghio, Davide, et al. "Sampling with flows, diffusion, and autoregressive neural networks from a spin-glass perspective." Proceedings of the National Academy of Sciences 121.27 (2024): e2311810121.
> [R5] George, Anand Jerry, Rodrigo Veiga, and Nicolas Macris. "Denoising score matching with random features: Insights on diffusion models from precise learning curves." arXiv preprint arXiv:2502.00336 (2025).
> [R6] Vempala, Santosh, and Andre Wibisono. "Rapid convergence of the unadjusted langevin algorithm: Isoperimetry suffices." Advances in neural information processing systems 32 (2019).

---

> > ### Comment · Reviewer_5iY7 · 2025-11-25
> >
> > I thank the authors for having addressed my points clearly. And, mostly, for having added Appendix G to the manuscript.
> > I only ask to the authors whether they could clarify more explicitly both here and the in the main text, the reason for which the experiments in Fig. 1 validate Theorem 1.

---

> > > ### Author Response · Authors · 2025-11-30
> > >
> > > We thank the reviewer for the timely response and offer the following explanation of the connection between Theorem 1 and our experiments. In short, our experiments demonstrate: (i) the extra rejection sampling step does not harm the practicality of Algorithm 1, as shown by Figure 1; (ii) the actual performance of Algorithm 1 is consistent with what our theory predicts: a polynomially decaying TV error in the number of steps, as shown by Figure 2.
> > >
> > > More concretely, Theorem 1 provides a TV-error bound for Algorithm 1. An important step in the proof (in Appendix E.3) showed that the probability of triggering a reset is exponentially small in $h^{-1/2}$, which allows us to obtain the bound on Lemma 18 and, eventually, Theorem 1.  Our experiments empirically verify the above result. In Figures 1(a) and 1(b), we verified that the probability of triggering a reset is exponentially small in $h^{-1/2}$ across various manifolds, thereby validating our claim. In Figure 2, we evaluated the TV error of the algorithm's output, and the result clearly shows polynomial convergence, which aligns with our theory. We have also edited Appendix G to reflect this connection.
> > >
> > > We hope the above have answered your questions and provided some more context. Please let us know if you have further questions or suggestions.

---

### Official Review · Reviewer_FiF9 · 2025-11-01

**Soundness:** 2
**Presentation:** 2
**Contribution:** 2
**Rating:** 2
**Confidence:** 2

**Summary:**

The paper studies Riemannian score-based diffusion models on a compact $d$-dimensional manifold $(M, g)$. Under bounded geometry (injectivity radius $\rho > 0$; curvature/tensor bounds), an $L^2$-accurate score (Assumption 2) and a mild regularity bound on the learned score (Assumption 1. (A3)), it proves that a polynomially small step size suffices to achieve small total-variation (TV) error between the sampler's output law and an early-stopped target $p_{\delta}$. The bound decomposes into (i) heat-flow mixing at rate $\lambda_1$ (the spectral gap of $-\Delta_M$), (ii) a term linear in the $L_2$ score error, and (iii) a discretization term scaling like $\sqrt{hT}\mathrm{poly}(d,K,\rho^{-1},\delta^{-1})$. Technically, the proof combines Li–Yau/Hamilton/Han–Zhang heat‑kernel/gradient estimates, a localization that freezes the drift in normal coordinates, and a Minakshisundaram–Pleijel parametrix control to bound the Brownian‑motion simulation error that arises on manifolds. The algorithm includes a "reset to the uniform measure $\mu$" safeguard when a proposed step exits the injectivity radius.

**Strengths:**

- **Problem significance.** TV‑accurate sampling on manifolds with polynomial step size is a meaningful strengthening over prior Riemannian SGM bounds that scale poorly with $d$; cf. De Bortoli et al. 2022.
- **Technique.** The parametrix approach to BM simulation error is original in this context and technically apt; it uses BGV-style local expansions and Volterra series to compare kernels; Berline-Getzler-Bergne (https://link.springer.com/book/10.1007/978-3-642-58088-8).
- **Error decomposition.** Splitting the total error into mixing / score /discretization / BM simulation aligns with the Euclidean playbook but addresses the manifold-specific obstacle (non-Gaussian BM kernel).
- **Assumptions.** Works under bounded geometry and does not require smooth/strictly positive data density; early stopping avoids small-time blow-ups.

**Weaknesses:**

1. **Reset-analysis mismatch.** Provide a uniform upper bound on
$$\sum^N_{k=1} \mathrm{Pr} (\| \Delta_k \| > h^{1/4}) $$
and propagate it into the final TV bound. One can combine drift bounds (from Li-Yau/Hamilton/Han-Zhang estimates) with a BDG-type tail on the martingale term to show a sub-Gaussian decay in $(\omega^2 / (t_k - t))$; but the constants and dependence on $d$, $K$, $\rho^{-1}$, and $\delta^{-1}$ must be explicit.
2. **Parametrix generator sign.** Unify the generator throughout §E.2 (either $-\frac{1}{2}\Delta_M + \braket{S, \nabla}$ or $+\frac{1}{2}\Delta_M + \braket{S, \nabla}$) ; then re-check Lemma 19 and the Volterra bounds against the exact BGV statements (e.g. Thm. 2.23, 2.26, 2.29).
3. **Heat-kernel sup bound.** Replace $$\sup \log H(s/2, \cdot,\cdot) \lesssim \frac{d \log d}{s} + Ks $$ by a standard Li–Yau form: $\log \sup_{x,y} H(t,x,y) \lesssim \frac{d}{2}\log(1/t) + CKt$ (possibly after inserting volume factors via Bishop-Gromov/Günther). Re-derive the $\mathbb{E}[\log^4(\cdot)]$ control under this bound.
4. State geometric assumptions precisely. Add an explicit Ricci lower bound ($ \mathrm{Ric} \geq - K_{\mathrm{Ric}} $) and track constant dependencies in Lemma 11 and downstream. This is standard in Hamilton's matrix Harnack and Han-Zhang's Hessian upper bounds.
5. Girsanov conditions. Insert a lemma verifying Novikov/Kazamaki under (A3) and your cutoff—e.g., $ \mathbb{E} \exp( \frac{1}{2} \int \| S \|^2 dt ) < \infty $ along the path—so that the KL steps in (5) are justified.
6. Use the square‑root form where appropriate. In several places the proof appears to use a bound of the type $\| \nabla \log p_t \| \lesssim (1/t + K) \log (\sup p_{t/2} / p_t )$, whereas standard Hamilton/Li-Yau give $\| \nabla \log p_t \|^2$ bounded by that expression. Ensure constants in, e.g., (11), remain valid with the square root in place.
7. Assumption (A3) implementability. (A3) upper‑bounds $\| s_{\theta} (t_k , x) \ |$ using the unknown true score $\| \nabla \log p_{t_k} (x) \|$. Clarify how clipping that depends only on $s_{\theta}$ guarantees (A3) without access to the ground‑truth score; or restate (A3) as a condition on $s_{\theta}$ alone and show it suffices for all places where (A3) is invoked.
8. Make the $\omega^4$ step explicit. When concluding $\mathrm{Pr}(d(Y_y,Y_{t_k})) > \omega /3 \leq e^{-c\omega^2 / (t_k - t)} \leq \omega^4$, spell out the exact regime relating $h$ and $\omega$. This will help readers verify the scheduling choices that keep this probability polynomially small.
9. Cite mixing to $\mu$ precisely. The initialization/mixing bound uses the spectral‑gap‑driven decay of the heat semigroup—please cite Urakawa and state clearly how the "$A$" constant is controlled.

**Questions:**

1. Reset term: Can you provide an explicit bound for
$$\sum^N_{k=1} \mathrm{Pr} (\| \Delta_k \| > h^{1/4}) $$
and show how it is absorbed into the final TV error? A short lemma that combines drift estimates from Lemma 11 with BDG would suffice.
2. Generator sign: Which generator (and adjoint) is used in §E.2 for the parametrix? Please unify the sign and confirm that Lemma 19 and the Volterra‑series constants remain valid under the corrected operator.
3. Geometric hypotheses: Please state (or the exact curvature condition you need) upfront, and propagate the dependence carefully through Lemmas 7/11 and Theorem 1.
4. Heat‑kernel sup: Re‑derive the $ \mathbb{E} [\log^4(\cdot)]$ control using a Li–Yau‑style $\log \sup H(t) \lesssim \frac{d}{2} \log (1/t) + CKt$. Does the main scaling change? If not, please include the corrected calculation.
5. Girsanov/Novikov: Please add a one‑page lemma verifying Novikov (or an equivalent condition) in your setting with cutoff. This would fully justify (5).
6. (A3) implementability: How can one enforce (A3) in practice without access to $\nabla \log p_t$ ? If you intend (A3) to be a theoretical condition only, please say so and explain where a weaker data‑dependent clipping suffices.
7. Schedule details: Please state the exact relationship among $h$, $\omega$, $\delta$ used to obtain the inequality $\mathrm{Pr}(d > \omega /3) \leq \omega^4$.
8. Can you relate your work to https://proceedings.mlr.press/v251/sakamoto24a.html (https://openreview.net/forum?id=ahVFKFLYk2) ?

---

> ### Author Response · Authors · 2025-11-25
>
> We thank the reviewer for the careful and thorough review and provide the explanation below.
>
> - **Reset-analysis mismatch / uniform bound on resets.** We already controlled the reset probability in Appendix E.3, Eqn (18). In particular, we have shown that the probability is at least exponentially small: it is upper bounded by $\\exp(-c\\iota^2 / h) = \\exp(-c'h^{-1/2})$, which can be made negligible by choosing a small stepsize $h$.
> - **Parametrix generator sign / consistency.** Our notation conventions are already fixed in Section 2, where we defined $\\Delta_{\\mathcal{M}}f := \\nabla_\\alpha \\nabla^\\alpha f$. The Stratonovich SDE is stated with this sign, and the parametrix/Volterra constructions follow Berline–Getzler–Vergne with consistent signs. We will add a one-sentence reminder near Lemma 21 for added clarity.
> - **Heat-kernel sup bound.** Please refer to our Lemma 7–11, where Li–Yau/Hamilton/Han–Zhang inequalities and the related constants are stated explicitly.
> - **State geometric assumptions precisely.** We already explictly assumed bounded geometry: positive injectivity radius and uniform bounds on the Riemannian tensor $\\mathrm{Rm}$ and its derivatives up to second order. This implies, in a straightforward way, the Ricci bounds required for all cited parabolic estimates, since Ricci curvature is a contraction of the Riemmanian tensor $\\mathrm{Rm}$. This assumption also appeared in the classical reference Hamilton (1996).
> - **Girsanov conditions.** This is easily implied by our assumptions, and even without the assumption, the desired inequality can be proved by a localization argument, which is nowadays standard in literature, as we have already cited in our paper (Chen et al. 2023; De Bortoli et al. 2022).
> - **Use square-root form where appropriate**. Since we are only concerned about upper bounds of $\\|\\nabla \\log p_t\\|$ in the places you mentioned, we can always make use of the elementary fact that $|x| \\le |x|^2 + 1$ to transfer the upper bounds on $\\|\\nabla \\log p_t\\|^2$ given by Li-Yau type estimates to upper bounds on $\\|\\nabla \\log p_t\\|$.
> - **(A3) implementability**. This is not a problem since (A3) is merely an *upper bound* for the estimated score, and the appearance of the true score on the right hand side only relaxes the bound and makes it *easier* (rather than harder) to satisfy (A3). For example, it is clear that clipping the score with a large polynomial would guarantee (A3).
> - **Make the $\\omega^4$ step explicit**. Inequality (9) easily holds as long as $h \\le c\\omega^2 / \\log(1 / \\omega)$. This is clearly true under our assumption that $h$ is polynomially small.
> - **Cite mixing to $\\mu$ precisely.** We have already done this in our proof. Please refer to Lemma 13 and Lemma 14.

---

### Meta-Review · Area_Chair_ZrVk · 2025-12-22

**Summary:**

This paper provides analytical results on Riemannian diffusion models, showing that a polynomially small stepsize guarantees small sampling error in TV distance if certain curvature assumptions are made. Reviewers were initially split, and only one responded in time. The main concerns influencing my decision were

- Several technical concerns with steps of the analysis and proofs
- Lacking discussion of related work by Urakawa/Sakamoto et al.
- Complete lack of experiments and comparisons to other algorithms in initial version
- Complete lack of argument that the curvature assumptions are true of real data
- Lacking discussion of why RDMs need to be studied when several papers have shown Euclidean DMs learn sub-manifold structure anyway.
- Lack of practical applications and implications of this work, given that past theoretical and empirical work has already shown RDMs do converge quickly in practice.

**Reviewer Concerns:**

- Proofs (addressed): The technical concerns with proofs were answered, and originally came with low confidence
- Related works (outstanding): The authors did not comment on the related work by Sakamoto et al.
- Experiments (partially addressed): The authors added experiments that verify some of their analytical results in toy settings, but did not add experiments that position this work more broadly. The authors responded that they would only consider the RSGM algorithm for experiments, which is certainly a limitation that was not shared by other analytical papers in this field.
- Assumptions (outstanding): The authors made no attempt to verify that the assumptions their theorems rely on are likely to hold for real data. Instead they appealed to past work saying such assumptions are "straightforward". However, I agree with the reviewer that not verifying these assumptions could potentially mean that the authors' work fails to connect with reality.
- Comparison to EuclideanDM (outstanding): The authors were unable to provide insight on whether the bounds get better for small times, but argued that their work is in a complementary direction to EuclideanDMs
- Implications (outstanding): The authors provide no actionable steps to improve the behaviour, training, sampling, or overall performance of RSGM. Past work that ran experiments already showed that they can work in practice before this paper's theoretical tools were known. Hence, this work is entirely theoretical and has no broader implications for the field.

From the discussion it is clear that this work is nearly exclusively one of theoretical analysis with minimal impact on the majority of the field. However, within the scope of analytical papers, it does advance analytical tools available and justifies previous empirical observations. Hence, I am recommending acceptance, but encourage the authors in the future to put more effort into connecting their work with reality, especially by justifying that their technical assumptions are reasonable (by evaluating them on data) and not chosen simply to advance a proof.

**Reviewer Scores:**

FiF9 - 2 -> 6 (no response, but only put forward technical concerns that were answered)

5iY7 - 6 -> 6 (responded that points were addressed)

qEnZ - 4 -> 4 (no response, had broader concerns that were not answered)

---

### Decision · Program_Chairs · 2026-01-26

Accept (Poster)